# Pruning at Initialisation through the lens of Graphon Limit: Convergence, Expressivity, and Generalisation

Hoang Pham [1]   The-Anh Ta [2]   Long Tran-Thanh [1]

## Abstract

Pruning at Initialisation methods discover sparse, trainable subnetworks before training, but their theoretical mechanisms remain elusive. Existing analyses are often limited to finite-width statistics, lacking a rigorous characterisation of the global sparsity patterns that emerge as networks grow large. In this work, we connect discrete pruning heuristics to graph limit theory via graphons, establishing the *graphon limit of PaI masks*. We introduce a *Factorised Saliency Model* that encompasses popular pruning criteria and prove that, under regularity conditions, the discrete masks generated by these algorithms converge to deterministic bipartite graphons. This limit framework establishes a novel topological taxonomy for sparse networks: while unstructured methods (e.g., Random, Magnitude) converge to homogeneous graphons representing uniform connectivity, data-driven methods (e.g., SNIP, GraSP) converge asymptotically to heterogeneous graphons that encode implicit feature selection. Leveraging this continuous characterisation, we derive two consequences. First, we prove a universal approximation theorem for sparse networks on active coordinate subspaces. Second, under the Graphon-NTK lazy-training regime, we connect the limiting graphon to NTK-style generalisation bounds and introduce a path-density interpretation of how sparse topology can modulate kernel alignment. Our results transform the study of sparse neural networks from combinatorial graph problems into a rigorous framework of continuous operators, offering a new mechanism for analysing expressivity and generalisation in sparse networks.

[1]Department of Computer Science, University of Warwick [2]CSIRO's Data61. Correspondence to: Hoang Pham <hoang.pham@warwick.ac.uk>.

*Proceedings of the 43rd International Conference on Machine Learning*, Seoul, South Korea. PMLR 306, 2026. Copyright 2026 by the author(s).

## 1. Introduction

Pruning at Initialisation (PaI) has established itself as a powerful paradigm for efficient deep learning, enabling the discovery of sparse, trainable subnetworks without the prohibitive cost of dense pre-training (Lee et al., 2019b; Wang et al., 2020; Tanaka et al., 2020; Liu et al., 2022; Pham et al., 2023; Xiang et al., 2025). Inspired by the Lottery Ticket Hypothesis (LTH) (Frankle & Carbin, 2018; Frankle et al., 2020; Zhou et al., 2019), PaI methods such as SNIP (Lee et al., 2019b), GraSP (Wang et al., 2020), and SynFlow (Tanaka et al., 2020) utilise initialisation data to identify pruning masks for producing sparse subnetworks whose performance often matches dense networks. Despite their empirical success, the theoretical nature of the resulting sparse architectures remains opaque (Pham et al., 2025). Prior works have made brilliant strides in analysing global sparse structure using finite-width combinatorial metrics, such as effective path and node balancing (Pham et al., 2023), path kernels (Gebhart et al., 2021; Patil & Dovrolis, 2021), and graph-theoretic connectivity properties (Hoang et al., 2023). However, translating these discrete, finite-width heuristics into a rigorous continuum limit remains an open challenge. A fundamental question is whether the discrete, algorithmic masks produced by these methods converge to stable mathematical objects as network width grows, or whether they remain fundamentally chaotic.

Understanding this asymptotic behaviour is necessary for developing a rigorous theory of sparse deep learning. While classical results rely on fixed random structures (e.g., Erdős-Rényi graphs) (Yang & Wang, 2023), modern PaI methods induce data-dependent correlations between input features and network weights. Recently, (Pham et al., 2025) proposed the Graphon Limit Hypothesis (GLH), postulating that pruning masks converge to continuous graphon operators, which are limit objects of convergent graph sequences (Lovász & Szegedy, 2006; Borgs et al., 2008). In particular, graphon provides a natural framework for describing large-scale network topology. Then, Pham et al. (2025) leveraged this graphon existence assumption to analyse optimisation dynamics via a Graphon-NTK - a sparse network counterpart of the fundamental Neural Tangent Kernel (NTK) of dense models (Jacot et al., 2018; Arora et al., 2019b; Lee et al., 2019a). While this is already a significant conceptual

leap, this hypothesis remains an assumption rather than a derived consequence of the algorithms themselves. A rigorous characterisation of which specific graphons arise from pruning criteria, and how these limit objects govern the network's expressivity and generalisation capabilities beyond optimisation, is still missing.

In this paper, we fill this gap by providing the first rigorous derivation of the graphon limits for a broad class of popular PaI methods. By modelling saliency scores via a newly proposed Factorised Saliency Model, we prove that under standard infinite-width conditions, the discrete masks generated by these PaI algorithms converge in probability to deterministic bipartite graphons. This result validates the GLH (Pham et al., 2025) and provides a mathematical foundation for treating sparse networks as discretisations of continuous operators.

Our analysis reveals that the limit graphon acts as a distinct asymptotic signature (in cut metric) for pruning algorithms, distinguishing methods whose masks appear similar at finite widths. We show that methods like Random and Magnitude converge to *homogeneous graphons* (constant functions). This implies that in the infinite-width limit, these methods essentially perform uniform downscaling weight matrices, which does not induce data-dependent structure beyond uniform sparsity. In contrast, gradient-based PaI methods like SNIP, or GraSP converge asymptotically to *heterogeneous graphons*. Specifically, the connectivity probability between layers is governed by input features $\varphi(u)$ and output neuron sensitivity $\psi(v)$. This mathematically formalises how data-dependent pruning performs implicit feature selection, concentrating connectivity density on "active" coordinates while filtering out noise. By lifting the analysis from discrete graphs to continuous graphons, we derive properties that are difficult to see in the finite regime. To summarise, our main contributions are:

- **Rigorous convergence analysis of PaI masks towards graphon limit**: We prove that masks generated by factorised saliency scores converge in the cut-norm metric to a deterministic limit graphon (Theorem 4.7). We further show how methods like SNIP and GraSP admit asymptotically rank-equivalent approximations that satisfy this model, formally justifying the GLH.

- **Universal approximation theorem for sparse networks**: We address the expressivity of sparse networks in the high-dimensional limit. We prove that networks sparsified by graphons retain the universal approximation capabilities of dense networks, provided the target function lies within the "active" coordinate subspace discovered by the saliency scores. By concentrating the sparsity budget on relevant features, data-driven pruning inherently preserves the necessary dense computational cores that Random pruning destroys at extreme sparsities.

- **Graphon-modulated generalisation interpretation.** Building on the Graphon-NTK framework (Pham et al., 2025), we show how the limiting graphon enters standard NTK-style generalisation bounds (Cao & Gu, 2019; Arora et al., 2019a) through the induced kernel complexity term. We further introduce a path-density proxy that interprets how sparse topology redistributes input-output connectivity. This provides a mechanistic explanation, supported by empirical evidence, for why data-dependent pruning can outperform uniform Random pruning at extreme sparsity.

## 2. Related works

**Sparse neural networks.** The observation that dense networks contain sparse, trainable subnetworks was popularised by the Lottery Ticket Hypothesis (Frankle & Carbin, 2018; Frankle et al., 2020; Burkholz, 2022a;b). To identify these subnetworks without expensive pre-training, Pruning at Initialisation (PaI) approach is getting attention. In particular, various "single-shot" criteria have been proposed, including Magnitude Pruning (Han et al., 2015), which selects weights based on absolute value, while data-driven methods like SNIP (Lee et al., 2019b), GraSP (Wang et al., 2020) utilise gradient information to estimate connection sensitivity. Other PaI methods, such as (Tanaka et al., 2020; Liu et al., 2024), iteratively prune the network until the targeted sparsity is achieved. Later, (Frankle et al., 2021; Su et al., 2020) empirically observe that applying layer-wise connection shuffling or weight reinitialisation on subnetworks found by those PaI methods retains subnetworks' performance. This raises questions about what the actual aspects are in finding lottery tickets. Prior works analyse the global structure using finite-width combinatorial metrics, including discrete path/node counting (Pham et al., 2023), path kernel (Gebhart et al., 2021; Patil & Dovrolis, 2021), and graph-theoretic properties (Hoang et al., 2023). Our novelty lies in transitioning to rigorous continuous limits through the graphon object. By characterising the asymptotic graphon limit of these masks, we provide a theoretical tool to analyse the effectiveness of sparse networks.

**Neural tangent kernels and generalisation error bounds.** The behaviour of neural networks as widths $n \to \infty$ is typically analysed through the lens of the NTK (Jacot et al., 2018; Arora et al., 2019b; Lee et al., 2019a;a). In this regime, under specific parameterisations, the network evolves as a linear model over a fixed kernel $\Theta_\infty$. Extensive work has established generalisation bounds for this regime. Arora et al. (2019a); Cao & Gu (2019) derived rigorous bounds scaling as $O(\sqrt{y^\top \Theta_\infty^{-1} y / m})$ where $m$ is the number of training samples. Crucially, these bounds depend on the Reproducing Kernel Hilbert Space (RKHS) norm of the target function, implying that generalisation is governed by the alignment between the fixed kernel and the target task. While powerful, standard NTK analysis typically assumes

dense connectivity.

**Graphon limit of sparse neural networks.** Recently, Pham et al. (2025) postulated the Graphon Limit Hypothesis and defined the "Graphon-NTK" to analyse the optimisation dynamics of pruned networks. In particular, Pham et al. (2025) assumed that pruning masks converge to graphons when the network's widths tend to infinity. The concept of graphons, limit objects describing the limits of convergent sequences of graphs, is used to analyse the behaviour of large graphs (Lovász & Szegedy, 2006; Borgs et al., 2008). In the machine learning domain, graphons have primarily been applied to Graph Neural Networks (GNNs) (Ruiz et al., 2020; Keriven et al., 2020; Krishnagopal & Ruiz, 2023), which utilised graphons to define convolution operations that are stable across graphs of varying sizes, effectively enabling "transfer learning" for GNNs on large-scale topological data. Our application differs fundamentally. Rather than using graphons to model input data or statistical networks, we employ them to model the internal architecture of the neural network itself. While Pham et al. (2025) recently introduced this perspective to analyse training dynamics, we deepen this connection by proving that the specific bipartite structures induced by PaI algorithms correspond to known classes of graphons.

**Approximation theory for neural networks.** Classical universal approximation theorems, which show that sufficiently large dense networks can well approximate any continuous function on a bounded domain, are well-established (Cybenko, 1989; Hornik, 1991; Barron, 1994; Eldan & Shamir, 2016; Nakada & Imaizumi, 2020). In contrast, the theoretical capability of sparse networks to approximate functions has been studied primarily through two lenses: constructive existence and random sparsity. Bolcskei et al. (2019) established fundamental lower bounds on the connectivity required to approximate specific function classes, often relating optimal sparse architectures to affine systems like wavelets and shearlets. While these results prove that optimal sparse networks exist, they do not guarantee that standard pruning algorithms can find them. Conversely, the "Strong Lottery Ticket Hypothesis" literature led by Malach et al. (2020); Orseau et al. (2020), focuses on the probability that a sufficiently wide random network contains a good sparse subnetwork that, without training, still approximates the target function with high accuracy. Instead of assuming a generic random mask or a handcrafted affine system, we derive the approximation specifically for the sparse networks induced by limit graphons.

# 3. Preliminaries and settings

## 3.1. Preliminaries

Graphons provide a continuous representation of large discrete graphs (Lovász & Szegedy, 2006; Borgs et al., 2008). For bipartite networks connecting input coordinates to hidden neurons, we work with measurable kernels $\mathcal{W} : [0,1]^2 \to [0,1]$ that encode the connectivity pattern between an infinite set of nodes (Pham et al., 2025).

**Bipartite graphons and step kernels.** Given a matrix $A \in [0,1]^{d \times n}$, we define its associated *step kernel* $W_A : [0,1]^2 \to [0,1]$ by $W_A(u,v) := A_{ij}$ for $u \in I_i^{(d)} := \left( \frac{i-1}{d}, \frac{i}{d} \right]$, $v \in J_j^{(n)} := \left( \frac{j-1}{n}, \frac{j}{n} \right]$. This construction provides the standard embedding of bipartite adjacency matrices into the space of measurable kernels on $[0,1]^2$.

**Cut norm and cut distance.** The cut norm measures the maximum discrepancy in average mass over all measurable rectangles. For an integrable kernel $\mathcal{W} : [0,1]^2 \to [0,1]$, the *cut norm* is defined as

$$\|\mathcal{W}\|_\square := \sup_{S,T \subseteq [0,1] \text{ measurable}} \left| \int_{S \times T} \mathcal{W}(u,v) \, du \, dv \right|. \quad (1)$$

The *bipartite cut distance* between kernels $\mathcal{W}$ and $\mathcal{U}$ is

$$\delta_\square^{\text{bip}}(W,U) := \inf_{\phi,\psi} \|\mathcal{W} - \mathcal{U}^{\phi,\psi}\|_\square \quad (2)$$

where $\mathcal{U}^{\phi,\psi}(u,v) := \mathcal{U}(\phi(u),\psi(v))$ and the infimum is over all measure-preserving maps $\phi, \psi : [0,1] \to [0,1]$. Crucially, in the bipartite setting, row and column relabelings are applied *independently*, unlike the symmetric graphon setting. Hence, we may sort rows/columns by latent features to reveal structure.

## 3.2. Setting and notations

We study pruning-at-initialisation in a one-hidden-layer network with hidden width $n$ on input $x \in \mathbb{R}^d$

$$f_n(x) = \frac{1}{\sqrt{n}} \sum_{j=1}^n \theta_j^{(2)} \sigma(h_j^{(1)}), \qquad h_j^{(1)} := \frac{1}{\sqrt{d}} \sum_{i=1}^d \theta_{ij}^{(1)} x_i,$$

and squared loss on a fixed label $y \in \mathbb{R}$: $\mathcal{L} = \frac{1}{2} (f_n(x) - y)^2$, with i.i.d. Gaussian initialization $\theta_{ij}^{(1)} \sim \mathcal{N}(0,1)$, i.i.d. output weights $\theta_j^{(2)} \sim \mathcal{N}(0,1)$, independent of $\theta_{ij}^{(1)}$. Here, the first layer weight matrix is $\theta^{(1)} \in \mathbb{R}^{d \times n}$. We focus on *PaI sparsification of the first layer*: a binary mask $M_n \in \{0,1\}^{d \times n}$ is computed at initialisation and applied elementwise to $\theta^{(1)}$ as $\theta^{(1)} \leftarrow \theta^{(1)} \odot M_n$, After this, training proceeds only on the active parameters. Our theoretical results concern the asymptotic structure of the mask $M_n$ as $d, n \to \infty$.

**Masks as bipartite graphs and step-kernel embedding.** The mask $M_n$ can be viewed as the adjacency matrix of a bipartite graph between input coordinates and hidden units. To study limits as $d, n \to \infty$, we associate $M_n$ with a step kernel $W_{M_n} : [0,1]^2 \to \{0,1\}$ defined by $W_{M_n}(u,v) = (M_n)_{ij}$ for $u \in \left( \frac{i-1}{d}, \frac{i}{d} \right]$, $v \in \left( \frac{j-1}{n}, \frac{j}{n} \right]$. We measure

convergence of these step kernels using the *bipartite cut distance*. From now on, we use $W_n$ interchangeably with $W_{M_n}$ for brevity.

**Other notations.** We write $[k] := \{1, \ldots, k\}$ for any $k \in \mathbb{N}^+$. $A_{n,ij}$ or $(A_n)_{ij}$ denotes the $(i, j)$ entry of size $n$ matrix, and $\odot$ denotes Hadamard multiplication.

## 4. Graphon convergence of PaI methods

Pruning-at-Initialisation methods output *binary mask matrices* $M_n \in \{0, 1\}^{d \times n}$ indicating which connections survive. However, analysing $M_n$ directly is non-trivial due to (i) the mask is a high-dimensional combinatorial object, and (ii) reordering neurons produces different masks with identical functionality. Graphon theory provides a principled way to talk about what such masks look like at the large-width limit. Concretely, we embed $M_n$ into a step kernel $W_{M_n}$ on $[0, 1]^2$ and measure convergence in the (bi)partite cut distance, which compares two kernels *after optimising over all measure-preserving transformations*. In this section, we ask whether the mask has a stable *large-scale connectivity pattern* up to relabeling, i.e. whether $W_{M_n}$ converges in cut distance to a deterministic graphon $\mathcal{W}$.

### 4.1. The factorised saliency model

Our analysis is driven by a simple structural observation shared by many PaI criteria where edge saliency often decomposes into an *input feature* (forward signal), a *neuron feature* (backward/gradient signal), and an *edge-level* randomness from initialisation. We formalise this via the factorised saliency model:

$$(S_n)_{ij} = \varphi_{n,i} \cdot \psi_{n,j} \cdot |\xi_{n,ij}|, \tag{3}$$

$$(M_n)_{ij} = \mathbf{1}\{(S_n)_{ij} > \tau_n\}, \quad i \in [d],\ j \in [n], \tag{4}$$

where $\tau_n > 0$ is a pruning threshold; $\varphi_{n,i} \geq 0$ is *input feature*, capturing forward-pass saliency of input coordinate $i$; $\psi_{n,j} \geq 0$ is *neuron feature*, capturing backward-pass (gradient) saliency of hidden unit $j$; $\xi_{n,ij}$ is *edge noise*, capturing initialisation randomness at edge $(i, j)$. This model helps us state and prove the graphon convergence theorem, which can apply to various PaI methods.

**PaI instantiations.** Many popular PaI methods fit this model either exactly (e.g., Magnitude/Random at i.i.d. initialisation), or asymptotically ranking-equivalent (e.g., SNIP/GraSP in the infinite width regimes) in the one-hidden-layer network setting. Table 1 summarises the corresponding $(\varphi, \psi, \xi)$ choices. Please refer to Appendix B, C for detailed derivations.

*Magnitude/Random.* Magnitude pruning sets $(S_n)_{ij} = |\theta_{ij}^{(1)}|$, thus $\varphi_i \equiv \psi_j \equiv 1$. Random pruning (Bernoulli($\rho$)) corresponds to $\varphi_i \equiv \psi_j \equiv 1$ with $\xi_{ij} \sim \mathrm{Unif}[0, 1]$.

*SNIP (Lee et al., 2019b).* With $f_n(x) = \frac{1}{\sqrt{n}} \sum_{j=1}^{n} \theta_j^{(2)} a_j^{(1)}$ and $h_j^{(1)} = \frac{1}{\sqrt{d}} \sum_{i=1}^{d} \theta_{ij}^{(1)} x_i$, SNIP introduces masks $m_{ij}$ via $\theta_{ij}^{(1)} \mapsto m_{ij} \theta_{ij}^{(1)}$ and evaluates at $m \equiv \mathbf{1}$. By the chain rule,

$$S_{ij}^{\mathrm{SNIP}} = \left| \frac{\partial \mathcal{L}}{\partial m_{ij}} \right| = \frac{|\delta|}{\sqrt{nd}}\, |x_i|\, \left( |\theta_j^{(2)}|\, |\sigma'(h_j^{(1)})| \right) |\theta_{ij}^{(1)}|, \tag{5}$$

where $\delta$ denotes the scalar loss derivative w.r.t. the network output (e.g., $\delta = f_n(x) - y$ for squared loss). Absorbing the global factor $|\delta|/\sqrt{nd}$ into $\tau_n$ yields the scale-free factorisation in Table 1.

*GraSP (Wang et al., 2020) (magnitude variant).* GraSP assigns a score proportional to $\theta^{(1)} \odot (Hg)$ where $g = \nabla_{\theta^{(1)}} \mathcal{L}(\theta^{(1)})$ and $H = \nabla_{\theta^{(1)}}^2 \mathcal{L}(\theta^{(1)})$. To align with our non-negative thresholding Eqn.4, we use the magnitude variant $S_{ij}^{\mathrm{GraSP}} := |\theta_{ij}^{(1)} (Hg)_{ij}|$[1]. In the one-hidden-layer setting, we have $g_{ij} = \partial \mathcal{L}/\partial \theta_{ij}^{(1)} = \delta \frac{1}{\sqrt{n}} \theta_j^{(2)} \sigma'(h_j^{(1)}) \frac{1}{\sqrt{d}} x_i$. For squared loss, $H = J^\top J + \delta \nabla_{\theta^{(1)}}^2 f_n$ with $J = \nabla_{\theta^{(1)}} f_n$. A direct computation yields $(J^\top J g)_{ij} = c_n g_{ij}$ where $c_n := \frac{\|x\|_2^2}{d} \frac{1}{n} \sum_{k=1}^{n} (\theta_k^{(2)})^2 \sigma'(h_k^{(1)})^2$. Under infinite width setting, the term $\delta(\nabla_{\theta^{(1)}}^2 f_n)g$ contributes only lower-order corrections, hence $(Hg)_{ij} = c_n g_{ij} + o(1)$ entrywise. Thus, up to a global positive factor (irrelevant for top-$\rho$ ranking),

$$S_{ij}^{\mathrm{GraSP}} \approx \varphi_i^{(n)} \psi_j^{(n)} |\xi_{ij}^{(n)}| = |x_i|\, (|\theta_j^{(2)}|\, |\sigma'(h_j^{(1)})|)\, |\theta_{ij}^{(1)}|. \tag{6}$$

*SynFlow (Tanaka et al., 2020) (one iteration).* SynFlow uses the data-agnostic synaptic saliency $S(\theta) = (\partial R/\partial \theta) \odot \theta$ with $R = \mathbf{1}^\top (\prod_{\ell=1}^{L} |\theta^{(\ell)}|) \mathbf{1}$. In a single-hidden-layer network, one iteration yields $S_{ij} \propto |\theta_j^{(2)}|\, |\theta_{ij}^{(1)}|$, corresponding to $\varphi_i \equiv 1$ and $\psi_j = |\theta_j^{(2)}|$.

### 4.2. Regularity assumptions

We establish convergence as $d \to \infty$ and $n \to \infty$ under the following conditions:

**Assumption 4.1** (Asymptotic width scaling). $\frac{\log(d+n)}{\min\{d,n\}} \to 0$.

**Assumption 4.2** (Deterministic input features). There exists a bounded continuous function $\varphi : [0, 1] \to [0, \infty)$ such that, after a permissible relabeling of input coordinates, $\max_{1 \leq i \leq d} |\varphi_{n,i} - \varphi(i/d)| \xrightarrow{\mathbb{P}} 0$.

**Assumption 4.3** (Neuron features empirical CDF convergence). Let $\widehat{F}_{\psi,n}(t) := \frac{1}{n} \sum_{j=1}^{n} \mathbf{1}\{\psi_{n,j} \leq t\}$. There exists a deterministic CDF $F_\psi$ supported on $[0, \infty)$ such that

$$\sup_{t \in \mathbb{R}} |\widehat{F}_{\psi,n}(t) - F_\psi(t)| \xrightarrow{\mathbb{P}} 0.$$

**Assumption 4.4** (Conditional independence of edge noise). Let $\mathcal{G}_n := \sigma(\{\varphi_{n,i}\}_{i \leq d}, \{\psi_{n,j}\}_{j \leq n}, \tau_n)$. Conditional on

---

[1]We provide analysis for original version in Appendix C

$\mathcal{G}_n$, the noises $\{\xi_{n,ij}\}_{i \leq d,\, j \leq n}$ are independent with common law $P_\xi$, and are independent of $\mathcal{G}_n$. Moreover, the CDF of $|\xi|$ on $[0, \infty)$ is continuous.

**Assumption 4.5** (Deterministic threshold). There exists $\tau \in (0, \infty)$ such that $\tau_n \xrightarrow{P} \tau$, and $\tau_n$ is $\mathcal{G}_n$-measurable.

*Remark* 4.6. Our aim is to show that the masked weight matrix $M_n$ behaves, at large $(d, n)$, like a bipartite random graph generated from a *deterministic* graphon $\mathcal{W}(u, v)$: the row index $i$ becomes a continuum coordinate $u = i/d$, the column index $j$ becomes $v = j/n$, and edges are kept with a probability that depends smoothly on the product of a row feature and a column feature. Assumptions 4.2 - 4.3 provide exactly these stable continuum coordinates by requiring that (after relabelling) the input (row) features $\{\varphi_{n,i}\}$ track a deterministic function $\varphi(u)$ and the neuron (column) features $\{\psi_{n,j}\}$ have a deterministic limiting distribution (hence order statistics $\psi_{n,(j)} \approx Q_\psi(v)$, where $Q_\psi(v) := \inf\{y \geq 0 : F_\psi(y) \geq v\}$ is the generalised quantile). Assumption 4.4 isolates the remaining randomness to i.i.d. edge noise $\{\xi_{n,ij}\}$ conditional on $(\varphi_n, \psi_n, \tau_n)$, which turns the mask into a conditionally independent Bernoulli matrix and enables cut-norm concentration around its conditional mean. Assumption 4.5 says that the top-$\rho$ threshold stabilises to a constant $\tau$, so the limiting edge-probability kernel is well-defined and does not inherit extra global randomness. Finally, assumption 4.1 is a mild growth condition ensuring that these concentration bounds hold *uniformly*, so the full bipartite cut distance converges.

**SNIP is asymptotically equivalent to a factorised model.** In the 1-hidden layer SNIP setting in Eqn.5, the factorisation Eqn.3 holds after absorbing the global factor into $\tau_n$, with $\varphi_{n,i} = |x_i|$, $\psi_{n,j} = |\theta_j^{(2)}| |\sigma'(h_j^{(1)})|$, and $\xi_{n,ij} = \theta_{ij}^{(1)}$. If the input coordinates are i.i.d. and bounded, then after sorting $\{|x_i|\}_{i \leq d}$ one has $\max_i \left| |x|_{(i)} - Q_{|X|}(i/d) \right| \to 0$ in probability, verifying A4.2 with $\varphi = Q_{|X|}$. Under infinite width regime, conditional on $x$ the pre-activations $h_j^{(1)}$ are approximately Gaussian with variance $\|x\|_2^2/d$, yielding a deterministic limit law for $\psi_{n,j}$ and hence A4.3. Moreover, although $\theta_{ij}^{(1)}$ appears inside $\psi_{n,j}$ through $h_j^{(1)}$, the influence of a single entry $\theta_{ij}^{(1)}$ on $h_j^{(1)}$ vanishes asymptotically as $d \to \infty$, satisfying A4.4. Finally, taking $\tau_n$ as the empirical $(1 - \rho)$-quantile of $\{(S_n)_{ij}\}_{i,j}$ yields A4.5 under mild continuity conditions. We provide full verifications in the Appendix B.

### 4.3. Graphon convergence of factorised masks

**Theorem 4.7** (Bipartite graphon convergence). *Let $d \to \infty$ and $n \to \infty$. Consider the factorised saliency model*

$$(S_n)_{ij} = \varphi_{n,i} \cdot \psi_{n,j} \cdot |\xi_{n,ij}|, \quad (M_n)_{ij} = \mathbf{1}\{(S_n)_{ij} > \tau_n\},$$

*where $i \in [d]$, $j \in [n]$, and let $W_n := W_{M_n}$ be the associated bipartite step-kernel on $[0,1]^2$. Under assumptions 4.1 - 4.5, we define the deterministic limit graphon*

*Table 1.* Factorisation of PaI methods. SNIP (Lee et al., 2019b) and GraSP (Wang et al., 2020) are different in global scaling.

| Method | Input $\varphi_i$ | Neuron $\psi_j$ | Noise $|\xi_{ij}|$ |
|---|---|---|---|
| SNIP | $|x_i|$ | $|\theta_j^{(2)}| |\sigma'(h_j^{(1)})|$ | $|\theta_{ij}^{(1)}|$ |
| GraSP | $|x_i|$ | $|\theta_j^{(2)}| |\sigma'(h_j^{(1)})|$ | $|\theta_{ij}^{(1)}|$ |
| Synflow | 1 | $|\theta_j^{(2)}|$ | $|\theta_{ij}^{(1)}|$ |
| Magnitude | 1 | 1 | $|\theta_{ij}^{(1)}|$ |
| Random | 1 | 1 | $\mathrm{Unif}[0, 1]$ |

$$\mathcal{W}(u, v) := \mathbb{P}(\varphi(u) \cdot Q_\psi(v) \cdot |\xi| > \tau), \quad (u, v) \in [0, 1]^2.$$
*Then $\delta_\square^{\mathrm{bip}}(W_n, \mathcal{W}) \xrightarrow{\mathbb{P}} 0$.*

*Remark* 4.8. The limit graphon $\mathcal{W}$ can be viewed as a topological signature of a PaI method. Unstructured pruning rules, such as Random or Magnitude pruning, converge to constant graphons $\mathcal{W}(u, v) \equiv \rho$, reflecting uniform connectivity across inputs and neurons. In contrast, data-dependent PaI methods induce heterogeneous graphons, where the probability of retaining a connection varies smoothly with latent input features and neuron sensitivities. From this perspective, different PaI rules are distinguished not by individual edge realisations, but by the large-scale connectivity patterns they impose in the infinite-width limit.

*Remark* 4.9. This convergence result extends naturally to deep architectures. For an $L$-hidden-layer network with widths $(n_0, n_1, \ldots, n_L)$, each layer $\ell \in \{1, \ldots, L\}$ has mask $M^{(\ell)} \in \{0, 1\}^{n_{\ell-1} \times n_\ell}$ generated by factorised scores in Eqn. (3). The corresponding bipartite step-kernel is $W_{M^{(\ell)}}$ or $W_n^{(\ell)}$ for short. Here $\varphi_{n,i}^{(\ell)}$ is a forward feature determined by the activation of neuron $i$ in layer $\ell - 1$, while $\psi_{n,j}^{(\ell)}$ is a backward feature determined by the normalised backpropagated sensitivity of neuron $j$ in layer $\ell$. Under standard infinite-width moment assumptions, these row and column features have deterministic empirical limits after sorting, and the edge noise is asymptotically decoupled from the neuron feature by a leave-one-out argument. Therefore each layer admits a deterministic limit graphon $W^{(\ell)}$. Applying Theorem 4.7 to each layer $\ell$ gives the graphon convergence for each layer $W_n^{(\ell)} \xrightarrow{\mathbb{P}} \mathcal{W}^{(\ell)}$. Please refer to Appendix D for details.

### 4.4. Empirical verification

We empirically illustrate Theorem 4.7 by comparing finite-width masks, produced by the original SNIP, to the predicted limit graphon. For visualisation, we generate masks $M_n \in \{0, 1\}^{n \times n}$ at fixed density $\rho = 0.2$ for $n \in \{200, 500, 1000, 2000, 4000\}$. Following Pham et al. (2025), we sort rows/columns by their latent factors ($\varphi_i$ for rows and $\psi_j$ for columns) and average the sorted masks over 100 seeds to obtain an empirical edge-probability matrix.

Besides, we compute the limit theoretical graphon, $\mathcal{W}(u, v)$, as a continuous function on the unit square $[0, 1]^2$. To

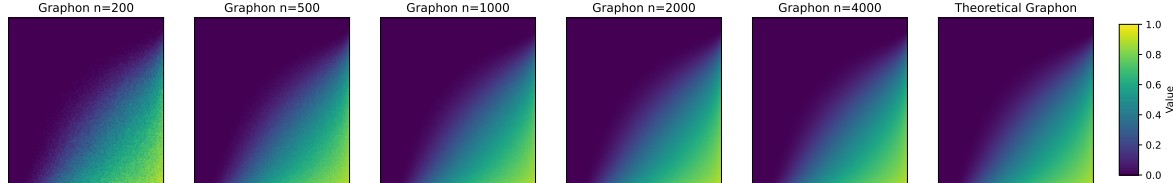

*Figure 1.* Visual convergence to the graphon limit. We compare averaged empirical masks (over 100 seeds) at increasing widths ($n = 200, 500, 1000, 2000, 4000$) against the analytically computed Theoretical Graphon. The density is fixed at $\rho = 0.2$.

make the limit object computable, we consider i.i.d. standard normal inputs and weights initialisation. Since inputs $x \sim \mathcal{N}(0,1)$, the magnitude $|x|$ follows a standard Half-Normal (HN) distribution. We define $\varphi(u) = F_{\mathrm{HN}}^{-1}(u) = \sqrt{2}\mathrm{erf}^{-1}(u)$. The column factor is defined as $\psi = |\theta^{(2)}||\sigma'(h)|$, where $\theta^{(2)} \sim \mathcal{N}(0,1)$. We determine the quantile function $Q_\psi(v)$ for this product distribution via Monte Carlo simulation ($N = 10^6$ samples). The edge noise $|\xi_{ij}|$ is Half-Normal distributed with scale $\sigma_{\theta^{(1)}} = 1$ where $\theta^{(1)} \sim \mathcal{N}(0,1)$. Given a global threshold $\tau$ (solved numerically to match density $\rho$), the theoretical edge probability is $\mathcal{W}_{(u,v)} = \mathbb{P}\left(\varphi(u)\psi(v)|\xi| > \tau\right) = 1 - \mathrm{erf}(\frac{\tau}{\varphi(u)Q_\psi(v)\sqrt{2}})$.

Figure 1 shows that the averaged empirical masks become progressively smoother and rapidly approach the theoretical $\mathcal{W}$ as $n$ increases; by $n \geq 2000$ the empirical and theoretical patterns are visually indistinguishable. This supports the interpretation of PaI masks as finite-sample discretisations of a deterministic graphon limit.

# 5. Expressivity

This section studies the approximation power of single-hidden-layer networks whose sparse connectivity is given by PaI masks. Classical Universal Approximation Theorems (UAT) rely on dense connectivity (Cybenko, 1989; Barron, 1994). For sparse networks, proving expressivity is highly non-trivial because algorithmic pruning is a random, combinatorial process that can abruptly destroy necessary computational paths.

Because the ambient input dimension satisfies $d \to \infty$, claiming a sparse model can approximate every continuous function of all $d$ coordinates is both too strong and misaligned with real-world learning problems. Instead, we adopt an intrinsic-dimension regime (Nakada & Imaizumi, 2020), where the target function depends only on a fixed (or slowly growing) subset of $k$ informative input coordinates. We ask: *does the pruned network retain universal approximation capabilities on this active subspace?*

Another challenge is that PaI masks are random, method-dependent *combinatorial* objects, so it is non-trivial to make precise connectivity claims directly at the level of the binary matrix $M_n$. Our graphon limit theory provides a deterministic continuum proxy to answer this: when the step-kernels $W_{M_n}$ converge to a limit graphon $\mathcal{W}$, large-scale connec-

tivity properties of $M_n$ can be stated as simple analytic conditions on $\mathcal{W}$. In particular, we demonstrate that if the limit graphon $\mathcal{W}$ concentrates sufficient edge probability on an active subspace, then as $n \to \infty$, with high probability the finite mask $M_n$ contains a fully connected $k \times \tilde{n}$ sub-network (with $\tilde{n} \to \infty$), allowing classical UAT (Cybenko, 1989) to apply on that core.

**Masked 1-hidden-layer networks and deterministic threshold masks.** Fix $d, n \in \mathbb{N}$. Let $M \in \{0,1\}^{d \times n}$ be a binary mask sampled from a given graphon $\mathcal{W}$ of the factorised saliency model studied in Sect. 4. We consider the class of masked two-layer neural networks

$$\mathcal{F}_n(M) := \Big\{ x \mapsto \sum_{j=1}^{n} a_j\, \sigma\,\big(b_j + \sum_{i=1}^{d} M_{ij}\, \theta_{ij}\, x_i\big)\Big\}, \quad (7)$$

where $(a_j), (b_j) \in \mathbb{R}^n, (\theta_{ij}) \in \mathbb{R}^{d \times n}$. Any conventional normalisation (e.g. $1/\sqrt{n}$) can be absorbed into $a_j$'s and does not affect expressivity.

## 5.1. Graphon coordinates and active rectangle

By the same measure-preserving relabelling used in the graphon convergence Theorem 4.7, we work in coordinates $u_i := i/d$ and $v_j := j/n$ such that $\varphi_{n,i} \approx \varphi(u_i)$ and the sorted neuron features satisfy $\psi_{n,(j)} \approx Q_\psi(v_j)$. In these coordinates, the limiting edge-probability kernel takes the explicit form $\mathcal{W}(u,v) = \mathbb{P}(\varphi(u)\,Q_\psi(v)\,|\xi| > \tau)$, matching the limit object from Sect. 4. Equivalently, one may view $u_i, v_j$ as i.i.d. uniform latents upto a random relabelling, since the cut distance is invariant under index permutations.

**Class of target functions depending on $k$ coordinates.** Fix $k \in \mathbb{N}$ and a compact set $K \subset \mathbb{R}^k$. For any set $I \subset [d]$ with $|I| = k$, we write $x_I \in \mathbb{R}^k$ for the subvector of $x \in \mathbb{R}^d$ restricted to coordinates in $I$. Given $f \in C(K)$, we define the lifted target $\tilde{f} : \mathbb{R}^d \to \mathbb{R}$ by $\tilde{f}(x) := f(x_I)$, where $x \in \mathbb{R}^d$, and the lifted compact domain $\tilde{K} := \{x \in \mathbb{R}^d : x_I \in K, \|x_{I^c}\|_\infty \leq B\}$ for some fixed $B < \infty$, where $I^c := [d]\setminus I$. Note that approximation on $\tilde{K}$ is equivalent to approximation of $f$ on $K$, as $\tilde{f}$ ignores $x_{I^c}$.

**Theorem 5.1** (UAT for dense 1-hidden-layer networks on $\mathbb{R}^k$, (Cybenko, 1989))**.** *Fix $k \in \mathbb{N}$ and a compact $K \subset \mathbb{R}^k$. Suppose $\sigma$ is a universal activation. Then the class $\{u \mapsto \sum_{r=1}^{n} a_r\sigma(\theta_r^\top u + b_r) : n \in \mathbb{N}\}$ is dense in*

$C(K)$. *In particular, for every $f \in C(K)$ and every $\varepsilon > 0$, there exist an integer $\tilde{n}_\varepsilon$ and parameters $\{(a_r, \theta_r, b_r)\}_{r=1}^{\tilde{n}_\varepsilon}$ such that $\sup_{u \in K} \left| \sum_{r=1}^{\tilde{n}_\varepsilon} a_r \sigma(\theta_r^\top u + b_r) - f(u) \right| < \varepsilon$.*

**Assumption 5.2** (Lower-bounded core connectivity). There exist measurable sets $U_\star, V_\star \subset (0,1)$ with Lebesgue measures $\beta := \lambda(U_\star) > 0$ and $\alpha := \lambda(V_\star) > 0$, and a constant $p_\star > 0$ such that

$$\inf_{u \in U_\star, v \in V_\star} \mathcal{W}(u,v) \geq p_\star \qquad (8)$$

In the factorised model in Eqn. (3), a sufficient condition is that $\varphi(u) \geq \varphi_0 > 0$ on $U_\star$ and $Q_\psi(v) \geq \psi_0 > 0$ on $V_\star$, in which case one may take $p_\star = \mathbb{P}(\varphi_0 \psi_0 |\xi| > \tau) > 0$.

**Assumption 5.3** (Availability of $k$ active inputs). Given a fixed $k$, with probability $1 - o(1)$ as $d \to \infty$, there exists an index set $I \subset [d]$ such that $|I| = k$ and $u_i \in U_\star, \forall i \in I$.

**Assumption 5.4** (Core width growth). Let $k$ be fixed. As $n \to \infty$, we have $(\alpha n) p_\star^k \longrightarrow \infty$.

*Remark* 5.5. Assumption 5.2 is a *continuum* statement about the limiting connectivity pattern. It says that there is a positive-measure block of input latents $U_\star$ and hidden-unit latents $V_\star$ such that edges in that block survive with probability uniformly bounded below. For Random pruning (constant graphon $W \equiv \rho$), one may take $U_\star \times V_\star = [0,1]^2$ and $p^\star = \rho$. For data-dependent PaI (e.g. SNIP-type), $U_\star \times V_\star$ can be interpreted as the region where the signal concentrates: $U_\star$ corresponds to input coordinates with non-negligible forward saliency $\varphi$, and $V_\star$ to hidden units with non-negligible backward/gradient feature $Q_\psi$.

### 5.2. Universal approximation on active coordinates

We show that the class $\mathcal{F}_n(M)$ is universal for approximating target functions depending on $k$ active coordinates. The following lemma ensures the existence of a dense subnetwork of size $k \times \tilde{n}$ in the graphon-sparsified network.

**Lemma 5.6** (Dense core in a graphon-sparsified mask). *Assume assumptions 5.2 -5.4 and let $I$ be as in Assumption 5.3. Fix $\tilde{n} \in \mathbb{N}$. Then, with overwhelming probability, there exists $J \subset [n]$ with $|J| = \tilde{n}$ such that $M_{ij} = 1$, $\forall i \in I, \forall j \in J$.*

**Theorem 5.7** (Universality on active coordinates). *Assume Theorem 5.1 and assumptions 5.2- 5.4. Fix $k \in \mathbb{N}$, a compact $K \subset \mathbb{R}^k$, a target $f \in C(K)$. For every $\varepsilon > 0$, there exist $B < \infty$ and an index set $I$ as in assumption 5.3. Define $\tilde{f}$ and $\tilde{K}$ as above. Then, with high probability, there exists $F_n \in \mathcal{F}_n(M)$ such that $\sup_{x \in \tilde{K}} |F_n(x) - \tilde{f}(x)| < \varepsilon$*

*Remark* 5.8. Theorem 5.7 states that universality on any fixed $k$-coordinate subspace holds whenever the limit graphon $\mathcal{W}$ contains an *active block* $U_\star \times V_\star$ with a uniform edge floor $\mathcal{W} \geq p^\star > 0$; equivalently, one needs $\alpha n (p^\star)^k \to \infty$ to ensure enough hidden units connect simultaneously to $k$ active inputs. This result highlights the fundamental difference between pruning methodologies at

extreme sparsities. For Random pruning ($\mathcal{W} \equiv \rho$), this reduces to $n\rho^k \to \infty$, which can fail at high sparsity since edges are spread uniformly. In contrast, data-dependent PaI can concentrate the same sparsity budget on task-relevant coordinates, yielding a larger effective floor $p^\star$ on a non-vanishing active "sub-graphon" $U_\star \times V_\star$ (with $\alpha, \beta$ bounded below), thereby better preserving dense cores and approximation power in the high-sparsity regime.

## 6. Generalisation error bound

Generalisation error bounds for overparameterised networks depend crucially on *architecture and training regime* (Novak et al., 2018; Arora et al., 2019a). For PaI, the architecture itself is random or data-dependent through masks $M_n$, making it unclear what the right limiting object is. The graphon limit framework makes this dependence precise: if $W_{M_n} \to \mathcal{W}$, then the associated Graphon-NTK admits a deterministic limit that depends only on $\mathcal{W}$ and the data distribution (Pham et al., 2025). This lifts generalisation analysis from a random finite mask to a deterministic operator defined by $\mathcal{W}$. Here, we derive the generalisation error bound for overparameterised sparse neural networks in the Graphon-NTK regime, in parallel with the bounds for dense models in (Cao & Gu, 2019; Arora et al., 2019a). Then, we interpret this bound through the input-output path density of layer graphons, which shows how the connectivity pattern of pruning masks affects generalisation capacity.

### 6.1. From graphon limits of PaI to Graphon-NTK

Let $\mathcal{D}$ be a data distribution and let $S = \{(x_i, y_i)\}_{i=1}^m$ be i.i.d. samples from $\mathcal{D}$. As discussed in Section 4, masks generated by PaI methods converge to specific graphons, which describes the infinite-width limit of the induced sparsity structure. Given such limiting graphon $\mathcal{W}$, the authors of (Pham et al., 2025) defines the associated *Graphon-NTK kernel* $\Theta_\mathcal{W} : \mathcal{X} \times \mathcal{X} \to \mathbb{R}$ as the deterministic infinite-width limit of the NTK for graphon-induced sparse networks. On the dataset $S$, we define the *Graphon-NTK Gram matrix* $(K_\mathcal{W})_{ij} := \Theta_\mathcal{W}(x_i, x_j) \in \mathbb{R}^{m \times m}$.

### 6.2. Graphon-NTK generalisation bound

We analyse the infinite-width *Graphon-NTK (lazy) regime*, in which training dynamics of the masked network on a fixed sample are governed by the limiting $K_\mathcal{W}$. This viewpoint parallels the classical NTK linearisation results (Lee et al., 2019a; Chizat et al., 2019). Under this regime, we can directly apply the generalisation bound of Cao & Gu (2019) with $K_\mathcal{W}$ in place of the finite-width NTK Gram matrix.

**Corollary 6.1** (Infinite-width Graphon-NTK generalisation bound). *Fix $\delta \in (0, e^{-1})$ and assume $\lambda_{\min}(K_\mathcal{W}) \geq \lambda_0 > 0$. As widths $\to \infty$, with probability at least $1 - \delta$ over $S$, the expected classification error of an associated* output

predictor *of the neural network $\hat{f}$ (see Appendix F.3) satisfies*

$$\mathbb{E}\left[\mathcal{L}_{0\text{-}1}^{\mathcal{D}}(\hat{f})\right] \leq C\,L \cdot \sqrt{\frac{y^\top K_{\mathcal{W}}^{-1} y}{m}} + C\sqrt{\frac{\log(1/\delta)}{m}}, \quad (9)$$

*where $y = (y_1, \ldots, y_m)^\top$, $C > 0$ is a universal constant, and $L$ is the depth factor.*

*Remark* 6.2. The complexity term $y^\top K_{\mathcal{W}}^{-1} y$ measures the RKHS norm of the label vector under kernel $\Theta_{\mathcal{W}}$. Smaller values indicate the target labels are easier to separate in the induced feature space, directly linking pruning topology to generalisation capability. We provide proof in Appendix F.

### 6.3. Path density proxy for the Graphon-NTK in 2-hidden-layer networks ($L = 2$)

Corollary 6.1 shows that the generalisation error bound depends on the complexity term $y^\top K_{\mathcal{W}}^{-1} y$. The effect of the pruning topology can be understood through the mass of surviving input-output paths.

**A path-density proxy.** To interpret how the pruning topology affects the Graphon-NTK complexity term in Corollary 6.1, we consider $L = 2$ hidden layers with graphon limits $\mathcal{W}^{(1)}, \mathcal{W}^{(2)}$, and $\mathcal{W}^{(3)}$ of layerwise pruning masks, where $u_0, u_1, u_2, u_3 \in [0, 1]$ index input coordinates, hidden neurons, and output positions. Assume the activation $\sigma$ is twice differentiable, and then the absolute value Graphon-NTK Gram matrix satisfies that

$$|K_{\mathcal{W}}(x, x')| \lesssim \left( \sum_{\ell=0}^{1} C_l \left| \int_0^1 \Sigma^{(\ell)}(u_l, x, x')du_l \right| \right) \cdot$$
$$\left( \int_0^1 |\Sigma^{(0)}(u_0, x, x')|\, P_{\mathcal{W}}(u_0)\, du_0 \right) \quad (10)$$

where $C_l$'s are constants, $\Sigma^l(u_l, x, x')$ is activation covariance, $P_{\mathcal{W}}(u_0) = \int_{[0,1]^3} \mathcal{W}^{(1)}(u_0, u_1)\mathcal{W}^{(2)}(u_1, u_2)$ $\mathcal{W}^{(3)}(u_2, u_3)du_3 du_2 du_1$ is the path density from input $u_0$ to output determined by layer graphons, and $\Sigma^0(u_0, x, x')$ is the input data correlation (See Appendix G for more details).

**Interpretation.** The path density $P_W(u_0)$ acts as a coordinate-dependent measure of how much architectural mass connects input coordinate $u_0$ to the output. In a dense or uniformly random-pruned network, the path density is approximately constant across $u_0$. For example, if $\mathcal{W}^{(\ell)} \equiv \rho_\ell$, then $P_{\mathcal{W}}(u_0) \equiv \rho_1\rho_2\rho_3$ up to the corresponding boundary interpretation for scalar output. Thus Random pruning uniformly rescales the contribution of all input coordinates.

In contrast, data-dependent PaI methods can induce heterogeneous graphons. If their graphons allocate larger connectivity mass to coordinates with larger forward saliency and to hidden units with larger backward sensitivity, then $P_{\mathcal{W}}(u_0)$ becomes larger on informative input regions. In

this case, the Graphon-NTK can place more weight on input correlations that are relevant for the target labels. This can improve both: *(i) spectral conditioning:* better eigenvalue spread in $K_{\mathcal{W}}$, avoiding spectral collapse; and *(ii) label alignment:* smaller complexity $y^\top K_{\mathcal{W}}^{-1} y$ when paths align with informative features.

**Caveat.** Overly peaked (concentrated) graphons risk spectral collapse if connectivity becomes too sparse in critical regions. Effective designs should maintain a balance between the core active region and the lower bounded probability structure: high path density on signal coordinates with nonzero background connectivity for stability. This is empirically observed and verified in (Pham et al., 2023).

### 6.4. Empirical results

We validate these theoretical insights by comparing the complexity measure $y^\top K_{\mathcal{W}}^{-1} y$ of data-driven pruning (SNIP Graphon) versus Random Pruning (Constant Graphon). We use wide finite-width MLP proxies ($n = 4096$) trained on binary CIFAR-10. We vary the label noise (ratio of flipped labels) and the network densities $\rho \in \{0.2, 0.4, 0.7, 0.8\}$.

Figure 2 shows the complexity bound $y^\top K_{\mathcal{W}}^{-1} y$ increases monotonically with label noise across densities, confirming that the bound correctly captures task difficulty. The gap between topologies is most pronounced in the sparse regime. At $\rho = 0.2$ and $\rho = 0.4$, the constant-graphon baseline yields much larger values, consistent with a more ill-conditioned $K_W$, whereas the SNIP graphon remains stable. At higher densities ($\rho \geq 0.7$), the two curves become close (and can slightly reverse), suggesting that once the network is sufficiently dense, the bound is less sensitive to the precise pruning topology.

To provide direct evidence linking our graphon path-density theory to practical performance, we evaluate SNIP and Random pruning on ResNet-20 trained on CIFAR-10 across varying sparsities, visualised in the rightmost plot in Figure 2. We trained all these subnetworks for 160 epochs using the SGD optimiser with a learning rate that started at 0.1 and decayed during training. The result shows that Random is competitive with SNIP at medium sparsity, but degrades much more sharply at extreme sparsity. This is qualitatively consistent with the sensitivity result above and also the graphon/path-density picture: homogeneous random pruning uniformly dilutes connectivity, which becomes harmful at high sparsity, while data-dependent pruning can preserve higher path mass on informative coordinates.

**Takeaway.** The graphon limit offers a convenient way to express how the pruning topology enters the generalisation bound through the complexity term. This offers a way to understand the behaviour of PaI methods. In particular, under low density (high sparsity), data-dependent PaI tends to exhibit a lower $y^\top K_{\mathcal{W}}^{-1} y$, consistent with more concen-

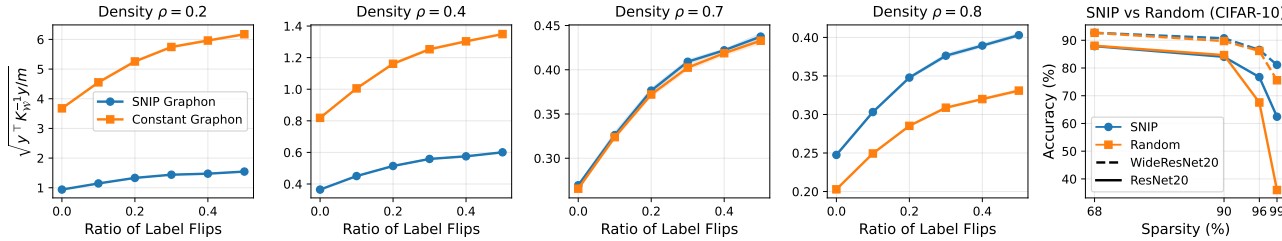

*Figure 2.* Sensitivity of Graphon-NTK Complexity vs. Label Noise and Empirical Pruning Performance. The first four panels plot the theoretical complexity $y^\top K_{\mathcal{W}}^{-1} y$ (y-axis) against the ratio of randomised labels (x-axis) for a specific density $\rho$. The rightmost panel shows the empirical test accuracy against sparsity for SNIP and Random pruning on CIFAR-10 with (Wide)ResNet-20.

trated (heterogeneous) path allocation. This explains why SNIP/GraSP perform better than Random pruning. In contrast, under mild pruning, the constant-graphon (Random pruning) yields a comparable or even smaller complexity term, matching its often competitive performance.

# 7. Conclusion

In this work, we provide the first rigorous derivation of the infinite-width limits for PaI methods. By introducing the Factorised Saliency Model, we prove that discrete masks generated by popular PaI methods converge to limit bipartite graphons. This limit captures the asymptotic sparsity geometry and yields a principled way to compare different pruning rules through their induced graphons. In particular, Random pruning converges to homogeneous graphons, and data-driven methods induce structured heterogeneity. Leveraging this graphon limit perspective, we establish two fundamental guarantees for sparse learning. First, we derive a Universal Approximation Theorem for sparse networks restricted to "active" coordinate subspaces, demonstrating that sparse networks retain the expressivity of dense models on the manifold of relevant features. Second, through the extension of the Graphon-NTK, we link topology to generalisation via the concept of path density. Our analysis confirms that heterogeneous graphons improve kernel alignment by concentrating path density on informative signals, whereas homogeneous ones dilute path density uniformly. Future work could leverage these topological insights to design principled pruning criteria that focus on maximising path density on important features. Additionally, investigating the finite-width corrections to this asymptotic theory would help further bridge the gap to practical implementation.

**Limitations and future works.** While our graphon-limit framework provides a rigorous mathematical foundation for analysing sparse topologies, we acknowledge several theoretical limitations that present exciting future research.

Firstly, our graphon convergence theorem currently assumes a joint infinite-width limit where both the hidden width and input dimension approach infinity. If the input (or output) dimension is kept fixed, as is standard in classical NTK analyses, the mask would no longer converge to a standard

continuous graphon. Instead, the natural limit object becomes a mixed discrete-continuous kernel. Extending our framework to this regime is a crucial next step for aligning the theory closer to standard architectures.

Secondly, to mathematically verify the convergence of the input saliency profile (Assumption 4.2), our single-layer analysis utilises bounded i.i.d. inputs. While this is a strong assumption for structured raw data (e.g., images), it is intentionally designed to be transferable to hidden layers. As outlined in our layer-wise blueprint for deep networks (Appendix D), hidden activations naturally converge to deterministic profiles under standard infinite-width limits, thereby organically satisfying this assumption at deeper layers. Nevertheless, establishing a fully rigorous deep PaI theorem remains an open challenge. Because pruning decisions at early layers perturb the forward activations and backward gradients of subsequent layers, future work must establish tighter, non-asymptotic control over these cross-layer dependencies.

Thirdly, our results all operate in the strict infinite-width limit. In practice, algorithmically pruned networks operate at finite widths where combinatorial variance still impacts the topology. Therefore, a highly impactful future direction is the derivation of finite-width convergence rates. Developing non-asymptotic bounds that quantify how fast empirical masks approach the graphon, and precisely how approximation power and NTK conditioning degrade at finite widths under sparsity, will further bridge the gap between asymptotic theory and practical implementation.

# Impact Statement

This work establishes a rigorous continuous-limit framework for Pruning-at-Initialisation (PaI), characterising the asymptotic limits of sparse network topologies via continuous function kernels. We prove that popular PaI methods converge to deterministic graphons, providing the mathematical tools guarantee the expressivity and generalisation of sparse networks. These insights pave the way for designing principled, highly efficient sparse training algorithms, directly contributing to the effort to reduce the computational and environmental costs of modern deep learning.

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

**Organisation of the Appendix.** This appendix provides detailed mathematical proofs and additional supporting experiments for the results in the main text.

- Appendix A provides the proof of Theorem 4.7 (Graphon Convergence) using the Factorised Saliency Model.

- Appendices B and C provide formal verification of the fact that SNIP and GraSP satisfy the regularity assumptions required for application of Theorem 4.7.

- Appendix D provides details on extension of the Graphon Convergence Theorem to deep neural networks.

- Appendix E provides the proof of Theorem 5.7 (Universal Approximation on Active Subspaces).

- Appendices F and G provide the proofs for the Corollary 6.1 (Generalisation error bound) and the path density decomposition.

## A. Graphon Convergence for Factorised Saliency Model

In this appendix, we prove the graphon convergence in bipartite cut-metric for the factorised saliency model

$$S_{n,ij} = \varphi_{n,i}\,\psi_{n,j}\,|\xi_{n,ij}|, \qquad M_{n,ij} = \mathbf{1}\{S_{n,ij} > \tau_n\}, \qquad i \in [d],\ j \in [n],$$

in the joint limit $d = d(n) \to \infty$ and $n \to \infty$. Let $W_A$ denote the bipartite step kernel associated with a matrix $A \in \mathbb{R}^{d \times n}$, and $\delta_\square^{\mathrm{bip}}$ denote the bipartite cut distance Eqn. (2). We first restate assumptions and main results, then provide detailed explanations and proofs.

### A.1. Assumptions

We will take the joint limit as $d \to \infty$ and $n \to \infty$. We state the assumptions at the necessary level of generality needed for our cut-metric convergence argument. We add remarks to indicate typical sufficient conditions.

**Assumption A.1** (Asymptotic width scaling).
$$\frac{\log(d+n)}{\min\{d,n\}} \to 0.$$

Assumption A.1 ensures that uniform concentration bounds over cut sets dominate logarithm of the cardinality terms arising from discretisation and union bounds. It is satisfied, for instance, when $d$ and $n$ grow at any polynomial rate.

**Assumption A.2** (Deterministic input features). There exists a bounded continuous function $\varphi : [0,1] \to [0,\infty)$ such that, after a relabeling of input coordinates it holds that

$$\max_{1 \le i \le d} \left|\varphi_{n,i} - \varphi(i/d)\right| \xrightarrow{\mathbb{P}} 0.$$

This assumption identifies a continuum limit for the discrete row features and guarantees that the row part of the edge-probability grid is well-approximated by $\varphi(u)$ uniformly in $i$. Continuity of $\varphi$ controls discretization error when passing from $i/d$ to $u$.

**Assumption A.3** (Neuron features empirical CDF convergence). Let $\widehat{F}_{\psi,n}(t) := \frac{1}{n}\sum_{j=1}^{n} \mathbf{1}\{\psi_{n,j} \le t\}$. There exists a deterministic CDF $F_\psi$ supported on $[0,\infty)$ such that

$$\sup_{t \in \mathbb{R}} \left|\widehat{F}_{\psi,n}(t) - F_\psi(t)\right| \xrightarrow{\mathbb{P}} 0.$$

Let $Q_\psi(v) := \inf\{y \ge 0 : F_\psi(y) \ge v\}$ be the generalised quantile. After sorting columns by $\psi_{n,j}$, the kernel becomes a deterministic function of the order statistics $\psi_{n,j}$. This assumption ensures that the sorted column features converge via order statistics to the quantile curve $v \mapsto Q_\psi(v)$, which defines the limiting column coordinate.

**Assumption A.4** (Conditional independence of edge noise). Let

$$\mathcal{G}_n := \sigma\big(\{\varphi_{n,i}\}_{i \le d}, \{\psi_{n,j}\}_{j \le n}, \tau_n\big).$$

Conditioned on $\mathcal{G}_n$, the noises $\{\xi_{n,ij}\}_{i \le d,\, j \le n}$ are independent with common law $P_\xi$, and are independent of $\mathcal{G}_n$. Moreover, the CDF of $|\xi|$ on $[0,\infty)$ is continuous.

In this assumption, independence of $\xi_{n,ij}$ yields conditional independence of mask entries given $(\varphi_n, \psi_n, \tau_n)$, enabling application of the matrix Bernstein concentration for the random mask around its conditional mean. Continuity of the CDF of $|\xi|$ ensures that the link function $z \mapsto \mathbb{P}(z|\xi| > \tau)$ is continuous, which is required to pass to the limit in the mean kernel.

**Assumption A.5** (Deterministic threshold). There exists $\tau \in (0, \infty)$ such that

$$\tau_n \xrightarrow{\mathbb{P}} \tau \quad \text{and} \quad \tau_n \text{ is } \mathcal{G}_n\text{-measurable.}$$

This assumption allows the threshold to control sparsity. Convergence of $\tau_n$ ensures that the edge-probability link function stabilizes asymptotically. Measurability with respect to $\mathcal{G}_n$ guarantees that conditioning on $\mathcal{G}_n$ indeed implies edges independent via Assumption A.4.

### A.2. Proof of Theorem 4.7

**Theorem A.6** (Bipartite graphon convergence, Theorem 4.7). *Let $d \to \infty$ and $n \to \infty$. Consider the model*

$$S_{n,ij} = \varphi_{n,i} \cdot \psi_{n,j} \cdot |\xi_{n,ij}|, \qquad M_{n,ij} = \mathbf{1}\{S_{n,ij} > \tau_n\}, \qquad i \in [d], \ j \in [n], \tag{11}$$

*and let $W_n := W_{M_n}$ denote the step kernel associated with $M_n$. Assume assumptions A.1-A.5. Define the limiting kernel $\mathcal{W} : [0,1]^2 \to [0,1]$ by*

$$\mathcal{W}(u, v) := \mathbb{P}(\varphi(u) \cdot Q_\psi(v) \cdot |\xi| > \tau), \tag{12}$$

*where $Q_\psi := F_\psi^{-1}(v) := \inf\{y : F_\psi(y) \geq v\}$ is the generalized quantile function. Then*

$$\delta_\square^{\mathrm{bip}}(W_n, \mathcal{W}) \xrightarrow{\mathbb{P}} 0. \tag{13}$$

*Remark* A.7. The limiting kernel $\mathcal{W}$ is monotonic in $v$ due to the use of the quantile function $Q_\psi(v)$.

**High-level summary of the proof.** We first sketch the main ideas of the proof. Detailed technical steps will be given later.

Let $\pi_n$ be a sorting permutation of the columns by $\psi_{n,j}$. We write $W_n^\pi$ for the corresponding column-relabeling which does not change $\delta_\square^{\mathrm{bip}}$. Let $\bar{W}_n^\pi$ be the conditional-mean kernel, whose matrix entries are $\bar{M}_{n,ij}^\pi = \mathbb{P}(\varphi_{n,i} \psi_{n,\pi_n(j)} |\xi| > \tau_n \,|\, \varphi_n, \psi_n, \tau_n)$. Then

$$\|W_n^\pi - \mathcal{W}\|_\square \leq \|W_n^\pi - \bar{W}_n^\pi\|_\square + \|\bar{W}_n^\pi - \mathcal{W}\|_\square. \tag{14}$$

(i) *Step 1: Concentration bound* (proved in A.3.3) Conditioned on $(\varphi_n, \psi_n, \tau_n)$, the centered entries $M_{n,ij}^\pi - \bar{M}_{n,ij}^\pi$ are independent, mean-zero, and bounded, so an application of the matrix Bernstein bound in combination with the estimate $\|\cdot\|_\square \lesssim (dn)^{-1/2} \|\cdot\|_{\mathrm{op}}$ yields $\|W_n^\pi - \bar{W}_n^\pi\|_\square \xrightarrow{\mathbb{P}} 0$ under Assumption 4.1.

(ii) *Step2: Mean-kernel limit* (proved in A.3.4) Assumptions 4.2–4.5 imply that $\bar{W}_n^\pi(u, v) \to \mathcal{W}(u, v)$. Using the boundedness of $(0 \leq \bar{W}_n^\pi \leq 1)$, it follows that $\|\bar{W}_n^\pi - \mathcal{W}\|_1 \xrightarrow{\mathbb{P}} 0$, hence $\|\bar{W}_n^\pi - \mathcal{W}\|_\square \xrightarrow{\mathbb{P}} 0$. Combining the two terms and using the invariance under relabelling of the cut metric gives the claim.

### A.3. Technical Details for the Proof of Theorem 4.7

A.3.1. EQUIVALENCE OF CUT NORMS

**Lemma A.8** (Step kernel cut norm equals matrix cut norm). *For $A \in \mathbb{R}^{d \times n}$, define the matrix cut norm*

$$\|A\|_{\square,\mathrm{mat}} := \frac{1}{dn} \max_{S \subseteq [d], T \subseteq [n]} \left| \sum_{i \in S} \sum_{j \in T} A_{ij} \right|. \tag{15}$$

*Then $\|W_A\|_\square = \|A\|_{\square,\mathrm{mat}}$.*

*Proof.* Any choice of subsets $S \subseteq [d]$ and $T \subseteq [n]$ corresponds to measurable sets $S^* = \bigcup_{i \in S} I_i$ and $T^* = \bigcup_{j \in T} J_j$. That yields $\|W_A\|_\square \geq \|A\|_{\square,\mathrm{mat}}$.

Conversely, let $S, T \subset [0, 1]$ be measurable sets. Let $s_i := d\,\mu(S \cap I_i) \in [0, 1]$ and $t_j := n\,\mu(T \cap J_j) \in [0, 1]$, where $\mu$ is the Lebesgue measure. Since $W_A$ is constant on each set $I_i \times J_j$

$$\int_{S \times T} W_A = \frac{1}{dn} \sum_{i=1}^{d} \sum_{j=1}^{n} A_{ij}\, s_i\, t_j. \tag{16}$$

For fixed $t = (t_j)$, the expression is linear in each $s_i \in [0, 1]$, so it is maximized at an extreme point $s_i \in \{0, 1\}$. The maximum over $[0, 1]^{d+n}$ of a multilinear form is achieved at a vertex. Similarly for $t_j$. Hence the supremum over measurable $S, T$ equals the maximum over $S_0 \subset [d]$, $T_0 \subset [n]$, i.e. $\|W_A\|_\square \le \|A\|_{\square, \mathrm{mat}}$. $\square$

$\square$

By Lemma A.8, we can work directly with matrix cut norms.

### A.3.2. COLUMN SORTING AND INVARIANCE

Let $\pi_n$ be a permutation such that $\psi_{n, \pi_n(1)} \le \psi_{n, \pi_n(2)} \le \cdots \le \psi_{n, \pi_n(n)}$. We define the column-permuted mask matrix and its step kernel as

$$M^\pi_{n, ij} := M_{n, i\pi_n(j)}, \quad W^\pi_n := W_{M^\pi_n}. \tag{17}$$

**Lemma A.9** (Invariance under row and column permutation)**.** *We have*

$$\delta^{\mathrm{bip}}_\square(W_n, \mathcal{W}) = \delta^{\mathrm{bip}}_\square(W^{\sigma, \pi}_n, \mathcal{W}) \le \|W^{\sigma, \pi}_n - \mathcal{W}\|_\square. \tag{18}$$

*Proof.* A row and column permutations correspond to a measure-preserving relabelling of the $u$- and $v$-axes that maps each interval $I_i$ to $I_{\sigma_n(i)}$ and $J_j$ to $J_{\pi_n(j)}$, respectively. Since the bipartite cut distance permits independent relabelings of rows and columns, it is invariant under this transformation. The final inequality follows by taking the trivial identity transformation in the infimum defining $\delta^{\mathrm{bip}}_\square$. $\square$

Lemma A.9 reduces the problem to showing $\|W^\pi_n - \mathcal{W}\|_\square \to 0$. To control this distance, we introduce an intermediate proxy, which is the conditional=mean kernel, denoted by $\bar{W}^\pi_n$. From the assumption A.4, conditioned on $\mathcal{G}_n$, the entries $M^\pi_{n, ij}$ are conditionally independent Bernoulli random variables with conditional means

$$\bar{M}^\pi_{n, ij} := \mathbb{E}[M^\pi_{n, ij} \mid \mathcal{G}_n] = \mathbb{P}\big(\varphi_{n, i} \cdot \psi_{n, \pi_n(j)} \cdot |\xi| > \tau_n\big). \tag{19}$$

Let $\bar{W}^\pi_n := W_{\bar{M}^\pi_n}$ be the step kernel induced by $\bar{M}^\pi_n$. By the triangle inequality, we decompose the error into a stochastic concentration term and a deterministic approximation term:

$$\|W^\pi_n - \mathcal{W}\|_\square \le \|W^\pi_n - \bar{W}^\pi_n\|_\square + \|\bar{W}^\pi_n - \mathcal{W}\|_\square \tag{20}$$

We analyse these two terms separately in Appendices A.3.3, A.3.4, respectively.

### A.3.3. CONCENTRATION AROUND CONDITIONAL MEAN

The term $\|W^\pi_n - \bar{W}^\pi_n\|_\square$ captures the deviation of the binary edges from their conditional probabilities due to the noise $\xi$. We define the difference matrix

$$X_n := M^\pi_n - \bar{M}^\pi_n \in \mathbb{R}^{d \times n}. \tag{21}$$

Conditioned on $\mathcal{G}_n$, we have

- The entries $\{X_{n, ij}\}_{i \le d, j \le n}$ are independent, since they depend on i.i.d. $\{\xi_{n, ij}\}$, and $\pi_n$ is a deterministic function of $\psi_n$,

- $\mathbb{E}[X_{n, ij} \mid \mathcal{G}_n] = 0$,

- $|X_{n, ij}| \le 1$.

Our goal is to show that $\|W_n^\pi - \bar{W}_n^\pi\|_\square = \|X_n\|_{\square,\mathrm{mat}} \xrightarrow{\mathbb{P}} 0$.

**Lemma A.10** (Cut norm controlled by operator norm). *For any matrix $B \in \mathbb{R}^{d \times n}$,*

$$\|B\|_{\square,\mathrm{mat}} \leq \frac{1}{\sqrt{dn}}\|B\|_{\mathrm{op}}. \tag{22}$$

*Proof.* By definition,

$$\|B\|_{\square,\mathrm{mat}} = \frac{1}{dn} \max_{S \subseteq [d],\, T \subseteq [n]} \left| \sum_{i \in S} \sum_{j \in T} B_{ij} \right|. \tag{23}$$

For any $S, T$, let $s \in \{0,1\}^d$ and $t \in \{0,1\}^n$ be their indicator vectors. Then

$$\sum_{i \in S} \sum_{j \in T} B_{ij} = s^\top B t. \tag{24}$$

By the Cauchy-Schwarz inequality and the definition of operator norm,

$$|s^\top B t| \leq \|B\|_{\mathrm{op}}\|s\|_2\|t\|_2. \tag{25}$$

Since $\|s\|_2 = \sqrt{|S|} \leq \sqrt{d}$ and $\|t\|_2 = \sqrt{|T|} \leq \sqrt{n}$,

$$|s^\top B t| \leq \|B\|_{\mathrm{op}}\sqrt{dn}. \tag{26}$$

Dividing by $dn$ and maximising over $S, T$ yields the result. $\square$ $\qquad\qquad\qquad\qquad\qquad\qquad\qquad\square$

By Lemma A.10 and Lemma A.8, it suffices to prove $\|X_n\|_{\mathrm{op}} = o(\sqrt{dn})$.

**Lemma A.11** (Matrix Bernstein implies cut-norm concentration). *Under the assumption A.1, we have*

$$\|W_n^\pi - \bar{W}_n^\pi\|_\square \xrightarrow{\mathbb{P}} 0. \tag{27}$$

*Proof.* By Lemma A.8, we have $\|W_n^\pi - \bar{W}_n^\pi\|_\square = \|X_n\|_{\square,\mathrm{mat}}$. By Lemma A.10, we have $\|X_n\|_{\square,\mathrm{mat}} \leq \frac{1}{\sqrt{dn}}\|X_n\|_{\mathrm{op}}$. We apply the rectangular matrix Bernstein inequality (Tropp et al., 2015) to the sum of independent mean-zero random matrices representing the entries of $X_n \in \mathbb{R}^{d \times n}$, conditioned on $\psi_n$: for all $t \geq 0$,

$$\mathbb{P}(\|X\|_{\mathrm{op}} \geq t \mid \mathcal{G}_n) = \mathbb{P}\left( \left\| \sum_{i,j} A_{ij} \right\|_{\mathrm{op}} \geq t \;\Bigg|\; \mathcal{G}_n \right) \leq (d+n)\exp\left( -\frac{t^2}{2(V + Lt/3)} \right). \tag{28}$$

where $A_{ij}$ is a zero matrix except the entry $(i,j)$ where it's value is $X_{ij}$. Therefore, each matrix $A_{ij}$ is zero mean and $|A_{ij}| \leq 1$. The bounding term is $L = \|A_{ij}\|_{\mathrm{op}} = 1$. In addition, $V = \max\{\|\sum_{ij} \mathbb{E}[A_{ij}A_{ij}^\top | \mathcal{G}_n]\|_{\mathrm{op}}, \|\sum_{ij} \mathbb{E}[A_{ij}^\top A_{ij} | \mathcal{G}_n]\|_{\mathrm{op}}\}$ is the norm of the matrix variance of the sum. Here, each entry of $X_n$ is a centered Bernoulli, so $\mathbb{E}[X_{ij}^2 | \mathcal{G}_n] \leq 1/4$. Thus, $V \leq \frac{1}{4}\max(d, n)$, and we obtain the unconditional bound

$$\mathbb{P}(\|X_n\|_{\mathrm{op}} \geq t) \leq (d+n)\exp\left( -\frac{t^2}{2\left(\frac{1}{4}\max(d,n) + t/3\right)} \right). \tag{29}$$

We now choose $t := C\left(\sqrt{d} + \sqrt{n}\right)\sqrt{\log(d+n)}$ with $C > 0$ sufficiently large. Note that $t^2$ is of order $(d+n)\log(d+n)$, which dominates $\max(d,n)$ for large $n$. More precisely, for large $n$,

$$\frac{t^2}{2\left(\frac{1}{4}\max(d,n) + t/3\right)} \geq c\log(d+n) \tag{30}$$

for some constant $c > 0$ that can be made arbitrarily large by increasing $C$. Choosing $C$ so that $c > 2$, we obtain $\mathbb{P}(\|X_n\|_{\mathrm{op}} \geq t) \to 0$.

Therefore, by Lemma A.10,

$$\|X_n\|_{\square,\text{mat}} \leq \frac{1}{\sqrt{dn}}\|X_n\|_{\text{op}} \leq \frac{t}{\sqrt{dn}} = C\left(\sqrt{\frac{\log(d+n)}{n}} + \sqrt{\frac{\log(d+n)}{d}}\right) \tag{31}$$

which converges to 0 by Assumption A.1. By Lemma A.8, we have $\|W_n^\pi - \bar{W}_n^\pi\|_\square = \|X_n\|_{\square,\text{mat}} \xrightarrow{\mathbb{P}} 0$. $\qquad\square$

### A.3.4. MEAN KERNEL CONVERGENCE

The term $\|\bar{W}_n^\pi - \mathcal{W}\|_\square$ measures the discretisation error between the finite grid of probabilities and the continuous limit graphon. In this step, we will bound the cut norm by the stronger $L^1$ norm: $\|\bar{W}_n^\pi - \mathcal{W}\|_\square \leq \|\bar{W}_n^\pi - \mathcal{W}\|_1$.

Define the functions

$$g(\tau, z) := \mathbb{P}(z|\xi| > \tau), \quad z \geq 0, \tau > 0, \tag{32}$$

with $g(\tau, 0) = 0$. Then

$$\bar{M}_{n,ij}^\pi = g\big(\tau_n, \varphi_{n,i}\,\psi_{n,\pi_n(j)}\big), \qquad \mathcal{W}(u,v) = g(\tau, \varphi(u)\,Q_\psi(v)). \tag{33}$$

Here, we call this functions $g()$ a link function that capture the fact that edge probability depends on the product of forward and backward factors.

**Lemma A.12** (Order statistics from empirical CDF convergence). *Assume Assumption A.3. Let $v \in (0,1)$ be a continuity point of $Q_\psi$ and $j_n := \lceil nv \rceil$. Then*

$$\psi_{n,(j_n)} \xrightarrow{\mathbb{P}} Q_\psi(v). \tag{34}$$

*Proof.* Let $q := Q_\psi(v)$ and fix $\eta > 0$. Continuity of $Q_\psi$ at $v$ implies $F_\psi(q-\eta) < v < F_\psi(q+\eta)$. Set

$$\Delta_\eta := \min\{v - F_\psi(q-\eta),\ F_\psi(q+\eta) - v\} > 0. \tag{35}$$

On the event $\{\sup_t |\widehat{F}_{\psi,n}(t) - F_\psi(t)| \leq \Delta_\eta\}$, we have $\widehat{F}_{\psi,n}(q-\eta) < v < \widehat{F}_{\psi,n}(q+\eta)$. By the generalized inverse identity $\widehat{F}_{\psi,n}^{-1}(v) = \psi_{n,\lceil nv \rceil}$, it holds that $q - \eta \leq \psi_{n,(j_n)} \leq q + \eta$. Therefore

$$\mathbb{P}\big(|\psi_{n,(j_n)} - q| > \eta\big) \leq \mathbb{P}\left(\sup_t |\widehat{F}_{\psi,n}(t) - F_\psi(t)| > \Delta_\eta\right) \to 0, \tag{36}$$

which shows $\psi_{n,j_n} \xrightarrow{\mathbb{P}} q$. $\qquad\square$

**Lemma A.13** ($L^1$ convergence of the mean kernel). *It holds that*

$$\|\bar{W}_n^\pi - \mathcal{W}\|_1 \xrightarrow{\mathbb{P}} 0, \quad \text{and} \quad \|\bar{W}_n^\pi - \mathcal{W}\|_\square \xrightarrow{\mathbb{P}} 0. \tag{37}$$

*Proof.* Let $D_n(u,v) := |\bar{W}_n^\pi(u,v) - \mathcal{W}(u,v)| \in [0,1]$. Fix $(u,v)$ such that $v$ is a continuity point of $Q_\psi$. Let $i_n := \lceil du \rceil$ and $j_n := \lceil nv \rceil$. By Assumption A.2 and continuity of $\varphi$, we have $\varphi_{n,i_n} \xrightarrow{\mathbb{P}} \varphi(u)$. By Lemma A.12, $\psi_{n,(j_n)} \xrightarrow{\mathbb{P}} Q_\psi(v)$. By Assumption A.5, $\tau_n \xrightarrow{\mathbb{P}} \tau$. Since $g$ is continuous and the limit is deterministic, the continuous mapping theorem yields

$$\bar{W}_n^\pi(u,v) = g\big(\tau_n, \varphi_{n,i_n}\psi_{n,(j_n)}\big) \xrightarrow{\mathbb{P}} g\big(\tau, \varphi(u)Q_\psi(v)\big) = \mathcal{W}(u,v), \tag{38}$$

so that $D_n(u,v) \xrightarrow{\mathbb{P}} 0$ for all $u$ and all continuity points $v$ of $Q_\psi$. As $Q_\psi$ has at most countably many discontinuities, this holds for Lebesgue-a.e. $(u,v)$. Because $0 \leq D_n(u,v) \leq 1$, for any $\varepsilon > 0$, $\mathbb{E}[D_n(u,v)] \leq \varepsilon + \mathbb{P}(D_n(u,v) > \varepsilon)$, this boundedness implies $\mathbb{E}[D_n(u,v)] \to 0$ for Lebesgue-a.e. $(u,v)$ when $D_n(u,v) \xrightarrow{\mathbb{P}} 0$. By dominated convergence theorem,

$$\mathbb{E}\|\bar{W}_n^\pi - \mathcal{W}\|_1 = \int_{[0,1]^2} \mathbb{E}[D_n(u,v)]\,du\,dv \to 0. \tag{39}$$

The Markov inequality yields $\|\bar{W}_n^\pi - \mathcal{W}\|_1 \xrightarrow{\mathbb{P}} 0$. Finally, $\|\cdot\|_\square \leq \|\cdot\|_1$ gives the cut-norm claim. $\qquad\square$

A.3.5. CONCLUSION

Combining the above results and the triangle inequality, we have

$$\|W_n^\pi - \mathcal{W}\|_\square \leq \|W_n^\pi - \bar{W}_n^\pi\|_\square + \|\bar{W}_n^\pi - \mathcal{W}\|_\square \xrightarrow{\mathbb{P}} 0 \tag{40}$$

By Lemma A.9, it holds that

$$\delta_\square^{\text{bip}}(W_n, \mathcal{W}) = \delta_\square^{\text{bip}}(W_n^\pi, \mathcal{W}) \leq \|W_n^\pi - \mathcal{W}\|_\square \xrightarrow{\mathbb{P}} 0, \tag{41}$$

which completes the proof of Theorem 4.7.

## B. Realisation of SNIP in the Factorised Saliency Model

In this appendix, we show that SNIP fits into the Factorised Saliency Model $S_{ij} \propto \phi_i \psi_j |\xi_{ij}|$ for a single hidden layer network.

### B.1. Model and SNIP score at initialisation

**Bounded input setting.** Let the input be a single random sample

$$x = (x_1, \ldots, x_d) \in \mathbb{R}^d, \qquad x_i \overset{\text{i.i.d.}}{\sim} X, \qquad |X| \leq B \text{ a.s.} \tag{42}$$

for some bound $B > 0$. Let $F_{|X|}$ denote the CDF of $|X|$ on $[0, B]$. We assume $F_{|X|}$ is continuous on $[0, B]$.

**Initialisation.** Let $(\theta_j^{(2)})_{j=1}^n$ and $(\theta_{ij}^{(1)})_{1 \leq i \leq d, \, 1 \leq j \leq n}$ be independent families with $\theta_j^{(2)} \overset{\text{i.i.d.}}{\sim} \mathcal{N}(0,1)$, $\theta_{ij}^{(1)} \overset{\text{i.i.d.}}{\sim} \mathcal{N}(0,1)$.

**One hidden layer network.** Consider the random network

$$f_n(x) = \frac{1}{\sqrt{n}} \sum_{j=1}^n \theta_j^{(2)} \sigma(h_j^{(1)}), \qquad h_j^{(1)} := \frac{1}{\sqrt{d}} \sum_{i=1}^d \theta_{ij}^{(1)} x_i, \tag{43}$$

and squared loss on a fixed label $y \in \mathbb{R}$,

$$L = \frac{1}{2} \big( f_n(x) - y \big)^2, \qquad \delta := f_n(x) - y. \tag{44}$$

**SNIP pruning score.** We introduce a binary mask $m_{ij}^{(1)} \in \{0, 1\}$ on first-layer weights via $\theta_{ij}^{(1)} \mapsto m_{ij}^{(1)} \theta_{ij}^{(1)}$ and evaluate SNIP at $m \equiv 1$. By the chain rule

$$\frac{\partial L}{\partial m_{ij}^{(1)}} = \delta \cdot \frac{\partial f_n(x)}{\partial m_{ij}^{(1)}} = \delta \cdot \frac{1}{\sqrt{n}} \theta_j^{(2)} \sigma'(h_j^{(1)}) \cdot \frac{1}{\sqrt{d}} \theta_{ij}^{(1)} x_i. \tag{45}$$

Hence, the SNIP pruning score magnitude factorises as

$$S_{ij}^{(n)} := \Big| \frac{\partial L}{\partial m_{ij}^{(1)}} \Big| = \frac{|\delta|}{\sqrt{nd}} \underbrace{|x_i|}_{\varphi_{n,i}} \underbrace{\big( |\theta_j^{(2)}| \, |\sigma'(h_j^{(1)})| \big)}_{\psi_{n,j}} \underbrace{|\theta_{ij}^{(1)}|}_{|\xi_{n,ij}|}. \tag{46}$$

Since top-$\rho$ ranking is invariant to multiplying all scores by the same positive constant, we may equivalently work with the scale-free scores

$$S_{ij}^{e(n)} := \frac{\sqrt{nd}}{|\delta|} S_{ij}^{(n)} = \varphi_{n,i} \, \psi_{n,j} \, |\xi_{n,ij}| = |x_i| \, |\theta_j^{(2)}| \, |\sigma'(h_j^{(1)})| \, |\theta_{ij}^{(1)}|. \tag{47}$$

All masks below are defined using $S_{ij}^{e(n)}$.

### B.2. Verification of Assumptions A1–A5 for the model

We verify the regularity assumptions in Section 4.2 for the SNIP factorisation $S_{ij}^{e(n)} = \varphi_{n,i} \psi_{n,j} |\xi_{n,ij}|$ with

$$\varphi_{n,i} = |x_i|, \qquad \psi_{n,j} = |\theta_j^{(2)}| \, |\sigma'(h_j^{(1)})|, \qquad \xi_{n,ij} = \theta_{ij}^{(1)}. \tag{48}$$

**(A.4.1) Growth rate.** Assumption A1 is a regime condition on $(d, n)$ which we assume in advance.

**(A.4.2) Deterministic input features via measure-preserving relabelling.** Since $(|x_i|)_{i=1}^d$ are i.i.d. supported on $[0, B]$, we may relabel rows by sorting $\{|x_i|\}_{i=1}^d$ increasingly. Let $|x|_{(1)} \leq \cdots \leq |x|_{(d)}$ be the order statistics and define $\varphi_{n,i} := |x|_{(i)}$. We define the deterministic limit profile $\varphi(u) := F_{|X|}^{-1}(u), u \in [0, 1]$, which is bounded by $B$ and continuous on $[0, 1]$ under continuity of $F_{|X|}$. Then standard uniform quantile consistency yields

$$\max_{1 \leq i \leq d} \Big| \varphi_{n,i} - \varphi(i/d) \Big| \overset{\mathbb{P}}{\to} 0 \tag{49}$$

This is exactly Assumption 4.2.

**(A.4.3) Neuron features empirical CDF convergence** Fix $x$ and consider $(h_j^{(1)})_{j=1}^n$ with $h_j^{(1)} := \frac{1}{\sqrt{d}} \sum_{i=1}^d \theta_{ij}^{(1)} x_i$. Because the columns $(\theta_{\cdot j}^{(1)})_{j=1}^n$ are i.i.d. and independent of $x$, we have:

- Conditional on $x$, the random variables $(h_j^{(1)})_{j=1}^n$ are i.i.d. across $j$ as $d \to \infty$. Moreover, since $(\theta_{ij}^{(1)})_i$ are Gaussian,

$$h_j^{(1)} \mid x \sim \mathcal{N}(0, s_d^2(x)), \qquad s_d^2(x) := \frac{\|x\|_2^2}{d}. \tag{50}$$

- $(\theta_j^{(2)})_{j=1}^n$ are i.i.d. and independent of $(h_j^{(1)})_{j=1}^n$. Therefore $\psi_{n,j} := |\theta_j^{(2)}| |\sigma'(h_j^{(1)})|$ are i.i.d. across $j$ conditional on $x$.

To match Assumption 4.3 as stated (a non-random CDF $F_\psi$), we identify a deterministic limit law by showing $s_d^2(x) \to \nu^2$ with $\nu^2 := \mathbb{E}[X^2]$. Since $|X| \le B$ a.s., Hoeffding's inequality implies for all $t > 0$,

$$\mathbb{P}\left( \left| \frac{1}{d} \sum_{i=1}^d x_i^2 - \nu^2 \right| > t \right) \le 2 \exp\left( -\frac{2dt^2}{B^4} \right), \tag{51}$$

hence $s_d^2(x) = \|x\|_2^2 / d \to \nu^2$ in probability. Combining with Eqn. (50), the conditional law of $h_j^{(1)} \mid x$ converges to $\mathcal{N}(0, \nu^2)$, and thus $\psi_{n,j}$ converges in distribution to the deterministic limit

$$\psi \overset{d}{=} |\Theta^{(2)}| |\sigma'(\nu G)|, \qquad \Theta^{(2)} \sim \mathcal{N}(0,1), \qquad G \sim \mathcal{N}(0,1). \tag{52}$$

Let $F_\psi$ denote the CDF of $\psi$ on $[0, \infty)$ and quantile $Q_\psi$. Then the empirical CDF of $\{\psi_{n,j}\}_{j=1}^n$, $\hat{F}_{\psi,n}(t) := \frac{1}{n} \sum_{j=1}^n \mathbf{1}\{\psi_{n,j} \le t\}$, satisfies $\hat{F}_{\psi,n}(t) \overset{\mathbb{P}}{\to} F_\psi(t)$, at continuity points $t$ of $F_\psi$, Lebesgue-a.e. $v \in (0,1)$.

**(A4.4) Edge noise and asymptotic decoupling.** Let

$$\xi_{n,ij} := \theta_{ij}^{(1)} \sim N(0,1),$$

so that $|\xi|$ has a continuous distribution on $[0, \infty)$. The only minor subtlety is that

$$\psi_{n,j} = |\theta_j^{(2)}| |\sigma'(h_j^{(1)})|, \qquad h_j^{(1)} = \frac{1}{\sqrt{d}} \sum_{r=1}^d \theta_{rj}^{(1)} x_r,$$

is not exactly independent of $\theta_{ij}^{(1)}$ at finite $d$, since $\theta_{ij}^{(1)}$ appears inside $h_j^{(1)}$. We remove this dependence by a leave-one-out replacement.

For each pair $(i, j)$, define

$$h_j^{(1),-i} := h_j^{(1)} - \frac{1}{\sqrt{d}} \theta_{ij}^{(1)} x_i = \frac{1}{\sqrt{d}} \sum_{r \ne i} \theta_{rj}^{(1)} x_r,$$

and

$$\psi_{n,j}^{-i} := |\theta_j^{(2)}| |\sigma'(h_j^{(1),-i})|.$$

Conditional on $x$, the variable $\psi_{n,j}^{-i}$ is independent of $\theta_{ij}^{(1)}$. Hence the proxy score

$$S_{ij}^{(n),-i} = |x_i| \, \psi_{n,j}^{-i} \, |\theta_{ij}^{(1)}|$$

has the desired factorised form with independent edge noise.

It remains to check that the leave-one-out replacement is asymptotically negligible. If $\sigma'$ is bounded and Lipschitz, then

$$|\psi_{n,j} - \psi_{n,j}^{-i}| \le |\theta_j^{(2)}| \operatorname{Lip}(\sigma') \frac{|x_i| |\theta_{ij}^{(1)}|}{\sqrt{d}}.$$

Since the Gaussian weights have finite moments and $|x_i| \leq B$,

$$\mathbb{E}\left[\frac{1}{dn}\sum_{i=1}^{d}\sum_{j=1}^{n}|\psi_{n,j} - \psi_{n,j}^{-i}| \;\middle|\; x\right] \leq \frac{C}{\sqrt{d}}.$$

Therefore, by Markov's inequality,

$$\frac{1}{dn}\sum_{i=1}^{d}\sum_{j=1}^{n}|\psi_{n,j} - \psi_{n,j}^{-i}| \xrightarrow{\mathbb{P}} 0.$$

Thus the true SNIP scores are asymptotically equivalent to the leave-one-out scores in the sense needed for thresholded masks. Indeed, multiplying the above difference by the bounded row factor $|x_i|$ and the finite-moment edge factor $|\theta_{ij}^{(1)}|$ gives an average $o(1)$ perturbation of the score array. By the standard threshold-stability argument at continuity points of the limiting score distribution, the masks generated by $S_{ij}^{(n)}$ and $S_{ij}^{(n),-i}$ differ on a vanishing fraction of edges. Consequently, the graphon limit may be proved for the decoupled leave-one-out proxy and then transferred back to the original SNIP mask.

**(A4.5) Threshold stability.** The main theorem assumes the pruning thresholds $\tau_n$ converge to a deterministic limit $\tau \in (0, \infty)$. In practical PaI methods such as SNIP, sparsity is enforced by top-$\rho$ selection, so the threshold is the empirical $(1 - \rho)$-quantile $\hat{\tau}_n$ of the scores. Let $\hat{F}_S$ denote the empirical CDF of the score array $\{S_{ij}\}_{i\leq d, \, j\leq n}$, and let $F_S$ be the deterministic limiting CDF. As $(d, n) \to \infty$, $\hat{F}_S$ concentrates uniformly around $F_S$, i.e., $\sup_{t\in\mathbb{R}}\left|\hat{F}_S(t) - F_S(t)\right| \xrightarrow{\mathbb{P}} 0$, which in our factorised setting follows from the conditional i.i.d. structure of $\{S_{ij}\}$ given the row and column features (A.4) together with a standard application of the Law of Large Number.

Define the theoretical threshold $\tau := F_S^{-1}(1 - \rho)$ and assume $F_S$ is continuous at $\tau$. Then $\hat{\tau}_n \to \tau$ in probability, and replacing $\hat{\tau}_n$ by the constant $\tau$ changes only a vanishing fraction of mask entries:

$$\frac{1}{dn}\sum_{i\leq d}\sum_{j\leq n}\left|\mathbf{1}\{S_{ij} > \hat{\tau}_n\} - \mathbf{1}\{S_{ij} > \tau\}\right| \xrightarrow{\mathbb{P}} 0. \tag{53}$$

Consequently, any graphon limit established for the deterministic-threshold mask $\mathbf{1}\{S_{ij} > \tau\}$ transfers directly to the practical top-$\rho$ mask $\mathbf{1}\{S_{ij} > \hat{\tau}_n\}$.

### B.3. Empirical Verification

We empirically illustrate Theorem 4.7 by comparing finite-width SNIP masks to the predicted limit graphon. For visualisation, we generate square masks $M_n \in \{0, 1\}^{n \times n}$ at two fixed densities $\rho = \{0.1, 0.2\}$ for $n \in \{200, 500, 1000, 2000, 4000\}$. Following Pham et al. (2025), we sort rows/columns by their latent factors ($\varphi_i$ for rows and $\psi_j$ for columns) and average the sorted masks over 100 random seeds to obtain an empirical edge-probability matrix.

Besides, we compute the limit theoretical graphon, $\mathcal{W}(u, v)$, as a continuous function on the unit square $[0, 1]^2$. To make the limit object computable, we consider i.i.d. standard normal inputs and weights initialisation. Since inputs $x \sim \mathcal{N}(0, 1)$, the magnitude $|x|$ follows a standard Half-Normal (HN) distribution. We define $\varphi(u) = F_{\text{HN}}^{-1}(u) = \sqrt{2}\,\text{erf}^{-1}(u)$. The column factor is defined as $\psi = |\theta^{(2)}||\sigma'(h)|$, where $\theta^{(2)} \sim \mathcal{N}(0, 1)$, and different activation functions $\sigma$ (ReLU, Tanh, Sigmoid). We determine the quantile function $Q_\psi(v)$ for this product distribution via Monte Carlo simulation ($N = 10^6$ samples). The edge noise $|\xi_{ij}|$ is Half-Normal distributed with scale $\sigma_{\theta^{(1)}} = 1$ where $\theta^{(1)} \sim \mathcal{N}(0, 1)$. Given a global threshold $\tau$ (solved numerically to match density $\rho$), the theoretical edge probability is $\mathcal{W}_{(u,v)} = \mathbb{P}\left(\varphi(u)\psi(v)|\xi| > \tau\right) = 1 - \text{erf}(\frac{\tau}{\varphi(u)Q_\psi(v)\sqrt{2}})$.

Figure 3 and 4 shows that the averaged empirical masks become progressively smoother and rapidly approach the theoretical $\mathcal{W}$ as $n$ increases; by $n \geq 2000$ the empirical and theoretical patterns are visually indistinguishable. This supports the interpretation of PaI masks as finite-sample discretisations of a deterministic graphon limit.

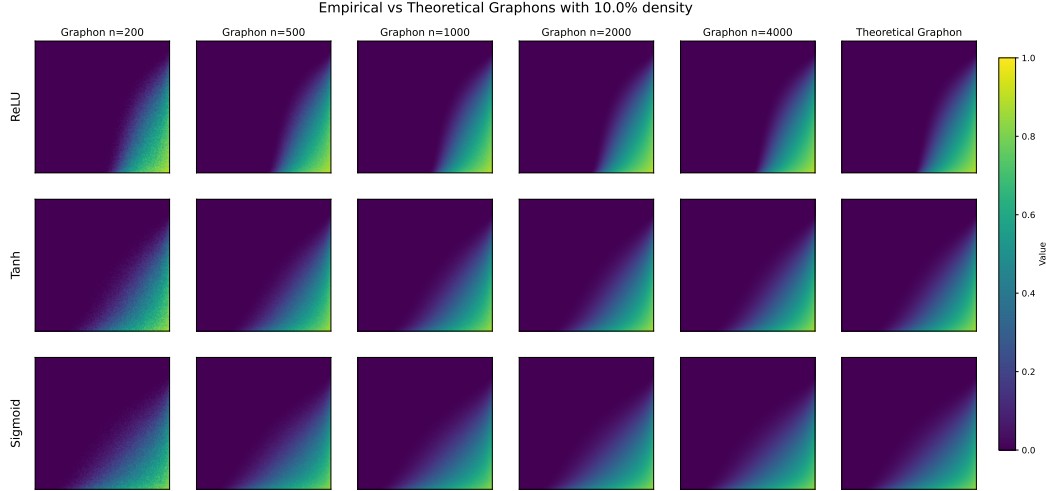

*Figure 3.* **Visual Convergence to the Graphon Limit.** We compare empirical masks with different activation functions at increasing widths ($n = 200, 500, 1000, 2000, 4000$) against the analytically computed Theoretical Graphon. The density is fixed at $\rho = 0.1$.

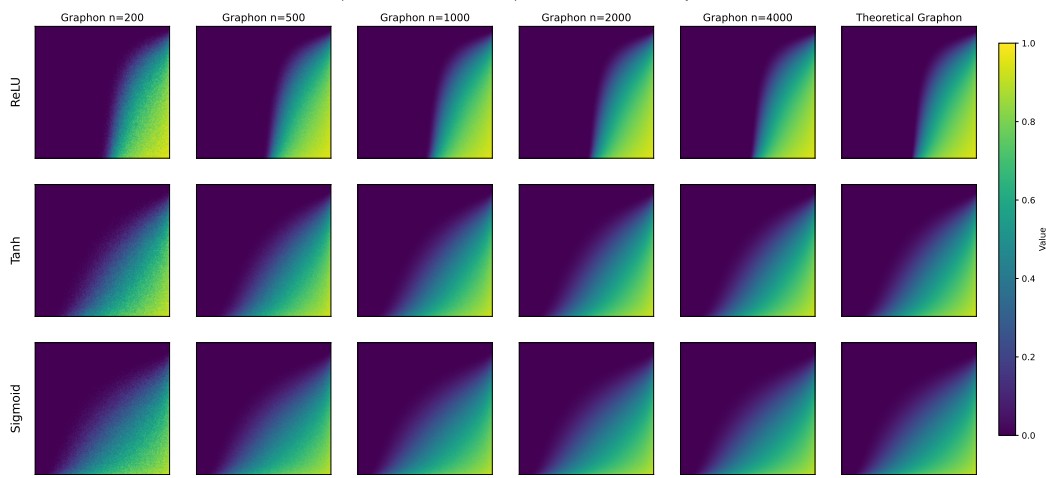

*Figure 4.* **Visual Convergence to the Graphon Limit.** We compare empirical masks with different activation functions at increasing widths ($n = 200, 500, 1000, 2000, 4000$) against the analytically computed Theoretical Graphon. The density is fixed at $\rho = 0.2$.

## C. Realisation of GraSP in the Factorised Saliency Model

In this appendix, we show that GraSP can be realised into the Factorised Saliency Model by an explicit decomposition of the Hessian–gradient product in the same setting as in Appendix B. We treat both (i) the original signed GraSP pruning score and (ii) its magnitude variant, and show that, up to an $(i, j)$-independent scalar, they are asymptotically rank-equivalent to the corresponding signed or magnitude SNIP scores. The discrepancy is either identically zero for piecewise-linear activations, or vanishes uniformly as $n \to \infty$.

Recall that GraSP assigns each parameter a signed score based on a Hessian–gradient product. Let $\theta \in \mathbb{R}^{dn}$ be the vector of first-layer weights $\{\theta_{ij}^{(1)}\}_{1 \leq i \leq d, 1 \leq j \leq n}$, and define the gradient and Hessian of the loss w.r.t. $\theta$ as

$$s(\theta) := \nabla_\theta L(\theta) \in \mathbb{R}^{dn}, \qquad H(\theta) := \nabla_\theta^2 L(\theta) \in \mathbb{R}^{dn \times dn}. \tag{54}$$

GraSP uses the elementwise product $Hs$ to score coordinates. In matrix indexing, define the signed GraSP score as

$$Z_{ij}^{(n)} := -\theta_{ij}^{(1)} (Hs)_{ij}, \qquad 1 \leq i \leq d, \ 1 \leq j \leq n. \tag{55}$$

**Two pruning conventions.** We consider:

1. **(Signed GraSP).** Define the signed retention saliency

$$S_{ij}^{\text{sgn}} := -Z_{ij}^{(n)} = \theta_{ij}^{(1)}(Hs)_{ij}. \tag{56}$$

Then the signed GraSP mask is

$$M_{ij}^{\text{sgn}} := \mathbf{1}\{S_{ij}^{\text{sgn}} > \tau_n^{\text{sgn}}\}. \tag{57}$$

Note that $\tau_n^{\text{sgn}}$ may be negative. This variant retains weights based on the signed influence captured by $Hs$.

2. **(Magnitude GraSP).** Define the magnitude saliency

$$S_{ij}^{\text{mag}} := |Z_{ij}^{(n)}|. \tag{58}$$

Then the magnitude GraSP mask is

$$M_{ij}^{\text{mag}} := \mathbf{1}\{S_{ij}^{\text{mag}} > \tau_n^{\text{mag}}\}, \qquad \tau_n^{\text{mag}} \geq 0. \tag{59}$$

This variant retains weights with large influence regardless of sign and fits the nonnegative Factorised Saliency Model most directly.

*Remark* C.1. If masks are produced by top-$\rho$ selection, then Eqn. (57) and Eqn. (59) correspond to taking $\tau^n$ to be the empirical $(1 - \rho)$-quantile of $\{S_{ij}\}$. All results below apply to either form, via the same threshold-replacement argument as in Appendix B.4.

### C.1. Hessian factorisation and the global scalar $c_n$

Let us write the Jacobian of the scalar output w.r.t. $\theta$ as $J := \nabla_\theta f_n(x) \in \mathbb{R}^{dn}$. By the chain rule, $g = \nabla_\theta L = \delta \nabla_\theta f_n = \delta J$. For squared loss, we have $H = \nabla_\theta^2 L = JJ^\top + \delta \nabla_\theta^2 f_n$. Therefore the Hessian–gradient product decomposes as

$$Hg = (JJ^\top)(\delta J) + \delta(\nabla_\theta^2 f_n)(\delta J) = \underbrace{\|J\|_2^2}_{=:c_n} g + \underbrace{\delta^2(\nabla_\theta^2 f_n)J}_{=:R}. \tag{60}$$

The key point is that $c_n = \|J\|_2^2$ is a single scalar, which is independent of $(i, j)$.

**Explicit formula for $c_n$.** In our model,

$$J_{ij} = \frac{\partial f_n}{\partial \theta_{ij}^{(1)}} = \frac{1}{\sqrt{n}}\theta_j^{(2)}\sigma'(h_j^{(1)}) \cdot \frac{1}{\sqrt{d}}x_i, \tag{61}$$

so that

$$c_n = \sum_{i=1}^{d}\sum_{j=1}^{n} J_{ij}^2 = \left(\frac{\|x\|_2^2}{d}\right) \cdot \left(\frac{1}{n}\sum_{j=1}^{n}(\theta_j^{(2)})^2\sigma'(h_j^{(1)})^2\right). \tag{62}$$

**Explicit formula for the remainder $R$.** A direct differentiation shows that the only nonzero second derivatives couple weights within the same neuron:

$$\frac{\partial^2 f_n}{\partial \theta_{ij}^{(1)} \partial \theta_{i'j'}^{(1)}} = \mathbf{1}\{j = j'\} \cdot \frac{1}{\sqrt{n}}\theta_j^{(2)}\sigma''(h_j^{(1)}) \cdot \frac{1}{d}x_i x_{i'}. \tag{63}$$

Using $J_{i'j} = \frac{1}{\sqrt{n}}\theta_j^{(2)}\sigma'(h_j^{(1)})\frac{1}{\sqrt{d}}x_{i'}$ yields

$$R_{ij} = \delta^2 \sum_{i'=1}^{d}\frac{\partial^2 f_n}{\partial \theta_{ij}^{(1)} \partial \theta_{i'j}^{(1)}} J_{i'j} = \delta^2 \cdot \frac{(\theta_j^{(2)})^2}{n}\sigma''(h_j^{(1)})\sigma'(h_j^{(1)}) \cdot \left(\frac{\|x\|_2^2}{d}\right) \cdot \frac{x_i}{\sqrt{d}}. \tag{64}$$

Combining Eqn. (55) and Eqn. (60) gives the entrywise score decomposition

$$Z_{ij}^{(n)} = -c_n \underbrace{(\theta_{ij}^{(1)}g_{ij})}_{\text{SNIP signed score}} - \underbrace{\theta_{ij}^{(1)}R_{ij}}_{=:E_{ij}}. \tag{65}$$

Recall from Appendix B that the SNIP score magnitude is $|\theta_{ij}^{(1)}g_{ij}| = \frac{|\delta|}{\sqrt{nd}}|x_i|\,|\theta_j^{(2)}|\,|\sigma'(h_j^{(1)})|\,|\theta_{ij}^{(1)}|$.

### C.2. Lower-order remainder bounds

We consider two regimes.

**Case 1: Piecewise-linear activations.** If $\sigma$ is piecewise linear (e.g. ReLU or leaky-ReLU), then $\sigma''(t) = 0$ for all $t \neq 0$. Since each $h_j^{(1)}$ has a continuous distribution at random initialization, $\mathbb{P}(h_j^{(1)} = 0) = 0$, hence $\sigma''(h_j^{(1)}) = 0$ for every $j$. By Eqn. (64),

$$R \equiv 0 \quad \text{a.s.} \qquad \Rightarrow \qquad Hg = c_n g. \tag{66}$$

Thus $Z_{ij}^{(n)} = -c_n(\theta_{ij}^{(1)} g_{ij})$ exactly, i.e. GraSP is an entrywise independent rescaling of SNIP.

**Case 2: Smooth activations.** Assume $\sigma$ is twice differentiable with bounded second derivative $\|\sigma''\|_\infty := \sup_{t \in \mathbb{R}} |\sigma''(t)| < \infty$, and assume the nondegeneracy condition

$$\mu_1 := \mathbb{E}\Big[(\Theta^{(2)})^2 \sigma'(\nu G)^2\Big] > 0, \qquad \Theta^{(2)}, G \overset{\text{i.i.d.}}{\sim} \mathcal{N}(0,1), \qquad \nu^2 := \mathbb{E}[X^2]. \tag{67}$$

Here $\nu^2$ is the same deterministic variance limit used in the SNIP verification; see Appendix B.2.

Define $\bar{c}_n := \frac{1}{n} \sum_{j=1}^n (\theta_j^{(2)})^2 \sigma'(h_j^{(1)})^2$, so $c_n = (\|x\|_2^2/d)\bar{c}_n$ by Eqn. (62). Then $\bar{c}_n \to \mu_1$ in probability by the same argument used for the SNIP column features in Appendix B.2. In particular, for any $\eta \in (0, \mu_1)$,

$$\mathbb{P}(\bar{c}_n \geq \mu_1 - \eta) \to 1. \tag{68}$$

Next, we compare the remainder to the leading term. Using Eqn. (64) and $g_{ij} = \delta J_{ij} = \delta \cdot \frac{1}{\sqrt{n}} \theta_j^{(2)} \sigma'(h_j^{(1)}) \frac{1}{\sqrt{d}} x_i$, we obtain

$$\frac{|R_{ij}|}{|c_n g_{ij}|} = \frac{|\delta|}{\sqrt{n}} \cdot \frac{|\theta_j^{(2)}| |\sigma''(h_j^{(1)})|}{\bar{c}_n}. \tag{69}$$

Thus, on the event $\{\bar{c}_n \geq \mu_1/2\}$ and using $|\sigma''(h_j^{(1)})| \leq \|\sigma''\|_\infty$,

$$\max_{i \leq d,\, j \leq n} \frac{|R_{ij}|}{|c_n g_{ij}|} \leq \frac{2\|\sigma''\|_\infty}{\mu_1} \cdot \frac{|\delta|}{\sqrt{n}} \cdot \max_{j \leq n} |\theta_j^{(2)}|. \tag{70}$$

Finally, $\max_{j \leq n} |\theta_j^{(2)}| = O(\sqrt{\log n})$ by Gaussian maxima, and $\delta = f_n(x) - y = O(1)$ at initialization under the $1/\sqrt{n}$ output scaling and bounded-input regime, as $f_n(x)$ is sub-Gaussian conditioned on $x$. Therefore the right-hand side of Eqn. (70) is $O(\sqrt{\log n/n}) \to 0$. Combining with Eqn. (68), we conclude the uniform vanishing:

$$\max_{i \leq d,\, j \leq n} \frac{|R_{ij}|}{|c_n g_{ij}|} \overset{\mathbb{P}}{\to} 0 \qquad \text{as } n \to \infty. \tag{71}$$

**Consequence: Asymptotic rank-equivalence to SNIP.** By Eqn. (65) and Eqn. (71),

$$Z_{ij}^{(n)} = -c_n(\theta_{ij}^{(1)} g_{ij}) \cdot (1 + o(1)) \quad \text{uniformly over the entries.}$$

Therefore the signed score $S_{ij}^{\text{sgn}} = -Z_{ij}^{(n)}$ is uniformly rank-equivalent to the signed SNIP score $\theta_{ij}^{(1)} s_{ij}$, while the magnitude saliency $S_{ij}^{\text{mag}} = |Z_{ij}^{(n)}|$ is uniformly rank-equivalent to the magnitude SNIP score $|\theta_{ij}^{(1)} s_{ij}|$.

### C.3. Realising GraSP in the factorised saliency model and assumption verification

The leading order GraSP score depends on $(i, j)$ only through the factor $\theta_{ij}^{(1)} g_{ij}$ up to the global scalar $c_n$. In particular, the leading term factorises exactly as in the SNIP appendix:

$$\theta_{ij}^{(1)} g_{ij} = \frac{\delta}{\sqrt{nd}} x_i \left( \theta_j^{(2)} \sigma'(h_j^{(1)}) \right) \theta_{ij}^{(1)}, \tag{72}$$

GraSP Variants Convergence vs Theory (Density=20.0%)

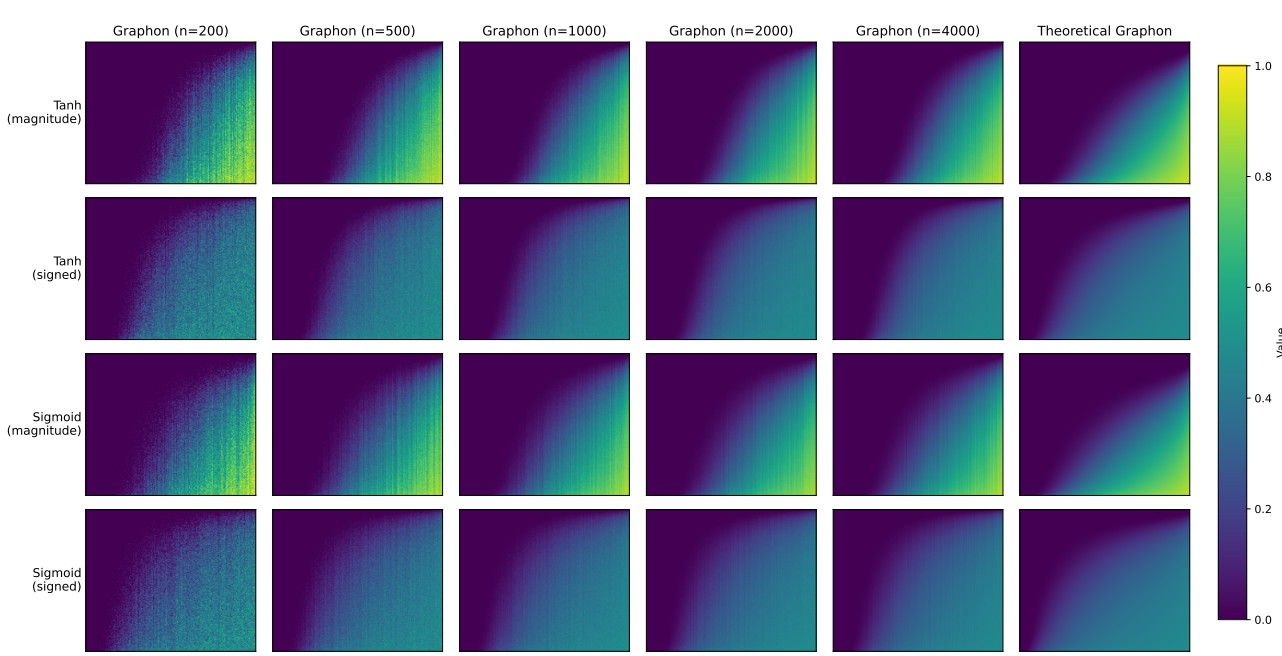

*Figure 5.* **Graphon Convergence of GraSP Variants.** Empirical pruning masks for GraSP at increasing widths ($n \in \{200, \ldots, 4000\}$) with density $\rho = 20\%$. We compare **Magnitude GraSP** (rows 1, 3) and **Signed GraSP** (rows 2, 4) for Tanh and Sigmoid activations.

Thus, the signed GraSP fits a *signed* factorised model

$$S_{n,ij}^{\text{sgn}} = \varphi_{n,i}\psi_{n,j}\xi_{n,ij}, \qquad M_{n,ij}^{\text{sgn}} = \mathbf{1}\{S_{n,ij}^{\text{sgn}} > \tau_n^{\text{sgn}}\}, \tag{73}$$

with $\varphi_{n,i} = x_i$, $\psi_{n,j} = \theta_j^{(2)}\sigma'(h_j^{(1)})$, $\xi_{n,ij} = \theta_{ij}^{(1)}$.

Meanwhile, with magnitude GraSP, we take absolute values yields

$$|\theta_{ij}^{(1)} s_{ij}| \propto |x_i| \left(|\theta_j^{(2)}|\,|\sigma'(h_j^{(1)})|\right)|\theta_{ij}^{(1)}|, \tag{74}$$

which is exactly of the non-negative factorised saliency model form:

$$S_{n,ij}^{\text{mag}} = \varphi_{n,i}\psi_{n,j}|\xi_{n,ij}|, \qquad M_{n,ij}^{\text{mag}} = \mathbf{1}\{S_{n,ij}^{\text{mag}} > \tau_n^{\text{mag}}\}, \quad \tau_n^{\text{mag}} \geq 0. \tag{75}$$

Therefore, the verification of Assumptions 4.1–4.5 for GraSP follows the same template as SNIP (Appendix B.2): row-profile quantile consistency for $\varphi_{n,i}$, empirical CDF convergence for $\psi_{n,j}$, asymptotic decoupling of $\xi_{n,ij}$ from $\psi_{n,j}$ under bounded inputs, and threshold stability/threshold replacement for top-$\rho$ pruning (Appendix B.2–B.4). The only additional ingredient for GraSP is the remainder control Eqn. (71), ensuring that the Hessian correction does not alter the limiting ranking and mask.

### C.4. Empirical verification and theoretical graphon details

We empirically validate the convergence of GraSP to its theoretical limit using the setup described in Appendix B. We vary network width $n \in \{200, \ldots, 4000\}$ and fix density $\rho = 0.2$. We exclude ReLU activations here, as the second derivative vanishes ($\sigma'' \equiv 0$) almost everywhere, rendering the Hessian term zero in standard implementations; instead, we focus on Tanh and Sigmoid to verify the non-trivial asymptotic equivalence in the smooth regime. Figure 5 presents the results. To visualise the latent structure, rows and columns are sorted by the theoretical latent factors $|\varphi(u)|$ and $\psi(v)$ respectively.

**Convergence to the limit.** For both activations, the empirical masks smooth out as $n$ increases, becoming visually indistinguishable from the theoretical prediction at $n = 4000$. This confirms that despite the complex Hessian interactions, the GraSP mask converges to a deterministic graphon.

**Theoretical Graphon Details:** While our derivation in Appendix C.2 proves that Signed and Magnitude GraSP each is rank-equivalent to its corresponding SNIP variant; the two induce different link functions, Figure 5 reveals a distinct visual difference: the *Signed* variant (rows 2 and 4) exhibits a much wider transition boundary than the *Magnitude* variant. This difference is explained by the specific link functions induced by their respective selection rules. Let $\varphi(u)\psi(v)$ be the latent signal strength and $\xi \sim \mathcal{N}(0,1)$ be the edge noise.

- **Magnitude GraSP.** This variant retains edges where $|\varphi(u)\psi(v) \cdot \xi| > \tau^{\mathrm{mag}}$. The limiting edge probability is:

$$W^{\mathrm{mag}}(u,v) = \mathbb{P}\left(|\xi| > \frac{\tau^{\mathrm{mag}}}{\varphi(u)\psi(v)}\right) = 1 - \mathrm{erf}\left(\frac{\tau^{\mathrm{mag}}}{\varphi(u)\psi(v)\sqrt{2}}\right). \tag{76}$$

  The error function decays exponentially fast. Furthermore, to target a sparsity of $\rho = 0.2$, the threshold $\tau^{\mathrm{mag}}$ must be relatively large (far into the tails), resulting in a fast transition from $W \approx 1$ to $W \approx 0$ as $\varphi(u)\psi(v)$ decreases.

- **Signed GraSP.** This variant retains edges where $\varphi(u)\psi(v) \cdot \xi > \tau^{\mathrm{sgn}}$. The limiting edge probability is:

$$W^{\mathrm{sgn}}(u,v) = \mathbb{P}\left(\xi > \frac{\tau^{\mathrm{sgn}}}{\varphi(u)\psi(v)}\right) = 1 - \Phi\left(\frac{\tau^{\mathrm{sgn}}}{\varphi(u)\psi(v)}\right), \tag{77}$$

  where $\Phi$ is the Standard Normal CDF. To achieve the *same* density (i.e., $\rho = 0.2$) using only the positive tail, the signed threshold $\tau^{\mathrm{sgn}}$ must be significantly lower (closer to the mean) than $\tau^{\mathrm{mag}}$. Operating closer to the probability mass center means that small changes in the signal $\varphi(u)\psi(v)$ result in more gradual changes in edge probability.

The rightmost column of Figure 5 plots these exact functions. The agreement between the $n = 4000$ empirical masks and their respective theoretical counterparts confirms that the wider spread in Signed GraSP is not an artifact of slow convergence, but the mathematically correct limiting behavior of a one-sided selection rule.

## D. Extension of Graphon Convergence Theorem to Deep Neural Network

Consider am $L$-hidden layer network at initialisation with widths $n_0 = d, n_1, \ldots, n_L$ and forward recursion

$$h_j^{(\ell)} := \frac{1}{\sqrt{n_{\ell-1}}} \sum_{i=1}^{n_{\ell-1}} \theta_{ij}^{(\ell)} z_i^{(\ell-1)}, \qquad z_j^{(\ell)} := \sigma\big(h_j^{(\ell)}\big), \qquad \ell = 1, \ldots, L, \tag{78}$$

where $\{\theta_{ij}^{(\ell)}\}$ are i.i.d. across entries and independent across layers, and $\sigma$ is an activation function.

In the deep factorised saliency model at layer $\ell$,

$$S_{n,ij}^{(\ell)} = \varphi_{n,i}^{(\ell)} \psi_{n,j}^{(\ell)} |\xi_{n,ij}^{(\ell)}|, \qquad M_{n,ij}^{(\ell)} = \mathbf{1}\{S_{n,ij}^{(\ell)} > \tau_n^{(\ell)}\}, \qquad i \in [n_{\ell-1}], \ j \in [n_\ell], \tag{79}$$

Now, for SNIP saliency score, we show that for a hidden layer $\ell$, Assumptions A4.1 - A4.5 are satisfied asymptotically.

**Deterministic input feature limit.** Fix a hidden layer $\ell \geq 2$. For the layer-$\ell$ mask, the input feature is the magnitude of the previous-layer activation,

$$\varphi_{n,i}^{(\ell)} = |z_i^{(\ell-1)}|, \qquad i \in [n_{\ell-1}]. \tag{80}$$

We verify that the empirical distribution of $\{\varphi_{n,i}^{(\ell)}\}_{i=1}^{n_{\ell-1}}$ has a deterministic limit. Conditional on $z^{(\ell-2)}$, the pre-activations in layer $\ell - 1$ satisfy

$$h_i^{(\ell-1)} = \frac{1}{\sqrt{n_{\ell-2}}} \sum_{r=1}^{n_{\ell-2}} \theta_{ri}^{(\ell-1)} z_r^{(\ell-2)}, \qquad i \in [n_{\ell-1}]. \tag{81}$$

Since the columns $\{\theta_{:i}^{(\ell-1)}\}_{i=1}^{n_{\ell-1}}$ are i.i.d., the random variables $\{h_i^{(\ell 1)}\}_{i=1}^{n_{\ell-1}}$ are i.i.d. conditional on $z^{(\ell-2)}$. In the Gaussian initialisation case,

$$h_i^{(\ell-1)} \mid z^{(\ell-2)} \sim \mathcal{N}\big(0, q_n^{(\ell-2)}\big), \qquad q_n^{(\ell-2)} := \frac{1}{n_{\ell-2}} \sum_{r=1}^{n_{\ell-2}} \big(z_r^{(\ell-2)}\big)^2. \tag{82}$$

Under the standard infinite-width moment assumptions, the empirical second moment satisfies

$$q_n^{(\ell-2)} \xrightarrow{\mathbb{P}} q^{(\ell-2)}, \tag{83}$$

where $q^{(\ell-2)}$ is deterministic. Hence, the conditional law of $h_i^{(\ell-1)}$ converges to the deterministic Gaussian law $\mathcal{N}\big(0, q^{(\ell-2)}\big)$.

Therefore the row feature $\varphi_{n,i}^{(\ell)} = |z_i^{(\ell-1)}|$ has the deterministic limiting law $\varphi^{(\ell)} = |\sigma(h^{(\ell-1)})|$. Let $F_{\varphi^{(\ell)}}$ denote the CDF of $\varphi^{(\ell)}$. The empirical CDF

$$\widehat{F}_{\varphi^{(\ell)},n}(t) := \frac{1}{n_{\ell-1}} \sum_{i=1}^{n_{\ell-1}} \mathbf{1}\{\varphi_{n,i}^{(\ell)} \leq t\} \tag{84}$$

then satisfies

$$\sup_{t \in \mathbb{R}} \left| \widehat{F}_{\varphi^{(\ell)},n}(t) - F_{\varphi^{(\ell)}}(t) \right| \xrightarrow{\mathbb{P}} 0. \tag{85}$$

Consequently, after sorting the row features increasingly, the empirical quantiles converge to the deterministic quantile curve

$$Q_{\varphi^{(\ell)}}(u) := \inf\{s \geq 0 : F_{\varphi^{(\ell)}}(s) \geq u\}. \tag{86}$$

In particular, at continuity points of $Q_{\varphi^{(\ell)}}$,

$$\varphi_{n,(\lceil n_{\ell-1} u \rceil)}^{(\ell)} \xrightarrow{\mathbb{P}} Q_{\varphi^{(\ell)}}(u), \qquad u \in (0,1). \tag{87}$$

This verifies the row-feature analogue of Assumption 4.2 for layer $\ell$.

**Neuron-feature empirical CDF convergence.** For hidden layers, the neuron feature should be defined through the normalised backpropagated sensitivity rather than through the raw outgoing weight norm. We have $\psi_j^{(\ell)} := |\sigma'(h_j^{(\ell)}) b_j^{(\ell)}|$, where $b_j^{(L)} = \theta_j^{(L+1)}$ for the last hidden layer, and recursively, $b_j^{(\ell)} = \frac{1}{\sqrt{n_\ell}} \sum_{k=1}^{n_{\ell+1}} \theta_{jk}^{(\ell+1)} \sigma'(h_k^{(\ell+1)}) b_k^{(\ell+1)}$.

Under the standard infinite-width moment assumptions, the forward variables satisfy $h_j^{(\ell)} \sim \mathcal{N}(0, q_{\ell-1})$ with deterministic variance recursion $q_\ell = \mathbb{E}[\sigma(h^{(\ell)})^2]$. The backward factors also have a deterministic limiting law. Starting from $b_j^{(L)} \sim \mathcal{N}(0,1)$, define recursively

$$b_j^{(\ell)} \sim \mathcal{N}(0, s_{\ell+1}), \qquad s_{\ell+1} = \mathbb{E}\left[\left(\sigma'(h^{(\ell+1)}) b^{(\ell+1)}\right)^2\right],$$

with $b^{(\ell)}$ independent of $h^{(\ell)}$ in the infinite-width limit. Hence, $\psi^{(\ell)}$ has a deterministic limiting distribution. Let $F_{\psi^{(\ell)}}$ be its CDF. Then the empirical CDF $\widehat{F}_{\psi^{(\ell)},n}(t) = \frac{1}{n_\ell} \sum_{j=1}^{n_\ell} \mathbf{1}\{\psi_{n,j}^{(\ell)} \leq t\}$ satisfies $\sup_{t \in \mathbb{R}} \left|\widehat{F}_{\psi^{(\ell)},n}(t) - F_{\psi^{(\ell)}}(t)\right| \xrightarrow{\mathbb{P}} 0$.

This verifies the layerwise analogue of Assumption 4.3.

**Edge noise asymptotic decoupling.** The subtle point is that the neuron feature

$$\psi_j^{(\ell)} = |\sigma'(h_j^{(\ell)}) b_j^{(\ell)}| \tag{88}$$

is not exactly independent of the edge noise $\theta_{ij}^{(\ell)}$. This dependence enters directly through $h_j^{(\ell)}$, and indirectly through the downstream factor $b_j^{(\ell)}$, since perturbing $h_j^{(\ell)}$ changes later-layer activations and hence the backpropagated sensitivities. For each edge $(i,j)$, define the leave-one-out pre-activation

$$h_j^{(\ell),-i} = h_j^{(\ell)} - \frac{1}{\sqrt{n_{\ell-1}}} \theta_{ij}^{(\ell)} z_i^{(\ell-1)}. \tag{89}$$

Let $b_j^{(\ell),-i}$ be the corresponding downstream factor computed in the leave-one-out network, and define

$$\psi_j^{(\ell),-i} = |\sigma'(h_j^{(\ell),-i}) b_j^{(\ell),-i}|. \tag{90}$$

Conditional on all weights and activations except $\theta_{ij}^{(\ell)}$, the leave-one-out feature $\psi_j^{(\ell),-i}$ is independent of $\theta_{ij}^{(\ell)}$.

Moreover, by Lipschitzness of $\sigma'$,

$$|\psi_j^{(\ell)} - \psi_j^{(\ell),-i}| \leq \frac{L_{\sigma'}}{\sqrt{n_{\ell-1}}} |b_j^{(\ell)}| |\theta_{ij}^{(\ell)}| |z_i^{(\ell-1)}| + \|\sigma'\|_\infty |b_j^{(\ell)} - b_j^{(\ell),-i}|. \tag{91}$$

The first term is $O(n_{\ell-1}^{-1/2})$ after averaging over $(i,j)$, using the uniform moment bounds on $b_j^{(\ell)}$, $\theta_{ij}^{(\ell)}$, and $z_i^{(\ell-1)}$. The second term is controlled by the standard forward–backward stability recursion: the perturbation created by removing one edge has normalised size $O((n_{\ell-1} n_\ell)^{-1/2})$ at layer $\ell$, remains of this order as it propagates forward through the fixed-depth network, and therefore induces an averaged backward-factor perturbation

$$\frac{1}{n_{\ell-1} n_\ell} \sum_{i=1}^{n_{\ell-1}} \sum_{j=1}^{n_\ell} |b_j^{(\ell)} - b_j^{(\ell),-i}| = O(n_{\ell-1}^{-1/2}). \tag{92}$$

Consequently,

$$\frac{1}{n_{\ell-1} n_\ell} \sum_{i=1}^{n_{\ell-1}} \sum_{j=1}^{n_\ell} |\psi_j^{(\ell)} - \psi_j^{(\ell),-i}| \xrightarrow{\mathbb{P}} 0. \tag{93}$$

Thus the original layer-$\ell$ scores and their leave-one-out decoupled versions differ by a vanishing average perturbation. At continuity points of the limiting score distribution, the corresponding thresholded masks differ on a vanishing fraction of entries, so their step-kernels differ by $o(1)$ in $L^1$, and hence also in cut norm.

**Threshold stability.** For top-$\rho$ pruning, let $\tau_n^{(\ell)}$ be the empirical $(1-\rho)$-quantile of the layer-$\ell$ scores. The layerwise empirical score distribution converges to a deterministic limit distribution. Assuming this limit distribution is continuous at its $(1-\rho)$-quantile, we obtain $\tau_n^{(\ell)} \xrightarrow{\mathbb{P}} \tau^{(\ell)}$. Replacing $\tau_n^{(\ell)}$ by $\tau^{(\ell)}$ changes only a vanishing fraction of mask entries.

**Corollary D.1** (Layerwise graphon convergence in deep neural networks). *Fix a depth $L \in \mathbb{N}$. For each layer $\ell = 1, \ldots, L$, let $M_n^{(\ell)} \in \{0,1\}^{n_{\ell-1} \times n_\ell}$ be the PaI mask at layer $\ell$ generated by a factorised saliency model*

$$S_{n,ij}^{(\ell)} = \varphi_{n,i}^{(\ell)} \psi_{n,j}^{(\ell)} |\xi_{n,ij}^{(\ell)}|, \qquad M_{n,ij}^{(\ell)} = \mathbf{1}\{S_{n,ij}^{(\ell)} > \tau_n^{(\ell)}\}. \tag{94}$$

*Let $W_n^{(\ell)} := W_{M_n^{(\ell)}}$ be the associated bipartite step-kernel. Assume that for each fixed $\ell$, the layerwise analogues of Assumptions A4.1 - A4.5 hold, with deterministic limiting graphon $\mathcal{W}^{(\ell)} \in L^\infty([0,1]^2)$. Then, as $\min_\ell n_\ell \to \infty$,*

$$\max_{\ell \in [L]} \delta_\square^{\mathrm{bip}}(W_n^{(\ell)}, \mathcal{W}^{(\ell)}) \xrightarrow{\mathbb{P}} 0. \tag{95}$$

*Proof sketch.* Fix $\ell \in [L]$. By the layerwise Assumptions A4.1 - A4.5, Theorem 4.7 applies to the bipartite mask matrix $M_n^{(\ell)}$ and yields $\delta_\square^{\mathrm{bip}}(W_n^{(\ell)}, \mathcal{W}^{(\ell)}) \to 0$ in probability. Moreover, in the deep setting the *input-feature* sequence at layer $\ell$ can be taken as $\varphi_{n,i}^{(\ell)} = z_i^{(\ell-1)}$, and as shown above the post-activations $\{z_i^{(\ell-1)}\}_{i=1}^{n_{\ell-1}}$ are (asymptotically) i.i.d. with a deterministic limit law. Hence, $\{\varphi_{n,i}^{(\ell)}\}$ satisfies the required input-feature assumption. Finally, since $L$ is fixed, a union bound gives for any $\varepsilon > 0$,

$$\mathbb{P}\left(\max_{\ell \leq L} \delta_\square^{\mathrm{bip}}(W_n^{(\ell)}, \mathcal{W}^{(\ell)}) > \varepsilon\right) \leq \sum_{\ell=1}^{L} \mathbb{P}\left(\delta_\square^{\mathrm{bip}}(W_n^{(\ell)}, \mathcal{W}^{(\ell)}) > \varepsilon\right) \to 0, \tag{96}$$

$\square$

### D.1. Empirical Verification for 2-Hidden Layer Neural Networks

We extend our empirical analysis beyond the single-hidden-layer setting to verify the Graphon Limit Hypothesis in deep architectures. We examine the convergence behavior of a 2-hidden-layer fully connected network ($d \to n \to n \to 1$) under varying sparsity levels $\rho \in \{0.1, 0.2\}$ and activation functions $\sigma \in \{\mathrm{ReLU}, \mathrm{Tanh}, \mathrm{Sigmoid}\}$.

**Experimental Setup.** Inputs are drawn from a standard normal distribution $x \sim \mathcal{N}(0, I_d)$. We generate SNIP masks for networks of increasing width $n \in \{200, 500, 1000, 2000, 4000\}$. To visualise the underlying graphon structure, the discrete adjacency matrices are sorted according to their latent factors:

- Layer 1 ($\mathcal{W}^{(1)}$): Rows are sorted by the input magnitude $|x_i|$.

- Layer 2 ($\mathcal{W}^{(2)}$): Rows are sorted by the magnitude of the incoming activations $|\sigma(h_j^{(1)})|$, where $h^{(1)}$ are the pre-activations of the first layer.

The results are averaged over 20 random seeds to smooth out finite-sample noise.

**Theoretical Limit Construction.** While the limit for layer 1 follows the Half-Normal profile derived in Section 4, the limit for layer 2 is derived recursively. As $n \to \infty$, the pre-activations $h^{(1)}$ converge to a Gaussian process with distribution $\mathcal{N}(0, 1)$ (up to variance scaling). Consequently, the theoretical row factors for layer 2 are modelled by the quantile function of $|\sigma(Z)|$ where $Z \sim \mathcal{N}(0, 1)$.

**Results and Discussion.** Figures 6 - 11 illustrate the convergence of empirical masks to the theoretical limits. Across all activations and densities, the empirical masks converge in the cut-metric topology to the predicted deterministic graphons. At $n = 4000$, the empirical masks are visually indistinguishable from the theoretical limit.

The choice of activation function significantly affects the topology of the graphon. For ReLU activations, since weights are initialised follow Normal distribution, the pre-activation $h$ is symmetric around 0, leading to roughly half of the columns in layer 1 graphon being pruned. Then, in the second layer, both the inputs and the gradients are subject to the ReLU cutoff. This explains why it looks like three-quarters of the graphon are pruned in layer 2. In contrast, for activations like Tanh and Sigmoid, the high-density region in layer 2 appears more diffuse.

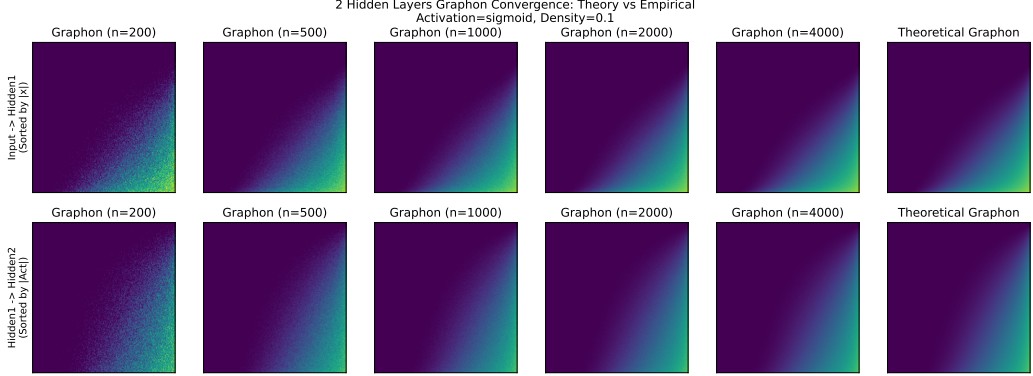

*Figure 6.* Graphon convergence in 2-hidden layer networks with sigmoid activation function on two densities $\rho = 0.1$.

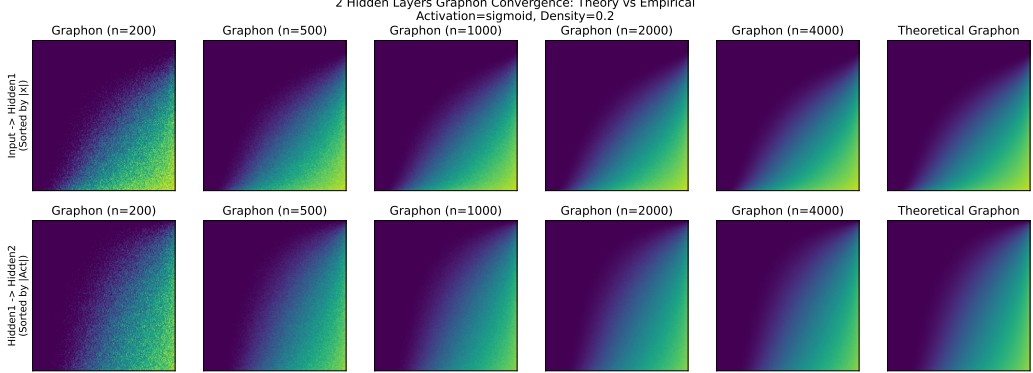

*Figure 7.* Graphon convergence in 2-hidden layer networks with sigmoid activation function on two densities $\rho = 0.2$.

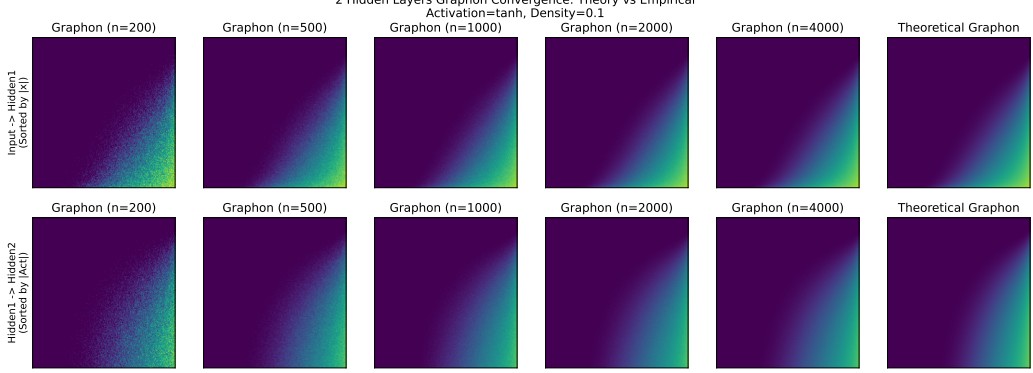

*Figure 8.* Graphon convergence in 2-hidden layer networks with tanh activation function on two densities $\rho = 0.1$.

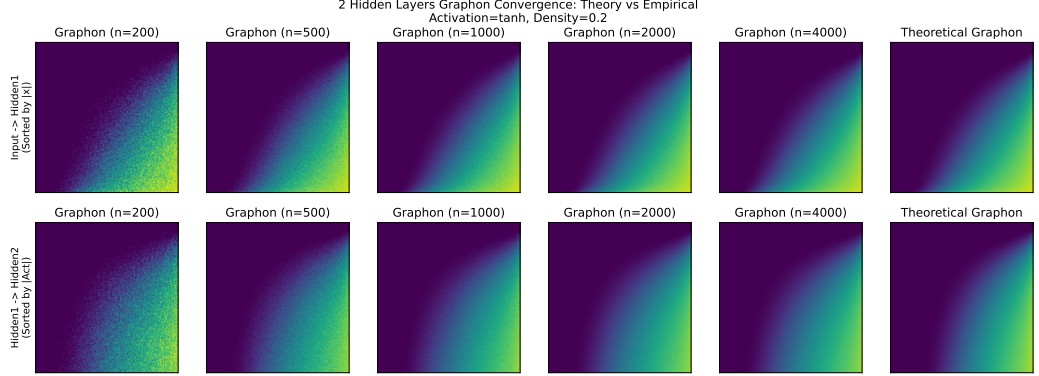

*Figure 9.* Graphon convergence in 2-hidden layer networks with tanh activation function on two densities $\rho = 0.2$.

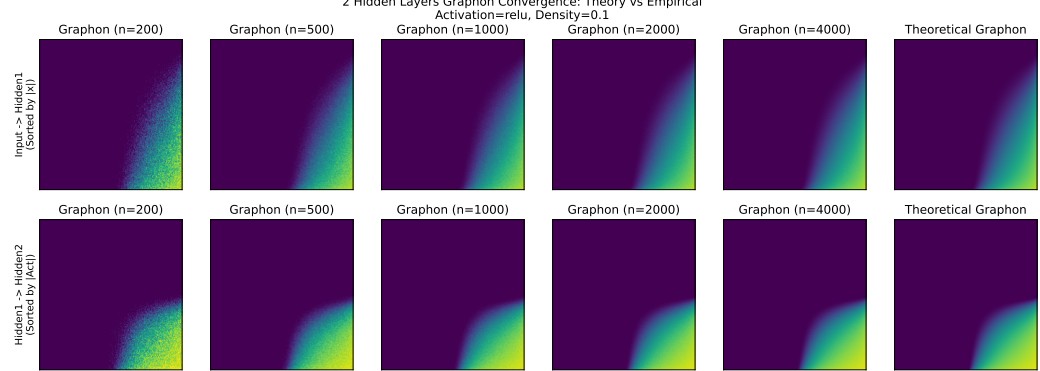

*Figure 10.* Graphon convergence in 2-hidden layer networks with ReLU activation function on two densities $\rho = 0.1$.

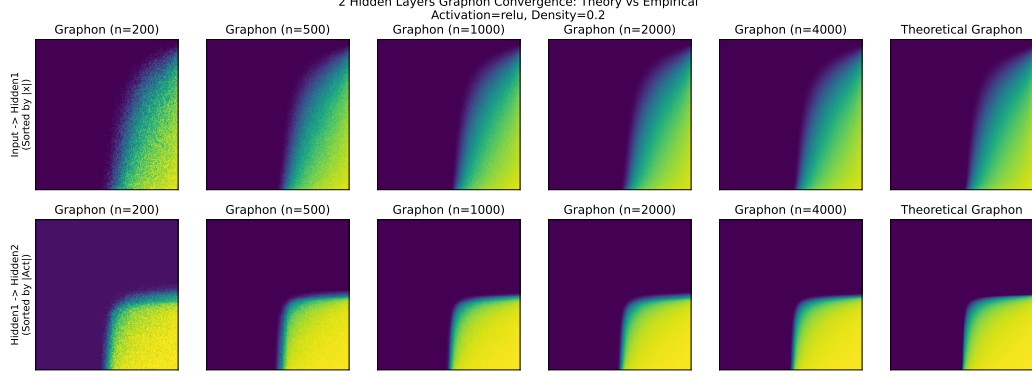

*Figure 11.* Graphon convergence in 2-hidden layer networks with ReLU activation function on two densities $\rho = 0.2$.

# E. Proofs of Results in Section 5

In this appendix, we prove Lemma 5.6 and Theorem 5.7. Throughout, the mask is defined by $M_{ij} = \mathbf{1}\{\varphi_i\,\psi_j\,|\xi_{ij}| > \tau\}$, and conditioned on $(\varphi, \psi)$ the noises $(\xi_{ij})_{i \leq d, j \leq n}$ are independent with common law $P_\xi$. Recall the survival link $g(\tau, z) := \mathbb{P}(z|\xi| > \tau)$ so that

$$\mathbb{P}(M_{ij} = 1 \mid \varphi, \psi) = g(\tau, \varphi_i \psi_j). \tag{97}$$

We work in the sorted coordinates $u_i = i/d$ and $v_j = j/n$ where $\max_{i \leq d} |\varphi_i - \varphi(u_i)| \to 0$ and $\max_{j \leq n} |\psi_j - Q_\psi(v_j)| \to 0$, and the limiting edge-probability kernel is

$$\mathcal{W}(u, v) = g(\tau, \varphi(u)Q_\psi(v)). \tag{98}$$

## E.1. A counting argument for the active column set

Let $V_\star \subset (0, 1)$ be as in Assumption 5.2 with $\lambda(V_\star) = \alpha > 0$. Define the set of *active columns*

$$J_n := \{j \in [n] : v_j = j/n \in V_\star\}. \tag{99}$$

Since $v_j = j/n$ is deterministic, we have $|J_n|/n \to \alpha$ provided $V_\star$ is Riemann measurable. that is $\partial V_\star$ has Lebesgue measure 0, or $V_\star$ is a finite union of intervals. In particular, for all sufficiently large $n$, it holds that

$$|J_n| \geq \frac{\alpha}{2} n. \tag{100}$$

## E.2. Proof of Lemma 5.6

*Proof of Lemma 5.6.* Fix $\tilde{n} \in \mathbb{N}$ and let $k := |I|$. Let $J_n$ be the deterministic active column set defined in Eqn. (99). First, we show a uniform edge-probability lower bound on the active rectangle. By Assumption 5.2, there exists $p_\star > 0$ such that $\mathcal{W}(u, v) \geq p_\star$ for a.e. $(u, v) \in U_\star \times V_\star$. Since $g(\tau, \cdot)$ is nondecreasing and continuous on bounded intervals, and since $\varphi_i \to \varphi(u_i)$ and $\psi_j \to Q_\psi(v_j)$ uniformly, we can choose $n$ large enough so that on an event $E_n$ with $\mathbb{P}(E_n) \to 1$,

$$\min_{i \in I} \min_{j \in J_n} g(\tau, \varphi_i \psi_j) \geq \frac{p_\star}{2}. \tag{101}$$

Secondly, we define good columns and compute their success probabilities. For each $j \in J_n$, define the indicator that column $j$ is fully connected to the $k$ active inputs:

$$G_j := \mathbf{1}\Big\{M_{ij} = 1 \text{ for all } i \in I\Big\}. \tag{102}$$

Let $\mathcal{G} := \sigma(\varphi, \psi)$ be the sigma-field generated by the row or column features. Conditional on $\mathcal{G}$, the edge noises are independent across $(i, j)$, hence the events $\{M_{ij} = 1\}_{i \in I}$ are independent for each fixed $j$. Therefore,

$$\mathbb{P}(G_j = 1 \mid \mathcal{G}) = \prod_{i \in I} \mathbb{P}(M_{ij} = 1 \mid \mathcal{G}) = \prod_{i \in I} g(\tau, \varphi_i \psi_j). \tag{103}$$

On the event $E_n$ from Eqn. (101), we obtain the lower bound

$$\mathbb{P}(G_j = 1 \mid \mathcal{G}) \geq \left(\frac{p_\star}{2}\right)^k \qquad \text{for all } j \in J_n. \tag{104}$$

Moreover, conditional on $\mathcal{G}$, the random variables $(G_j)_{j \in J_n}$ are independent: indeed, $G_j$ depends only on the noises $\{\xi_{ij} : i \in I\}$, which are disjoint across distinct columns $j$.

Next, we use Chernoff bound for the number of good columns. Particularly, we define the number of good columns

$$N_{\text{good}} := \sum_{j \in J_n} G_j. \tag{105}$$

Conditionws on $\mathcal{G}$ and on $E_n$, the $(G_j)_{j \in J_n}$ are independent Bernoulli variables with success probabilities bounded below as in Eqn. (104). Hence the conditional mean satisfies

$$\mu_n := \mathbb{E}[N_{\text{good}} \mid \mathcal{G}] = \sum_{j \in J_n} \mathbb{P}(G_j = 1 \mid \mathcal{G}) \geq |J_n|\left(\frac{p_\star}{2}\right)^k. \tag{106}$$

Using the deterministic bound Eqn. (100), we obtain on $E_n$:

$$\mu_n \; \geq \; \frac{\alpha}{2} n \Big(\frac{p_\star}{2}\Big)^k. \tag{107}$$

We now apply a standard multiplicative Chernoff bound for sums of independent Bernoulli variables: for any $\delta \in (0,1)$,

$$\mathbb{P}\Big(N_{\text{good}} \leq (1-\delta)\mu_n \,\Big|\, \mathcal{G}\Big) \; \leq \; \exp\Big(-\frac{\delta^2}{2}\mu_n\Big). \tag{108}$$

Take $\delta = 1/2$. Then on $E_n$,

$$\mathbb{P}\Big(N_{\text{good}} \leq \tfrac{1}{2}\mu_n \,\Big|\, \mathcal{G}\Big) \; \leq \; \exp(-\mu_n/8). \tag{109}$$

Then, we ensure at least $\tilde{n}$ good columns exist. By Assumption 5.4, $(\alpha n)p_\star^k \to \infty$, the lower bound Eqn. (107) implies $\mu_n \to \infty$, and therefore $\exp(-\mu_n/8) \to 0$. In particular, for all large $n$ we have $\mu_n/2 \geq \tilde{n}$ and

$$\mathbb{P}\Big(N_{\text{good}} < \tilde{n} \,\Big|\, \mathcal{G}\Big) \; \leq \; \mathbb{P}\Big(N_{\text{good}} \leq \tfrac{1}{2}\mu_n \,\Big|\, \mathcal{G}\Big) \; \leq \; \exp(-\mu_n/8). \tag{110}$$

Taking expectations and using $\mathbb{P}(E_n) \to 1$ yields

$$\mathbb{P}\big(N_{\text{good}} \geq \tilde{n}\big) \; \to \; 1. \tag{111}$$

On this event, choose any $J \subset J_n$ of size $\tilde{n}$ consisting of good columns; then by definition of $G_j$ we have $M_{ij} = 1$ for all $i \in I$ and $j \in J$, which proves the lemma. $\qquad\square$

### E.3. Proof of Theorem 5.7

*Proof of Theorem 5.7.* Fix $\varepsilon > 0$. By the classical universal approximation theorem on $\mathbb{R}^k$ (Cybenko, 1989), recalled as Theorem 5.1 in our paper, there exists an integer $\tilde{n}_\varepsilon < \infty$ and parameters $\{(a_r, \theta_r, b_r)\}_{r=1}^{\tilde{n}_\varepsilon}$ such that

$$\sup_{u \in K}\Big|f(u) - \sum_{r=1}^{\tilde{n}_\varepsilon} a_r\, \sigma(\theta_r^\top u + b_r)\Big| < \varepsilon. \tag{112}$$

Let $I$ be the active input set from Assumption 5.3 and define $\tilde{f}(x) = f(x_I)$ and $\tilde{K} = \{x \in \mathbb{R}^d : x_I \in K, \|x_{I^c}\|_\infty \leq B\}$ as in the theorem. By Lemma 5.6, with probability $1 - o(1)$ there exists a set $J = \{j_1, \ldots, j_{\tilde{n}_\varepsilon}\} \subset [n]$ such that $M_{ij_r} = 1$ for all $i \in I$ and all $r \leq \tilde{n}_\varepsilon$. On this event, we construct $F_n \in \mathcal{F}_n(M)$ by choosing parameters $(a, b, \theta)$ in Eqn. (7) as follows:

- For each $r \leq \tilde{n}_\varepsilon$, set $a_{j_r} := a_r$ and $b_{j_r} := b_r$.

- For each $r \leq \tilde{n}_\varepsilon$ and each $i \in I$, set $(\theta_{ij_r})_{i \in I} := \theta_r$, so that $\sum_{i \in I} \theta_{ij_r} x_i = \theta_r^\top x_I$.

- Set all remaining parameters $a_j, b_j, \theta_{ij}$ (for $j \notin J$ or $i \notin I$) equal to 0.

Because $M_{ij_r} = 1$ for all $i \in I$, these weights are not masked out, and thus for all $x \in \mathbb{R}^d$, we have

$$F_n(x) = \sum_{r=1}^{\tilde{n}_\varepsilon} a_r\, \sigma\Big(b_r + \sum_{i \in I} \theta_{ij_r} x_i\Big) = \sum_{r=1}^{\tilde{n}_\varepsilon} a_r\, \sigma\big(b_r + \theta_r^\top x_I\big). \tag{113}$$

Hence, for any $x \in \tilde{K}$ we have $x_I \in K$ and therefore by Eqn. (112),

$$|F_n(x) - \tilde{f}(x)| = \Big|\sum_{r=1}^{\tilde{n}_\varepsilon} a_r\sigma(\theta_r^\top x_I + b_r) - f(x_I)\Big| \leq \sup_{u \in K}\Big|\sum_{r=1}^{\tilde{n}_\varepsilon} a_r\sigma(\theta_r^\top u + b_r) - f(u)\Big| < \varepsilon. \tag{114}$$

Taking the supremum over $x \in \tilde{K}$ gives $\sup_{x \in \tilde{K}} |F_n(x) - \tilde{f}(x)| < \varepsilon$ on the same high-probability event. This proves

$$\mathbb{P}\Big(\exists\, F_n \in \mathcal{F}_n(M) \text{ s.t. } \sup_{x \in \tilde{K}} |F_n(x) - \tilde{f}(x)| < \varepsilon\Big) \to 1. \tag{115}$$

$\qquad\square$

# F. Proof of Corollary 6.1

In this appendix, we prove Corollary 6.1. The argument works as follows: in the infinite-width Graphon-NTK regime, training the graphon-masked network induces the same output predictor as kernel (S)GD with the Graphon-NTK Gram matrix $K_{\mathcal{W}}$ on the fixed sample $S$; we can then invoke Cao & Gu (2019, Cor. 3.10) with $K_{\mathcal{W}}$ in place of the finite-width NTK Gram matrix.

## F.1. Preliminaries: predictions, Jacobian, and the empirical NTK Gram matrix

Let $S = \{(x_i, y_i)\}_{i=1}^m$ be i.i.d. samples from $\mathcal{D}$. Let $f_\theta : \mathcal{X} \to \mathbb{R}$ denote the masked network predictor. Define the prediction vector on the sample by

$$f(\theta) := \big(f_\theta(x_1), \ldots, f_\theta(x_m)\big)^\top \in \mathbb{R}^m. \tag{116}$$

Let the empirical risk be

$$\mathcal{L}_S(\theta) := \frac{1}{m} \sum_{i=1}^m \ell\big(f_\theta(x_i), y_i\big), \tag{117}$$

where $\ell$ is a differentiable surrogate loss like logistic or cross-entropy. Let $J(\theta) \in \mathbb{R}^{m \times p}$ denote the Jacobian of $f(\theta)$ with respect to parameters $\theta \in \mathbb{R}^p$: $J(\theta)_{i,:} = \nabla_\theta f_\theta(x_i)^\top$. Define the empirical NTK Gram matrix as

$$K(\theta) := J(\theta) J(\theta)^\top \in \mathbb{R}^{m \times m}. \tag{118}$$

**Training dynamics.** Under gradient flow $\dot{\theta}(t) = -\nabla_\theta \mathcal{L}_S(\theta(t))$, the chain rule gives the exact prediction dynamics

$$\dot{f}(t) = -K(\theta(t)) \nabla_f \mathcal{L}_S\big(f(t)\big), \tag{119}$$

where $\nabla_f \mathcal{L}_S(f) \in \mathbb{R}^m$ denotes the gradient of $\mathcal{L}_S$ w.r.t. the prediction vector. For full-batch gradient descent $\theta_{t+1} = \theta_t - \eta \nabla_\theta \mathcal{L}_S(\theta_t)$, one obtains the standard first-order approximation in prediction space

$$f_{t+1} \approx f_t - \eta K(\theta_t) \nabla_f \mathcal{L}_S(f_t), \tag{120}$$

up to a second-order term controlled by smoothness of $f(\theta)$ and the stepsize $\eta$.

## F.2. Graphon-NTK and lazy training regime

Let $M_n$ be the PaI mask at width $n$, and suppose the associated step kernel converges to a limiting graphon $W_{M_n} \to \mathcal{W}$ (Section 4). Let $\Theta_{\mathcal{W}}$ denote the Graphon-NTK kernel defined in Pham et al. (2025), and define the deterministic Graphon-NTK Gram matrix on $S$ by

$$(K_{\mathcal{W}})_{ij} := \Theta_{\mathcal{W}}(x_i, x_j). \tag{121}$$

We work under the following standard NTK lazy training assumption specialised to the graphon-masked setting.

**Assumption F.1** (Infinite-width Graphon-NTK lazy training regime). Fix a training horizon $T \in \mathbb{N}$ for GD, or $T > 0$ for gradient flow and a stepsize schedule. As width $n \to \infty$, we assume:

1. **Deterministic kernel limit at initialization:** $K_n(\theta_{n,0}) \xrightarrow{\mathbb{P}} K_{\mathcal{W}}$ on the fixed sample $S$.

2. **Kernel stability along training (lazy training):** $\sup_{t \leq T} \|K_n(\theta_{n,t}) - K_n(\theta_{n,0})\|_{\mathrm{op}} \xrightarrow{\mathbb{P}} 0$.

3. **Kernelised output predictor:** The network output predictor used in Cao & Gu (2019, Cor. 3.10) converges in law to the corresponding output predictor obtained by running kernel gradient descent using Gram matrix $K_{\mathcal{W}}$ on $S$ with the same stepsize schedule.

Assumption F.1 places our analysis in the Graphon-NTK lazy training regime, in which the masked-network training dynamics on a fixed sample are governed by the limiting kernel $K_{\mathcal{W}}$. The assumption is supported by the same mechanism as in classical NTK analyses: under NTK scaling, each individual parameter moves by a vanishing amount as widths grow, which in turn keeps the network Jacobian, and hence the empirical NTK, nearly constant throughout training (Lee et al., 2019a; Arora et al., 2019b; Chizat et al., 2019).

To see the intuition, let $f(\theta) \in \mathbb{R}^m$ be the vector of predictions on the training set and let $J(\theta) \in \mathbb{R}^{m \times p}$ be the Jacobian with entries $J_{i,:} = \nabla_\theta f_\theta(x_i)^\top$. The empirical NTK Gram matrix is $K(\theta) = J(\theta)J(\theta)^\top$. Gradient flow satisfies the exact identity

$$\dot{f}(t) = -K(\theta(t)) \nabla_f \mathcal{L}_S(f(t)), \tag{122}$$

so kernel-like dynamics follow once $K(\theta(t))$ remains close to $K(\theta(0))$.

In sparse networks, weights are still independent at initialisation but have variances modulated by the layer graphons. Under bounded graphons and standard Lindeberg/LLN conditions, as used to establish the deterministic Graphon-NTK limit (Pham et al., 2025), forward activations remain $O(1)$ and backpropagated sensitivities obey the usual NTK scaling. Consequently, the gradient of the loss with respect to any single weight is of order $O((n_{\ell-1}n_\ell)^{-1/2})$ in layer $\ell$, so each parameter update is $o(1)$ as width grows. Aggregating across finitely many training steps, this implies that weights remain close to their initial values in a row-wise sense, pre-activations shift by $o(1)$, and therefore $\sigma'$ remain stable. Since the Jacobian $J(\theta)$ is composed of products of these stable forward/backward factors, it follows that

$$\sup_{t \leq T} \big\| J(\theta(t)) - J(\theta(0)) \big\|_{\mathrm{op}} \to 0 \qquad \text{and} \qquad \sup_{t \leq T} \big\| K(\theta(t)) - K(\theta(0)) \big\|_{\mathrm{op}} \to 0, \tag{123}$$

for any fixed horizon $T$, yielding lazy training. A complete proof in the graphon-modulated setting would follow the standard NTK blueprint, with two additional technical steps: (i) concentration/CLT bounds for non-i.i.d. sums induced by the graphon variance profile, and (ii) uniform control over neurons and layers of forward and backward moments to ensure no small subset of nodes dominates the dynamics. Both are compatible with the bounded-graphon assumptions used to define the Graphon-NTK limit (Pham et al., 2025).

### F.3. Proof of Corollary 6.1

We first recall from Section 3 of (Cao & Gu, 2019) the definition of the associated output predictor $\hat{f}$ in Corollary 6.1. For a neural network with input $x \in \mathbb{R}^d$ and output $f_\theta(x)$ The associated output predictor $\hat{f}$ is defined as

$$\hat{f}(x) := \mathbf{1}\big\{y \cdot f_\theta(x) < 0\big\}. \tag{124}$$

*Proof of Corollary 6.1.* Fix $\delta \in (0, e^{-1})$ and assume $\lambda_{\min}(K_{\mathcal{W}}) \geq \lambda_0 > 0$. By Assumption F.1, in the infinite-width Graphon-NTK (lazy) regime, the output predictor $\hat{f}$ produced by training the graphon-masked network coincides with the output predictor produced by kernel gradient descent using Gram matrix $K_{\mathcal{W}}$ on the same sample $S$, with the same output convention as in Cao & Gu (2019, Cor. 3.10).

Therefore, we apply Cao & Gu (2019, Cor. 3.10) with the Gram matrix $\Theta^{(L)}$ in their bound replaced by $K_{\mathcal{W}}$. This yields that, with probability at least $1 - \delta$ over $S$,

$$\mathbb{E}\Big[\mathcal{L}_{0\text{-}1}^{\mathcal{D}}(\hat{f})\Big] \leq C\,L \cdot \sqrt{\frac{y^\top K_{\mathcal{W}}^{-1} y}{m}} + C\sqrt{\frac{\log(1/\delta)}{m}},$$

where $y = (y_1, \ldots, y_m)^\top$, $C > 0$ is the constant from Cao & Gu (2019), and $L$ is the depth factor appearing there. This is exactly the bound in Eqn. (9). $\square$

# G. Path-density proxy for the Graphon-NTK

From Section 4, masks generated by PaI method converge to a specific graphon, which describes the infinite-width limit of the induced sparsity. Given such converged graphon $\mathcal{W}$, (Pham et al., 2025) defines the associated *Graphon-NTK kernel* $\Theta_{\mathcal{W}} : \mathcal{X} \times \mathcal{X} \to \mathbb{R}$ as the deterministic infinite-width limit of the neural tangent kernel for the graphon-modulated network. On the sample $S$, and $L$-hidden-layer network, we define the *Graphon-NTK Gram matrix* $(K_{\mathcal{W}})_{ij} := \Theta_{\mathcal{W}}(x_i, x_j) \in \mathbb{R}^{m \times m}$:

$$K_{\mathcal{W}}(x, x') = \sum_{\ell=1}^{L} \int_0^1 \Sigma^{(\ell-1)}(u_{\ell-1}; x, x') \, du_{\ell-1}$$

$$\int_0^1 \dot{\Sigma}^{(\ell)}(u_\ell; x, x') \left( \int_{[0,1]^{L-\ell+1}} \prod_{k=\ell+1}^{L+1} \mathcal{W}^{(k)}(u_{k-1}, u_k) \, \dot{\Sigma}^{(k)}(u_k; x, x') \, du_{\ell+1:L+1} \right) du_\ell, \tag{125}$$

## G.1. Setup and notation

We consider a 2-hidden layer graphon network, with graphons $\mathcal{W}^{(1)}, \mathcal{W}^{(2)}, \mathcal{W}^{(3)} : [0,1]^2 \to [0,1]$. Let's $u_0$ index the input coordinate, $u_1, u_2$ index the two hidden layers, and $u_3$ index the output side. For inputs $x, x'$, the Graphon-NTK decomposes as $K_{\mathcal{W}} = K_{\mathcal{W}}^{(1)} + K_{\mathcal{W}}^{(2)}$ where

$$K_{\mathcal{W}}^{(1)}(x, x') = B_0(x, x') \int_0^1 \dot{\Sigma}^{(1)}(u_1; x, x') \, P_1(u_1; x, x') \, du_1, \tag{126}$$

$$K_{\mathcal{W}}^{(2)}(x, x') = B_1(x, x') \int_0^1 \dot{\Sigma}^{(2)}(u_2; x, x') \, P_2(u_2; x, x') \, du_2, \tag{127}$$

with

$$B_0(x, x') := \int_0^1 \Sigma^{(0)}(u_0; x, x') \, du_0, \qquad B_1(x, x') := \int_0^1 \Sigma^{(1)}(u_1; x, x') \, du_1, \tag{128}$$

and the downstream passages

$$P_2(u_2; x, x') := \int_0^1 \mathcal{W}^{(3)}(u_2, u_3) \, \dot{\Sigma}^{(3)}(u_3; x, x') \, du_3, \tag{129}$$

$$P_1(u_1; x, x') := \int_0^1 \mathcal{W}^{(2)}(u_1, u_2) \, \dot{\Sigma}^{(2)}(u_2; x, x') \, P_2(u_2; x, x') \, du_2. \tag{130}$$

The pre-activation Gaussian cross-covariance at layer 1 is

$$\widetilde{\Sigma}^{(1)}(u_1; x, x') = \int_0^1 \mathcal{W}^{(1)}(u_0, u_1) \, \Sigma^{(0)}(u_0; x, x') \, du_0. \tag{131}$$

**Assumption G.1** (Activation regularity). $\sigma$ is a.e. twice differentiable with $\|\sigma'\|_\infty \le C_{\sigma,1}$ and $\|\sigma''\|_\infty \le C_{\sigma,2}$.

The key steps of the proof are: (i) decompose $K_{\mathcal{W}} = K_{\mathcal{W}}^{(1)} + K_{\mathcal{W}}^{(2)}$ by layer contributions; (ii) bound activation covariances and derivative activation covariances by pre-activation covariances; (iii) recursively substitute graphon-weighted integrals and apply Fubini to isolate the path density term.

## G.2. Two Gaussian comparison lemmas

**Lemma G.2** (Gaussian covariance contraction). *Let $(Z, Z')$ be centered jointly Gaussian. Then*

$$\left| \operatorname{Cov}(\sigma(Z), \sigma(Z')) \right| \le C_{\sigma,1}^2 \left| \operatorname{Cov}(Z, Z') \right|. \tag{132}$$

*Equivalently, if $F(s) := \mathbb{E}[\sigma(Z_s)\sigma(Z'_s)]$ where $(Z_s, Z'_s)$ is centered Gaussian with fixed marginal variances and cross-covariance $s$, then*

$$|F(s) - F(0)| \le C_{\sigma,1}^2 |s|. \tag{133}$$

*Proof.* Fix the marginals and interpolate the cross-covariance: let $(Z_t, Z'_t)$ be centered Gaussian with $\operatorname{Cov}(Z_t, Z'_t) = t \operatorname{Cov}(Z, Z')$ for $t \in [0,1]$, and set $f(t) := \mathbb{E}[\sigma(Z_t)\sigma(Z'_t)]$. Then $f(1) - f(0) = \operatorname{Cov}(\sigma(Z), \sigma(Z'))$ since $t = 0$ yields independence. A standard Gaussian differentiation identity gives $f'(t) = \operatorname{Cov}(Z, Z') \, \mathbb{E}[\sigma'(Z_t)\sigma'(Z'_t)]$. Using $|\sigma'| \le C_{\sigma,1}$ yields $|f'(t)| \le C_{\sigma,1}^2 |\operatorname{Cov}(Z, Z')|$, and integrating over $t \in [0,1]$ gives the claim. $\square$

**Lemma G.3** (Lipschitz dependence of $\dot{\Sigma}$ on cross-covariance). *Let $(Z_s, Z_s')$ be centered Gaussian with fixed marginal variances and cross-covariance $s$. Define $\dot{F}(s) := \mathbb{E}[\sigma'(Z_s)\sigma'(Z_s')]$. Then $\dot{F}$ is Lipschitz and*

$$|\dot{F}(s_1) - \dot{F}(s_2)| \leq C_{\sigma,2}^2 |s_1 - s_2|. \tag{134}$$

*In particular, we have*

$$\left|\dot{F}(s) - \dot{F}(0)\right| \leq C_{\sigma,2}^2 |s|. \tag{135}$$

*Proof.* We apply Lemma G.2 with the activation replaced by $\sigma'$. Indeed, $(\sigma')'$ exists a.e. and is bounded by $\|\sigma''\|_\infty \leq C_{\sigma,2}$, hence

$$\left| \text{Cov}(\sigma'(Z), \sigma'(Z')) \right| \leq C_{\sigma,2}^2 |\text{Cov}(Z, Z')|. \tag{136}$$

For the Gaussian family indexed by $s$, the centered covariance of $\sigma'(Z_s)$ and $\sigma'(Z_s')$ equals $\dot{F}(s) - \dot{F}(0)$, which gives the desired inequality. □

## G.3. Path-density bounds for $K_{\mathcal{W}}^{(1)}$ and $K_{\mathcal{W}}^{(2)}$

**Bounding $K_{\mathcal{W}}^{(1)}$ by path density.** Since the activation derivative is bounded $\|\sigma'\|_\infty \leq C_{\sigma,1}$, then the expected corelation between activation derivative $\dot{\Sigma}^{(l)}(\sigma'(z), \sigma'(z')) \leq C_{\sigma,1}^2$ From Eqn. (129) and Eqn. (130), we have

$$0 \leq |P_1(u_1; x, x')| \leq C_{\sigma,1}^4 \int_{[0,1]^2} \mathcal{W}^{(2)}(u_1, u_2)\, \mathcal{W}^{(3)}(u_2, u_3)\, du_2 du_3. \tag{137}$$

Next, let us fix $u_1, x, x'$. The quantity $\dot{\Sigma}^{(1)}(u_1; x, x')$ is of the form $\dot{F}(s)$ with $s = \widetilde{\Sigma}^{(1)}(u_1; x, x')$. By Lemma G.3, we have

$$\left|\text{Cov}\big(\sigma'(z^{(1)}(u_1, x)), \sigma'(z^{(1)}(u_1, x'))\big)\right| \leq C_{\sigma,2}^2 \left|\widetilde{\Sigma}^{(1)}(u_1; x, x')\right|, \tag{138}$$

$$\left|\mathbb{E}\big[\sigma'(z^{(1)}(u_1, x))\sigma'(z^{(1)}(u_1, x'))\big] - \mathbb{E}\big[\sigma'(z^{(1)}(u_1, x))\big]\mathbb{E}\big[\sigma'(z^{(1)}(u_1, x'))\big]\right| \leq C_{\sigma,2}^2 \left|\widetilde{\Sigma}^{(1)}(u_1; x, x')\right|, \tag{139}$$

$$\left|\dot{\Sigma}^{(1)}(u_1; x, x') - \mathbb{E}\big[\sigma'(z^{(1)}(u_1, x))\big]\mathbb{E}\big[\sigma'(z^{(1)}(u_1, x'))\big]\right| \leq C_{\sigma,2}^2 \left|\widetilde{\Sigma}^{(1)}(u_1; x, x')\right|, \tag{140}$$

$$\left|\dot{\Sigma}^{(1)}(u_1; x, x')\right| \leq C_{\sigma,2}^2 \left|\widetilde{\Sigma}^{(1)}(u_1; x, x')\right| + \left|\mathbb{E}\big[\sigma'(z^{(1)}(u_1, x))\big]\mathbb{E}\big[\sigma'(z^{(1)}(u_1, x'))\big]\right|, \tag{141}$$

Then, we have

$$\left|\dot{\Sigma}^{(1)}(u_1; x, x')\right| \leq C_{\sigma,2}^2 \left|\widetilde{\Sigma}^{(1)}(u_1; x, x')\right| + c_1, \tag{142}$$

where $c_1$ is bounded by $C_{\sigma,1}^2$. Plugging Eqn. (142) and Eqn. (137) into Eqn. (126) gives

$$|K_{\mathcal{W}}^{(1)}(x, x')| \leq |B_0(x, x')| \int_0^1 \left(C_{\sigma,2}^2 |\widetilde{\Sigma}^{(1)}(u_1; x, x')| + c_1\right) P_1(u_1; x, x')\, du_1 \tag{143}$$

$$\leq |B_0(x, x')| C_1'\left(\int_0^1 |\widetilde{\Sigma}^{(1)}(u_1; x, x')| \int_{[0,1]^2} \mathcal{W}^{(2)}(u_1, u_2)\, \mathcal{W}^{(3)}(u_2, u_3)\, du_2 du_3\, du_1\right), \tag{144}$$

where $C_1'$ is a constant depend on the $C_{\sigma,1}, C_{\sigma,2}, c_1$. Now, we substitute Eqn. (131) and use Fubini theorem:

$$\int_0^1 |\widetilde{\Sigma}^{(1)}(u_1; x, x')| \int_{[0,1]^2} \mathcal{W}^{(2)}(u_1, u_2)\, \mathcal{W}^{(3)}(u_2, u_3)\, du_3 du_2\, du_1 \tag{145}$$

$$\leq \int_0^1 \left(\int_0^1 \mathcal{W}^{(1)}(u_0, u_1)\, |\Sigma^{(0)}(u_0; x, x')|\, du_0\right) \int_{[0,1]^2} \mathcal{W}^{(2)}(u_1, u_2)\, \mathcal{W}^{(3)}(u_2, u_3)\, du_3 du_2\, du_1 \tag{146}$$

$$= \int_0^1 |\Sigma^{(0)}(u_0; x, x')| \left(\int_0^1 \mathcal{W}^{(1)}(u_0, u_1) \int_{[0,1]^2} \mathcal{W}^{(2)}(u_1, u_2)\, \mathcal{W}^{(3)}(u_2, u_3)\, du_3 du_2\, du_1\right) du_0 \tag{147}$$

$$= \int_0^1 |\Sigma^{(0)}(u_0; x, x')| \int_{[0,1]^3} \mathcal{W}^{(1)}(u_0, u_1)\mathcal{W}^{(2)}(u_1, u_2)\, \mathcal{W}^{(3)}(u_2, u_3)\, du_3 du_2 du_1\, du_0. \tag{148}$$

Combining Eqn. (144) and Eqn. (148) yields

$$|K_{\mathcal{W}}^{(1)}(x,x')| \lesssim |B_0(x,x')| \Big( \int_0^1 |\Sigma^{(0)}(u_0;x,x')| \times$$
$$\int_{[0,1]^3} \mathcal{W}^{(1)}(u_0,u_1)\mathcal{W}^{(2)}(u_1,u_2)\,\mathcal{W}^{(3)}(u_2,u_3)\,du_3 du_2 du_1\,du_0. \Big) \tag{149}$$

**Bounding $K_{\mathcal{W}}^{(2)}$ by path density.** Similarly, using the bounded second derivative of activation function, we have

$$0 \le |P_2(u_2;x,x')| \le C_{\sigma,1}^2 \int_0^1 W^{(3)}(u_2,u_3)\,du_3, \tag{150}$$

Next, we fix $u_2, x, x'$. The quantity $\dot\Sigma^{(2)}(u_2;x,x')$ is of the form $\dot F(s)$ with $s = \widetilde\Sigma^{(2)}(u_2;x,x')$. By Lemma G.3,

$$\big|\dot\Sigma^{(2)}(u_2;x,x')\big| \le C_{\sigma,2}^2 \big|\widetilde\Sigma^{(2)}(u_2;x,x')\big| + c_2. \tag{151}$$

Plugging Eqn. (150) and Eqn. (151) into Eqn. (127) gives

$$|K_{\mathcal{W}}^{(2)}(x,x')| \le |B_1(x,x')| \int_0^1 \Big( C_{\sigma,2}^2 |\widetilde\Sigma^{(2)}(u_2;x,x')| + c_2 \Big) P_2(u_2;x,x')\,du_2 \tag{152}$$

$$\le |B_1(x,x')| C_2' \Big( \int_0^1 |\widetilde\Sigma^{(2)}(u_2;x,x')| \int_0^1 \mathcal{W}^{(3)}(u_2,u_3)\,du_3\,du_2 \Big), \tag{153}$$

where $C_2'$ is a constant depend on the $C_{\sigma,1}, C_{\sigma,2}, c_2$. The pre-activation Gaussian cross-covariance at layer 2 is

$$\widetilde\Sigma^{(2)}(u_2;x,x') = \int_0^1 \mathcal{W}^{(2)}(u_1,u_2)\,\Sigma^{(1)}(u_1;x,x')\,du_1. \tag{154}$$

With fixed $u_1, x, x'$, the quantity $\Sigma^{(1)}(u_1;x,x')$ is of the form $F(s)$ with $s = \widetilde\Sigma^{(1)}(u_1;x,x')$. By Lemma G.2,

$$\big|\Sigma^{(1)}(u_1;x,x')\big| \le C_{\sigma,1}^2 \big|\widetilde\Sigma^{(1)}(u_1;x,x')\big| + c_1'. \tag{155}$$

where $c_1' = \big|\mathbb{E}[z^{(1)}(u_1;x)]\,\mathbb{E}[z^{(1)}(u_1;x')]\big|$. Combining Eqn. (155), Eqn. (154), and Eqn. (131) gives

$$\big|\widetilde\Sigma^{(2)}(u_2;x,x')\big| \lesssim \int_{[0,1]^2} \mathcal{W}^{(2)}(u_1,u_2)\,\mathcal{W}^{(1)}(u_0,u_1)\,\big|\Sigma^{(0)}(u_0;x,x')\big|, du_1 du_0. \tag{156}$$

Plugging Eqn. (156) into Eqn. (153) gives

$$|K_{\mathcal{W}}^{(2)}(x,x')| \lesssim |B_1(x,x')| \Big( \int_0^1 |\Sigma^{(0)}(u_0;x,x')| \times$$
$$\int_{[0,1]^3} \mathcal{W}^{(1)}(u_0,u_1)\mathcal{W}^{(2)}(u_1,u_2)\,\mathcal{W}^{(3)}(u_2,u_3)\,du_3 du_2 du_1\,du_0. \Big) \tag{157}$$

Combine the bounds for $K_{\mathcal{W}}^{(1)}$ and $K_{\mathcal{W}}^{(2)}$, and absorb all scaling terms we have

$$|K_{\mathcal{W}}(x,x')| \le |K_{\mathcal{W}}^{(1)}(x,x')| + |K_{\mathcal{W}}^{(2)}(x,x')| \tag{158}$$

$$\lesssim \Big( C_1 \Big| \int_0^1 \Sigma^{(1)}(u_1,x,x')du_1 \Big| + C_0 \Big| \int_0^1 \Sigma^{(0)}(u_0,x,x')du_0 \Big| \Big) \times \tag{159}$$

$$\Big( \int_0^1 |\Sigma^{(0)}(u_0;x,x')| \int_{[0,1]^3} \mathcal{W}^{(1)}(u_0,u_1)\mathcal{W}^{(2)}(u_1,u_2)\,\mathcal{W}^{(3)}(u_2,u_3)\,du_3 du_2 du_1\,du_0. \Big) \tag{160}$$

where $C_0, C_1$ are constant depending on $C_{\sigma,1}, C_{\sigma,2}$ and variance of activations and derivative activations $z^{(l)}$.

