# OpenReview forum: "Pruning at Initialisation through the lens of Graphon Limit: Convergence, Expressivity, and Generalisation"
_ICML.cc/2026/Conference — ICML 2026 regular_

### Official Review · Reviewer_GuHw · 2026-03-06

**Soundness:** 4
**Presentation:** 4
**Significance:** 2
**Originality:** 2
**Overall Recommendation:** 4
**Confidence:** 4

**Summary:**

This paper introduces the Factorized Saliency Model to analyze a broad class of PaI methods and shows that the discrete masks generated by these algorithms converge to graphons. This result validates the "graphon limit hypothesis" proposed in prior work. Building on this, the paper (1) proves a universal approximation theorem for the resulting sparse networks, and (2) derives a generalization bound using the graphon NTK.

**Compliance With Llm Reviewing Policy:**

Affirmed.

**Final Justification:**

The authors' rebuttal clarifies some of my original concerns regarding the novelty and significance of the theoretical contribution, as well as the work's practical insights. I acknowledge that I initially overlooked the technical depth of the theoretical results and, consequently, underestimated the contribution. I agree with the authors that this is an original, technically solid first step toward a formal theory for PaI methods. Hence, I have decided to raise my evaluation from 3 to 4, provided that the authors make revisions to the paper to clarify the *practical insights* of the theory.

However, some concerns remain: specifically, the strong assumptions regarding fixed input/output dimensions, the “few relevant coordinates” constraint, and the limited depth of the insights provided by the UAT. While the authors acknowledged that extensions in these directions are likely, they are not explored in the current manuscript, which largely limits the significance of this work.

**Key Questions For Authors:**

1. The paper studies a one-hidden-layer network and takes both the input dimension d and the hidden dimension n to infinity. I am concerned about the assumption $d\to\infty$, especially Assumption 4.2, which posits that inputs are discretizations of a bounded continuous function. This seems rather strong. In Appendix D, the setting is extended to L-hidden-layer networks, where the assumptions appear even stronger, as both input and output dimensions are taken to infinity. These assumptions diverge from standard infinite-width analyses, which typically keep input and output dimensions fixed. Is it possible to consider the more conventional case in which PaI masks for the hidden weights converge to graphons, while the masks for the input and output weights converge to  functions in $L([0,1], \mathbb{R}^d)$?

2. Relatedly, this set-up leads the authors to consider a universality result for “continuous targets that depend only on a fixed or slowly growing subset of k input coordinates.” This assumption feels unnatural: I do not agree with the claim that “learning problems typically have only a few coordinates carry signal while the rest behave as noise.” In many settings, the learning target genuinely depends on the entire signal (e.g., if the input is a discretized continuous function as in the assumption 4.2, the target could be a functional of that entire function). Could the authors discuss the expressivity in those cases?

**Limitations:**

The author does not seem to discusse the limitation of their work. I do not see any potential negative social impact of their work.

**Strengths And Weaknesses:**

**Strengths:**
1. The paper is clearly written. The necessary background is well explained, and the authors consistently provide intuitive remarks, making the work accessible.
2. The theoretical results are rigorous, with assumptions stated explicitly. To the best of my knowledge, the proofs appear correct.

**Weaknesses:**

My main concern is that the central contributions (the factorized saliency model, graphon convergence, universality, and generalization bounds) seem relatively straightforward and unsurprising, which limits the significance of the results beyond the closely related prior work Pham et al. (2025).

Particularly, the theoretical perspective does not appear to yield new insights that can improve our understanding of various PaI algorithms or inform practice, making me wonder what's the point of this theoretical point of view. The universality result relies on strong assumptions (see key questions below), and it's not really surprising; the generalization bound focus on the NTK lazy regime and thus not very realistic; it is also not supported by empirical validation linking the bound to observed generalization behavior in experiments, which makes it unclear when and how the bound explains generalization improvements.

---

> ### Author Rebuttal · Authors · 2026-03-28
>
> We thank the reviewer for your constructive comments. Below are our responses to your concerns:
>
> **Q1: Contributions feel straightforward relative to prior graphon-limit work. Theoretical perspective does not appear to yield new insights for PaI or inform practice.**
>
> **A1**: We appreciate this concern, but we respectfully disagree that the paper’s contribution is merely straightforward relative to prior graphon limit. The main contribution is of a theoretical nature. Previous work has postulated the Graphon Limit hypothesis, the current paper identifies rigorous mathematical settings that allow mathematical proof of graphon convergence, and subsequently derives universal approximation and generalization error bounds for sparse neural networks, which were not feasible prior to our work. Through these, we provided insights on why data-dependent pruning methods perform better than random at high sparsities. We would refer the reviewer to answers A2,3,4 of reviewer **jev6** for more details.
>
> **Q2: Strong assumptions, especially $d \to \infty$ and Assumption 4.2.**
>
> **A2**: We agree that these assumptions are strong. They reflect the asymptotic question studied here: to obtain a graphon limit for the bipartite pruning mask, both sides of the masked layer must grow. Assumption 4.2 is then the mechanism that provides the row-side continuum coordinate after relabeling. In the one-hidden-layer SNIP verification, this is realized through quantile convergence of sorted input features. More importantly, these assumptions are designed to be transferable to hidden layers: the same type of deterministic profile is what one would need later when raw inputs are replaced by hidden activations, isolating the core layerwise graphon mechanism in a mathematically clean form.
>
> **Q3: Appendix D extends to deep networks under even stronger assumptions than standard infinite-width analyses.**
>
> **A3**: We wish to clarify that we only consider an infinite L-hidden-layer network with a fixed output dimension. While we agree that infinite input dimension is a strong assumption, its role can be transferred to hidden layers as discussed above and in Appendix D.
>
> **Q4: Is it possible to consider the more conventional case in which PaI masks for the hidden weights converge to graphons, while the masks for the input and output weights converge to functions in $L([0,1], \mathbb{R}^d)$?**
>
> **A4**: Yes - we think this is a very natural and promising direction, and we appreciate the suggestion. A mixed graphon/function limit would be closer to standard infinite-width formulations and could broaden applicability. We do not claim to cover that regime in the current paper. However, we view our theorem as a first step that isolates the core graphon limit mechanism in a simpler setting. One reason we find the reviewer’s suggestion compelling is that it reinforces our main message: the graphon viewpoint is flexible and can serve as a genuine analytical framework for sparse architectures, rather than only a statement about one specific asymptotic regime.
>
> **Q5: The universality result relies on a restrictive “few relevant coordinates” assumption. What about targets depending on the full signal?**
>
> **A5**: We agree that the universality theorem is restricted. However, it is not true that this makes the result uninformative. In the joint limit $d \to \infty$, some notions of target complexity are unavoidable. Our theorem adopts an intrinsic-dimension regime [1] and shows that, under pruning-induced graphon structure, sparse networks can still retain approximation power through a persistent dense core. While sparse network approximation is understudied precisely because discrete random masks are difficult to analyze directly, the graphon convergence theorem gives a deterministic continuum object that makes this analysis possible.
>
> For targets that genuinely depend on the full discretized signal, we agree that the current theorem does not apply. Those cases require different ideas and likely a different structural notion of expressivity. But this does not weaken the present contribution: Section 5 identifies, for the first time in a rigorous way, a concrete mechanism by which PaI can preserve approximation power under the sparsity regime. We will revise the text to make this scope clearer.
>
> **Q6: The generalization bound is in the NTK lazy regime, may be unrealistic, and is not clearly linked to experiments.**
>
> **A6**: We also address a closely related question in our response to questions 3 and 4 of reviewer **jev6**, where we clarify that Section 6 is not a new NTK theorem from scratch, but a way to incorporate algorithmic sparsity into an existing lazy-training framework. We would refer the reviewer there for additional details.
>
>
> [1] Nakada et. al., Adaptive approximation and generalization of deep neural network with intrinsic dimensionality. JMLR 2020.

---

> > ### Author Rebuttal · Reviewer_GuHw · 2026-04-01
> >
> > While I agree that this work provides the first rigorous theoretical proof of what used to be only intuitively understood about the PaI method, I still think the contribution of the paper is somewhat limited by: (1) the strong assumptions which may be relaxed; (2) the theory is not surprising and does not lead to much useful insights.
> >
> > **(Minor) Technical details**
> >
> > I appreciate the authors’ honest acknowledgment of the strong assumptions. While assumptions are often necessary for rigorous theory, it seems that a few assumptions might be relaxable.
> >
> > > They reflect the asymptotic question studied here: to obtain a graphon limit for the bipartite pruning mask, both sides of the masked layer must grow.
> >
> > This justification seems driven by the choice of graphon tools rather than the intrinsic nature of the PaI problem. As mentioned in the review, alternative mathematical frameworks could handle limits without requiring both sides to grow.
> >
> > > We wish to clarify that we only consider an infinite L-hidden-layer network with a fixed output dimension.
> >
> > Thank you for the clarification. If I understand correctly, fixing the output dimension prevents analyzing PaI on the final layer (n_L × d_out). Is this consistent with how PaI is used in practice?
> >
> > > While we agree that infinite input dimension is a strong assumption, its role can be transferred to hidden layers as discussed above and in Appendix D.
> >
> > I agree that the infinite input assumption make sense for hidden layers. However, the analysis for input/output weights of a deep NN still relies on strong assumptions, leaving inherent limitations unresolved. I also agree with reviewer jev6 that much of the main text emphasizes shallow networks, whereas deeper, practically relevant cases are buried in the appendix.
> >
> > >  In the joint limit $d\to\infty$, some notions of target complexity are unavoidable.
> >
> > I agree, and the regime you study is interesting. Given that inputs are discretizations of continuous signals, could you consider alternative complexity notions aligned with this modeling choice, such as Lipschitzness of the underlying continuous signal?
> >
> > **(Major) Novelty and significance of theoretical contribution**
> >
> > I'd clarify why I found some contributions relatively straightforward at a high level. Could you clarify possible misunderstandings and stress the technicalities that I might have overlooked?
> >
> > - Graphon convergence: With the factorized saliency model, PaI decomposes where $\phi_i$ and $\psi_j$ discretize continuous signals, and $\xi_{ij}$ are iid conditional on $\phi,\psi$. This resembles classical random bipartite graph models whose convergence to deterministic graphons may follow from known results?
> >
> > - Universality: Graphon is a standard model for "dense graphs" where the size of the clique typically $\to\infty$ as the graph size $n\to\infty$ under mild assumptions. In this sense, universality is not surprising.
> >   - Honestly, I'm not sure whether universality is necessarily the theoretical result of interest here, because such an asymptotic result does not explain theusefulness of the PaI method. Instead, it's important to understand: what's the lost expressivity in applying PaI, and how the pruned wide model compares with a narrow model in terms of expressivity. The understanding of these questions seems to require non-asymptotic analysis.
> >
> > - Generalization bound: As noted by reviewer jev6, this also directly applies the known result. While I agree it's the first time aplying this to PaI setting, the contribution remains somewhat limited.
> >
> > **(Major) How much does the theory inform practical insights**
> >
> > I am comfortable with results being relatively straight forward if they yield actionable insights, but the practical guidance currently seems limited.
> >
> > >we provided insights on why data-dependent pruning methods perform better than random at high sparsities.
> >
> > This appears to be the primary practical insight, but the discussion (Remark 5.8 and Section 6.3) feels brief, and hand-wavy. Can those arguments be tested empirically? Does this suggest a comparison between different PaI methods? Does this inform us how to design good PaI methods? I think such insights are arguably most important, so exploring them and emphasizing in the narrative would strengthen the paper.

---

> > > ### Author Response · Authors · 2026-04-05
> > >
> > > We thank the reviewer for thoughtful comments. Below are our responses.
> > >
> > > **Q1: Fixed input/output dimension**
> > >
> > > A1: We agree that an important question is what happens when the input or output dimension is fixed (which is standard in NTK of dense setting) rather than sending them to infinity. If the input or output dimension is fixed, these layers would no longer converge to a standard graphon, instead, the natural limit object becomes a mixed discrete–continuous kernel, which needs non-trivial further investigation. Importantly, this does not remove the relevance of the framework: the hidden layers still admit genuine graphon limits under our layerwise assumptions.
> > >
> > > **Q2: Alternative complexity notions aligned with modeling choice, such as Lipschitzness of the underlying continuous signal?**
> > >
> > > A2: We agree this is a natural direction. While our current paper does not directly link input signal regularity to active-rectangle geometry, our graphon-convergence framework provides the right foundation. Informally, if inputs are discretizations of a Lipschitz signal $f$, then the row profile $\phi$ should inherit this regularity - e.g., under a regular grid, $\max_i|\phi_{n,i}-\phi(i/d)|=O(L_f/d)$ - which then shapes the limiting graphon. Signal complexity notions (Lipschitz/Sobolev) thus appear compatible with our framework. We therefore view this as an important next step: using the graphon limit to connect structural properties of the input signal to structural properties of the induced sparse topology.
> > >
> > > **Q3: Graphon convergence result resembles classical random bipartite graph convergence.**
> > >
> > > A3: While our abstract limit belongs to the graphon tradition, our novelty is *deriving* such a limit from concrete PaI algorithms rather than assuming one. To the best of our knowledge, Wolfe et al. [1] assume an exchangeable graphon $p_{ij}=\rho_n f(\xi_i,\xi_j)$, and RDPG [2] assumes latent vectors with edge probabilities $x_i^\top x_j$. By contrast, we start from a PaI saliency rule and *prove* convergence to $W(u,v)=\mathbb{P}(\phi(u)Q_\psi(v)|\xi|>\tau)$.
> > >
> > > The resulting limit is a threshold multiplicative link, not a bilinear kernel. Row/column coordinates derive from distinct neural objects (forward vs. backward features), requiring independent relabelings and quantile convergence on the column side. Finally, in SNIP/GraSP, weight $\theta_{ij}$ and pre-activation $h_j$ are deterministically coupled through the forward pass - precluding standard concentration inequalities. Thus, one cannot directly invoke a classical latent-position graph theorem, we instead rigorously bound the correlation remainder to establish asymptotic independence in the cut-norm topology (Appendices B.2, C.2).
> > >
> > > **Q4: Concerns on universality results**
> > >
> > > A4: While we agree that this universality result does not directly explain the usefulness of PaI methods, its purpose is basic but still important. In particular, for dense networks, UAT provides a foundational expressivity guarantee, for sparse networks, the corresponding question becomes even more pressing, because aggressive pruning can destroy connectivity in a highly combinatorial way. Thus, the natural question is whether a PaI-induced sparse topology still retains expressivity.
> > >
> > > And, we agree that a more informative next question is non-asymptotic expressivity loss under PaI. Addressing that would require controlling how often finite-width graphon-sparsified masks realize the needed dense cores, and how approximation degrades at finite width and very high sparsity.
> > >
> > > **Q5: Concerns on the practical insights**
> > >
> > > A5: Our key message is that, at high sparsity, good PaI masks should preserve path density on informative coordinates rather than allocating connectivity uniformly - the core intuition behind our active-rectangle argument and generalization analysis.
> > >
> > > Tables 1, 2 support this: random pruning remains competitive at moderate sparsity but degrades sharply at extreme sparsities, where uniform connectivity dilution becomes harmful. Our framework provides the first rigorous formalization of empirical pruning heuristics. Works like NPB [3] heuristically observed that good subnetworks must preserve "effective paths"  and balance surviving nodes. Our graphon viewpoint sharpens this - given a fixed connectivity budget, preserving path density on informative coordinates outperforms uniform distribution.
> > >
> > > Table 2: WideResNet20 - CIFAR10
> > > |Sparsity|68|90|96|99|
> > > |-|-|-|-|-|
> > > |SNIP|92.70|90.72|86.53|81.13|
> > > |Random|92.68|89.72|85.15|75.57|
> > >
> > > In addition, we believe this framework will allow us to extend the muP [4] rule to the training of more general sparse networks.
> > >
> > > [1] Wolfe et al. Nonparametric graphon estimation.
> > >
> > > [2] Athreya et al. Statistical inference on random dot product graphs: a survey.
> > >
> > > [3] Pham et al. Towards data-agnostic pruning at initialization: What makes a good sparse mask?
> > >
> > > [4] Dey et al. Sparse maximal update parameterization: A holistic approach to sparse training dynamics.

---

### Official Review · Reviewer_jev6 · 2026-03-12

**Soundness:** 3
**Presentation:** 3
**Significance:** 3
**Originality:** 4
**Overall Recommendation:** 5
**Confidence:** 2

**Summary:**

This paper studies pruning-at-initialization (PaI) through the lens of graph limit theory. The authors propose a factorised saliency model that aims to capture several PaI methods and analyze the asymptotic structure of the resulting pruning masks. Under certain regularity conditions, the paper shows that the binary connectivity patterns converge to deterministic bipartite graphons in the infinite-width limit. Building on this characterization, the paper further studies the implications for expressivity and generalization via an approximation result for sparse networks and a Graphon-NTK-based generalization bound, together with illustrative empirical visualizations.

**Compliance With Llm Reviewing Policy:**

Affirmed.

**Final Justification:**

Thank you for the detailed rebuttal. The responses are helpful and address several of my questions. And I will keep my original recommendation.

**Key Questions For Authors:**

1. The graphon convergence theorem is a central technical contribution. Could the authors clarify more explicitly which practical PaI methods satisfy the assumptions required for the convergence theorem in a strict sense, and which parts of the framework should instead be interpreted as asymptotic or heuristic approximations?
2. The universal approximation theorem relies on conditions such as the existence of an active rectangle and sufficiently many active coordinates. To what extent does the approximation result arise from the pruning mechanism itself, versus from these structural assumptions that effectively guarantee a dense subnetwork? Clarifying this would help understand how broadly the result applies in practice.
3. The generalization result is derived in a Graphon-NTK / lazy-training regime and appears to rely on Assumption F.1 together with existing NTK generalization bounds. Could the authors clarify more precisely which aspects of Theorem 6.1 constitute new theoretical contributions beyond prior NTK analyses?
4. The experiments provide visual evidence of graphon convergence and analyze a Graphon-NTK complexity proxy. Do the authors have additional empirical evidence showing that the heterogeneous graphon structures produced by methods such as SNIP or GraSP correlate with improved downstream test performance?

**Limitations:**

yes

**Strengths And Weaknesses:**

Strengths
1. The paper studies pruning-at-initialization (PaI) from a graph limit perspective and rigorously derives the graphon limit of pruning masks, which was previously conjectured but not formally established.
2. The proposed Factorised Saliency Model provides a clean abstraction that covers several PaI methods  and allows the analysis of pruning masks within a common mathematical framework.
3. The paper presents a structured progression from graphon convergence to implications for expressivity and generalization, rather than presenting isolated theoretical results.
4. The visualizations comparing finite-width pruning masks with their predicted graphon limits provide useful qualitative evidence for the theory.

Weaknesses
1. The graphon convergence result appears to be the most original contribution, while the later expressivity and generalization results are comparatively less strong.
2. The universal approximation theorem relies on assumptions such as the existence of an active rectangle with sufficiently many active coordinates, which may be difficult to guarantee in practice.
3. The generalization analysis is derived under a Graphon-NTK / lazy-training regime, which may not fully capture practical training dynamics.
4. The experiments mainly illustrate graphon convergence and a complexity proxy, but provide limited direct evidence linking the theoretical insights to improved predictive performance.
5. Many derivations focus on shallow network settings, and the extension to deeper architectures is less fully developed in the main analysis.

---

> ### Author Rebuttal · Authors · 2026-03-28
>
> We appreciate the reviewer for constructive feedback. Below are our answers to your concerns:
>
> **Q1: Graphon convergence seems the main novelty. Sections 5, 6 seem weaker.**
>
> **A1**: We agree that the graphon-convergence theorem is the paper’s strongest contribution. Our intent in Sections 5, 6 is to demonstrate why the resulting graphon limit is a useful analytical object. Only after the graphon convergence result is mathematically proved that we can have analysis on approximation and generalization. Put it differently, without the graphon convergence theorem, it is not clear to us how one can derive the universal approximation theorem and generalization error bound in a rigorous manner.
>
> **Q2: The universal approximation theorem relies on assumptions**
>
> **A2**: We agree that Theorem 5.7 is a restricted sufficient-condition result. However, the "active rectangle" is not an arbitrary assumption; it is the direct mathematical formalization of the data-dependent pruning mechanisms analyzed in Section 4. Our graphon convergence results prove that methods like SNIP/GraSP inherently concentrate their edge budget on task-relevant features. As Remark 5.8 shows, Random pruning ($\mathcal{W} \equiv \rho$) maintaining a dense core requires $n\rho^k \to \infty$, which becomes difficult at high sparsity. In contrast, data-dependent pruning can concentrate the same sparsity budget on relevant coordinates, yielding a larger effective floor $p^\star$ on an active block. This is also qualitatively consistent with our additional ResNet-20/CIFAR-10 results in Table 1, where Random is competitive at moderate sparsity but degrades much more sharply than SNIP at extreme sparsity.
>
> **Q3: Theorem 6.1 uses a Graphon-NTK / lazy-training regime and existing NTK bounds. What is new beyond prior NTK analyses?**
>
> **A3**: We agree that Theorem 6.1 relies on established NTK bounds [1]. Our contribution is therefore not a new NTK theorem from scratch, but the incorporation of algorithmic sparsity into this framework. Our graphon-convergence result translates discrete, chaotic PaI masks into a deterministic continuum kernel ($K_\mathcal{W}$), enabling us to mathematically evaluate sparse networks. This provides the missing bridge that makes existing NTK generalization machinery applicable to algorithmically pruned sparse networks, whereas prior NTK analyses are formulated for dense models and prior Graphon-NTK work assumes the graphon object a priori.
>
> Beyond the bound itself, Proposition 6.3 provides an interpretive decomposition showing how graphon-dependent path density shapes the $K_\mathcal{W}$. We view this as an additional analytical benefit of the graphon limits, helping explain which topological features of the limiting sparse connectivity may improve conditioning or label alignment, and therefore suggests design principles for sparse topologies.
>
> **Q4: Is there evidence linking the theory to downstream performance?**
>
> A4: To provide a more direct link to downstream performance, we add ResNet-20/CIFAR-10 results comparing SNIP and Random pruning at matched sparsities. These results show that Random is competitive with SNIP at medium sparsity, but degrades much more sharply at extreme sparsity. This result is qualitatively consistent with results in Figure 2 and the graphon/path-density picture: homogeneous random pruning uniformly dilutes connectivity, which becomes harmful at high sparsity, while data-dependent pruning can preserve higher path mass on informative coordinates.
>
> Table 1
> |Sparsity|68%|90%|96%|99%|
> |-|-|-|-|-|
> |SNIP|87.88|84.02|76.72|62.43|
> |Random|88.03|84.65|67.56|35.99|
>
> **Q5: The extension to deeper architectures is less fully developed.**
>
> **A5**: The deep networks extension is currently a conditional layerwise blueprint. Its role is to show that the graphon convergence mechanism is not inherently restricted to one hidden layer: once suitable layerwise analogues of the one-layer assumptions hold, the same limit picture can propagate through depth. Our current assumptions are set up so that we can obtain the result for deep networks by simply applying induction on layers. It would be desirable to have more refined analysis for deep models and more general settings. However, many technical difficulties arise with control conditional estimates on practical PaI saliency scores, which we leave for future work.
>
> **Q6: Which PaI methods satisfy the assumptions exactly, and which asymptotically?**
>
> **A6**: We will revise the paper to separate the regimes explicitly. Theorem 4.7 is proved for the abstract Factorized Saliency Model (covering Random and Magnitude). SNIP/GraSP are treated as one-hidden-layer asymptotic realizations / rank-equivalent approximations in the infinite-width limit, not exact finite-width instantiations.
>
> [1] Cao & Gu. Generalization bounds of stochastic gradient descent for wide and deep neural networks. NeurIPS 2019.

---

> > ### Author Rebuttal · Reviewer_jev6 · 2026-04-02
> >
> > Thank you for the detailed rebuttal. I am generally satisfied with the authors’ responses.

---

### Official Review · Reviewer_fXD1 · 2026-03-12

**Soundness:** 3
**Presentation:** 2
**Significance:** 2
**Originality:** 2
**Overall Recommendation:** 4
**Confidence:** 3

**Summary:**

This paper studies pruning at initialization through graphon limits. The core technical contribution is a convergence theorem showing that for one-hidden layer neural networks under a factorized saliency model, the resulting binary masks converge in bipartite cut distance to a deterministic graphon. The theoretical statements are supported by proof-of-concept experiments.

**Compliance With Llm Reviewing Policy:**

Affirmed.

**Final Justification:**

See rebuttal acknowledgment.

**Key Questions For Authors:**

- A sufficient condition for Assumption 4.2 for SNIP is given in section 4.2 using iid bounded input coordinates. Could a similar graphon convergence potentially also be established if this is relaxed, e.g. in structured domains such as images, where it seems unlikely that the coordinates would be converging to a single deterministic profile? perhaps the right object would be a random graphon or some dataset-averaged object? Or could you provide a realistic real-world example where this assumption is fulfilled?
- Figure 2 uses “binary CIFAR-10”, but I didn’t see the task construction being specified anywhere in the script. Could you clarify this?
- Could you explain the connection between proposition 6.3 and theorem 6.1?

**Limitations:**

yes

**Strengths And Weaknesses:**

Strengths:

- The core contribution is clear and interesting, namely turning the recently introduced Graphon Limit Hypothesis for pruning masks [1] into a convergence theorem under a factorized saliency model (eq. 3-4). This is a meaningful addition to a previously mainly empirical observation. The factorized saliency model seems fairly general and (approximately) applies to a variety of popular pruning at init methods (table 1).
- The exposition is clear and relatively easy to follow despite the theoretical nature.
- The paper also connects the graphon limit to expressivity and generalization. Even if these later results seem a bit more conditional to me, the attempt to build a more general theory of sparse network topology is ambitious and could inspire future work.

Weaknesses:

- The practical scope of Theorem 4.7 seems to be somewhat narrower than how the result is framed, as it is quite assumption-heavy. My main concern is about the assumption on the data distribution (Assumption 4.2, “deterministic input features”). Essentially, the main theorem needs the input feature of the saliency model to converge to a deterministic function under relabeling of the input coordinates. For SNIP specifically (l. 237 onwards, left) this is fulfilled by assuming input coordinates are bounded and iid. This is quite a strong exchangeability assumption and it seems implausible this would hold for many real-world data distributions, say, e.g., with fixed spatial semantics (such as MNIST, CIFAR etc).
- The expressivity result in section 5 seems fairly modest to me, as it is only proved in the joint limit $d, n \to \infty$ (where $d$ = ambient data dimension, $n$ = number of hidden neurons), while keeping the number of relevant coordinates $k$ for the target fixed, and the argument effectively relies on finding a dense subnetwork and then applying classical universal approximation.
- Section 6 on generalization seems slightly heuristic. Theorem 6.1 is essentially a direct invocation of standard NTK generalization results for the graphon NTK. The theory also does not predict the empirically observed better conditioning of the graphon NTK for heterogenous graphons, and it is not entirely clear to me how proposition 6.3 makes “the bound [from theorem 6.1] concrete”, as claimed in l. 365. Figure 2 evaluates only the term $\sqrt{y^\top K_W^{-1} y/m}$ on a binary CIFAR-10 task while varying the ratio of flipped labels and density. It might be more informative to also plot test error or generalization gap against this quantity.

[1] Hoang Pham, The-Anh Ta, Tom Jacobs, Rebekka Burkholz, and Long Tran-Thanh. *The Graphon Limit Hypothesis: Understanding Neural Network Pruning via Infinite Width Analysis.* NeurIPS 2025.

---

> ### Author Rebuttal · Authors · 2026-03-28
>
> We thank the reviewer for your constructive comments. Below are our point-to-point responses to concerns:
>
> **Q1: Assumption 4.2 is strong; what about structured domains / random graphons / realistic examples?**
>
> **A1**: We agree Assumption 4.2 is strong, and structured domains may require different limit objects, such as random-graphon or dataset-averaged limits. It is intended as a sufficient condition for the asymptotic question studied here. The bounded-i.i.d. SNIP setting is meant as one clean verification route, not as a universal data model. As typical for theoretical analysis of neural networks (universal approximation, NTK limit and theoretical generalization error bounds), simplified models and assumptions are necessary to enable rigorous mathematical analysis. One aims to find a certain theoretical setting which allows for the analysis of certain aspects of neural networks, which inherently requires sacrificing features of real-world models. In this sense, our framework and assumptions provide a new mathematical setting for pruning mask convergence of PaI methods, and consequently approximation theorem and generalization error of sparse neural networks.
>
> Besides, this assumption is designed to be transferable to hidden layers: the same type of deterministic profile is what one would need later when raw inputs are replaced by hidden activations (Appendix D), isolating the core layerwise graphon mechanism in a mathematically clean form.
>
> **Q2: The expressivity result in section 5 seems fairly modest and tied to fixed $k$.**
>
> **A2**: We agree that the expressivity theorem is restricted to fixed $k$. That fixed $k$ is intentional in our framework, since the paper studies the joint limit $d,n\to\infty$, a full universal approximation theorem (UAT) over all $d$ coordinates is not the right notion. Rather, Section 5 is a restricted sufficient-condition result showing one concrete mechanism by which sparsity can still preserve approximation power: when the limit graphon contains an active block with a nontrivial edge floor, the sparse mask contains a dense core, and classical approximation arguments can be transferred to that core. It should be noted that sparse network approximation is much less understood than the dense case, largely because discrete pruning masks are hard to analyze rigorously. Our graphon convergence result makes such questions tractable. We believe that it is possible to have more refined and quantitative subsequent results.
>
> **Q3: Novelty of Section 6 and link between Theorem 6.1 and Proposition 6.3**
>
> **A3**: We agree that Theorem 6.1 relies on established NTK bounds [1]. Our contribution is therefore not a new NTK theorem from scratch, but the incorporation of algorithmic sparsity into this bound. Standard NTK analysis assumes dense connectivity, making it fundamentally incapable of explaining pruning. Our graphon convergence result provides the missing bridge: it translates discrete, chaotic PaI masks into a deterministic continuum kernel ($K_\mathcal{W}$), enabling us to mathematically evaluate sparse networks.
>
> Based on our graphon convergence results, we are able to derive from [1], as a consequence, a generalization error bound for sparse neural networks (Theorem 6.1). Crucially, we can interpret the obtained generalization error bound in terms of path density of sparse models (Proposition 6.3). This provides rigorous mathematical proof for the widely accepted heuristics from previous empirical works that path density plays an important role in the performance of sparse networks [2, 3].
>
> **Q4: Generalization error bound experiments.**
>
> **A4**: We thank the reviewer for pointing this out. We consider 2-hidden-layer network on the first two classes of CIFAR10 dataset. We use wide finite-width proxies (n = 4096), pruning is applied to all layers. We vary the label noise (ratio of flipped labels) and the network densities.
>
> To add downstream evidence, we report the performance of SNIP and Random pruning on ResNet-20 on CIFAR10 in Table 1. Random is competitive at moderate sparsity but degrades sharply at extreme sparsities. This aligns with our observations in Figure 2 and graphon/path-density picture, demonstrating the usefulness of graphon in understanding PaI methods. In particular, Random pruning (a homogeneous graphon) uniformly dilutes path density, causing the NTK to collapse at extreme sparsity. Conversely, SNIP (a heterogeneous graphon) concentrates path density on informative features, preserving kernel alignment.
>
> Table 1
> |Sparsity|68%|90%|96%|99%|
> |-|-|-|-|-|
> |SNIP|87.88|84.02|76.72|62.43|
> |Random|88.03|84.65|67.56|35.99|
>
> [1] Cao & Gu. Generalization bounds of stochastic gradient descent for wide and deep neural networks. NeurIPS 2019.
>
> [2] Tanaka et. al., Pruning neural networks without any data by iteratively conserving synaptic flow. NeurIPS 2020.
>
> [3] Pham et. al., Towards data-agnostic pruning at initialization: What makes a good sparse mask?. NeurIPS 2023.

---

> > ### Author Rebuttal · Reviewer_fXD1 · 2026-04-03
> >
> > I thank the authors for their thoughtful rebuttal. The clarification of the connection between Theorem 6.1 and Proposition 6.3 adequately addressed my concern.
> >
> > I nevertheless continue to have some reservations about the scope of the work, which I think is limited by the rather strong data assumption (as also discussed in the review/rebuttal, under more realistic assumptions one might arrive at a different limiting object altogether). I also remain unconvinced by the universal approximation argument that relies on the sparse network just containing a large dense subnetwork (I was wondering if one could show similar statements for more realistic data distributions on the ambient space $\mathbb{R}^d$, perhaps under some assumptions on spectral decay, i.e. that low-frequency components which vary smoothly in the ambient coordinates dominate). On these points, the rebuttal did not substantially change my assessment.
> >
> > That said, the authors have addressed my main objections well enough that I am willing to raise my score to weak accept. However, this remains a very borderline decision for me, so while I lean slightly in favor of acceptance, I would not advocate for it particularly strongly, as I think the work reads as somewhat preliminary and could greatly benefit from further development that would confirm the "graphon limit hypothesis" in more general settings.

---

### Official Review · Reviewer_c6UV · 2026-03-12

**Soundness:** 3
**Presentation:** 3
**Significance:** 2
**Originality:** 2
**Overall Recommendation:** 4
**Confidence:** 4

**Summary:**

This paper studies pruning at initialization through a graphon-limit perspective. Its main claim is that, for a factorized family of saliency scores intended to cover methods such as Random, Magnitude, SNIP, GraSP, and SynFlow, the induced sparse masks converge in the infinite-width limit to deterministic graphons. On top of that limit object, the paper 1) develops a structural interpretation of different pruning rules, 2) an active-subspace approximation result and 3) a Graphon-NTK-based generalization analysis. Overall the paper’s contribution is a mathematical attempt to recast pruning-at-initialization as an asymptotic problem about the limiting topology of sparse networks, rather than only about finite-width saliency scores.

**Compliance With Llm Reviewing Policy:**

Affirmed.

**Final Justification:**

My initial assessment (that the paper cannot be published in its current form) has been partially addressed with the authors' rebuttal. I hope that the authors will make a serious effort to address all the review comments (not only mine) in the final version of the paper.

**Key Questions For Authors:**

I do not think that the paper can be published yet because it clearly omits several highly relevant prior works, without comparing the proposed scheme with them and without even discussing how they are relevant:

NPB (“Towards Data-Agnostic Pruning at Initialization”) is also highly relevant because it studies PaI from the joint perspective of effective paths and effective nodes, arguing that good masks arise from balancing these global structural properties rather than from local scores alone.

Gebhart et al. (“A Unified Paths Perspective for Pruning at Initialization”) should be cited because it provides a path-centric NTK unification of SNIP, GraSP, and SynFlow via the Path Kernel, explicitly separating architecture-dependent from data-dependent effects.

PHEW (Patil and Dovrolis) is important because it analyzes pruning-at-initialization through path structure and NTK/path-kernel considerations, arguing that global architectural properties such as width preservation can strongly affect sparse network trainability and generalization,

Hoang et al. (“Revisiting Pruning at Initialization through the Lens of Ramanujan Graph”) is relevant because it interprets PaI masks through graph-theoretic structure and proposes connectivity metrics to characterize sparse subnetworks, showing that graph properties can correlate with performance.

Liu et al. (“Learning effective pruning at initialization from iterative pruning”) should also be cited because it shows that PaI criteria can be learned from stronger iterative-pruning signals rather than handcrafted analytically, which is relevant for judging the practical significance of a theory centered on classical saliency rules such as SNIP/GraSP/SynFlow.

Navarrete et al. (“What Scalable Second-Order Information Knows for Pruning at Initialization”) is directly relevant to the paper’s discussion of GraSP-like criteria, since it revisits scalable second-order PaI methods and reports strong empirical results with curvature-informed approximations.

----

Some additional questions/concerns about the math results:
1) The proof of the main graphon-convergence result relies on conditional independence of edge noises, whereas in SNIP/GraSP the saliency terms are coupled through shared hidden activations. The appendix only argues an asymptotic decoupling heuristic rather than establishing the independence structure needed by the theorem.

2) The deep-network extension is rather weak. Rather than proving the result for practical deep PaI methods, the paper assumes layerwise analogues of the one-layer assumptions and then applies the same theorem independently to each layer

3) Proposition 6.3 and its proof do not seem rigorous in their current form. That part has notation inconsistencies, loosely introduced constants and several inequalities I could not explain (within the limited time I could spend on the proofs), making it hard to verify the final bound

**Limitations:**

the limitations section should be more clear about scope: most of the theory applies to an abstract factorized saliency model or to simplified one-hidden-layer settings, while the connection to practical SNIP/GraSP-style pruning and to deeper networks is more limited than the paper’s broader framing suggests

**Strengths And Weaknesses:**

Soundness: The paper is interesting from the technical perspective but I have concerns about how far the results really go. The main graphon-convergence theorem appears to apply cleanly only to an abstract factorized saliency model. The extension to actual methods such as SNIP and GraSP seems only partial and relies on asymptotic arguments that are weaker than the independence structure used in the proof. The expressivity and generalization results also seem less informative than advertised: the approximation theorem is fairly limited, while the Graphon-NTK result depends on strong assumptions. On the experimental side, the evidence is mostly visual and qualitative with limited quantitative validation that the proposed graphon objects actually explain downstream performance.

Originality: The paper has some originality in trying to prove graphon convergence for a factorized saliency family . But the paper is much less original than it claims. The broader framing (namely that PaI methods should be analyzed through global sparse structure rather than only local scores) is developed in multiple prior works (see next section of the review), and even the graphon viewpoint itself had already been introduced in the earlier "Graphon Limit Hypothesis" paper. So the paper feels novel in a narrow technical sense but substantially incremental in how it positions its ideas relative to the literature.

Presentation: The related-work discussion is clearly incomplete, omitting several highly relevant papers that affect the paper’s positioning and novelty claims (see next review section). In addition, the paper blurs the distinction between what is rigorously proved for the abstract factorized model, what is argued only asymptotically for SNIP/GraSP in simplified settings, and what is merely assumed for deeper networks or Graphon-NTK training dynamics. The empirical section is also too qualitative to support the paper’s broader claims with confidence

Significance: The pruning at initialization problem is important. However, the contribution is more limited than the paper suggests. Much of the broader motivation (that pruning-at-initialization should be understood through global structure, paths, topology, and architecture-level effects) has already been explored in several cited or uncited papers, including PHEW, NPB, the unified-paths paper, and the Ramanujan-graph paper. The paper’s significance rests mainly on its specific graphon formalization but the empirical section does not show that this formalism yields substantially new predictive or practical insight beyond those earlier perspectives.

---

> ### Author Rebuttal · Authors · 2026-03-28
>
> We thank the reviewer for your constructive feedback. Below are our point-to-point responses to your concerns and questions:
>
> **Q1: Exact theorem vs asymptotic SNIP/GraSP**
>
> A1: We will revise the paper to separate the regimes explicitly. The Graphon Convergence Theorem is rigorously proven for the Factorised Saliency Model, covering Random and Magnitude. SNIP/GraSP are treated as one-hidden-layer asymptotic realizations / rank-equivalent approximations in the infinite-width limit, not exact finite-width instantiations.
>
> **Q2: The expressivity and generalization results also seem less informative than advertised**
>
> **A2**: We agree that Section 5, 6 are conditional consequences of the graphon convergence theorem. Our convergence result provides a theoretical ground for deriving basic approximation and generalization results. In other words, without a rigorous proof of graphon limit convergence, it is not clear to us how one can approach universal approximation theorem and NTK-based generalization error bounds for sparse neural networks. Proposition 6.3 further gives an interpretive path-density view of how topology can affect the bound.
>
> **Q3: Empirical section is mostly visual/qualitative and does not establish downstream predictive insight**
>
> **A3**: We thank the reviewer for raising this point. To add downstream evidence, we report the performance of SNIP and Random pruning on ResNet-20 on CIFAR10 in Table 1. Random is competitive at moderate sparsity but degrades sharply at extreme sparsity, qualitatively consistent with the graphon/path-density picture and results in Figure 2. In particular, Random pruning (a homogeneous graphon) uniformly dilutes path density, causing the NTK to collapse at extreme sparsity. Conversely, SNIP (a heterogeneous graphon) concentrates path density on informative features, preserving kernel alignment.
>
> Table 1
> |Sparsity|68%|90%|96%|99%|
> |-|-|-|-|-|
> |SNIP|87.88|84.02|76.72|62.43|
> |Random|88.03|84.65|67.56|35.99|
>
> **Q4: Overclaiming novelty relative to prior global path/topology/structure papers and missing literature**
>
> **A4**: We thank the reviewer for highlighting excellent relevant PaI papers, which we will add and discuss them rigorously in the next version. While these works brilliantly analyze global structure using finite-width combinatorial metrics (e.g., discrete path/node counting, path kernel, graph-theoretic properties, etc.,), our novelty lies in transitioning to rigorous continuous limits through the graphon object. For instance, the "effective paths" and "node-path balance" empirically identified in NPB/PHEW correspond naturally in our framework to the continuous path-density integrals within the Graphon-NTK (Proposition 6.3). We do not claim to invent path-based pruning analysis; rather, we provide its foundational infinite-width mathematical theory.
>
> **Q5: Conditional independence in the theorem vs coupling in SNIP/GraSP**
>
> **A5**: We agree that for SNIP/GraSP the theorem’s conditional-independence structure is not verified in the exact finite-width sense. Indeed, our current appendices consider the infinite-width setting in the one-hidden-layer network, the scores admit an asymptotically factorized/rank-equivalent approximation, with residual coupling terms vanishing under the infinite scaling.
>
> **Q6: Deep-network extension is weak, assumes layerwise analogues**
>
> **A6**: We agree that Appendix D is a conditional extension blueprint, not a full theorem for practical deep PaI. Its purpose is to show that the graphon-convergence framework is not inherently restricted to one hidden layer: our setup is designed so that, once suitable layerwise analogues are established, the one-layer argument can be applied recursively through depth. This is meant in the same spirit as early infinite-width NTK analyses, which first identified the recursive limiting structure of deep networks, while sharper quantitative finite-width control came later. We already sketched one key ingredient - deterministic empirical-law/quantile convergence of hidden activations under standard infinite-width concentration. The harder remaining step is to show that practical deep PaI saliency scores satisfy the full layerwise factorized/decoupled structure of Theorem 4.7, with stable relabelings across depth and stronger quantitative control at finite width. We will revise Appendix D to present it explicitly.
>
> **Q7: Proposition 6.3 appears not rigorous, notation and inequalities unclear**
>
> **A7**: We will revise the notational inconsistencies in Appendix G to clear the confusion. Our intent there was to provide an interpretive path-density decomposition that makes the role of graphon topology in the Graphon-NTK more transparent.

---

> > ### Author Rebuttal · Reviewer_c6UV · 2026-04-03
> >
> > I appreciate the authors' rebuttal but, as I explained in my review, the paper is not ready for publication yet. Please refer to the sections about Soundness and Significance in the review. The rebuttal comments do not address those concerns -- they mostly acknowledge them.

---

> > > ### Author Response · Authors · 2026-04-06
> > >
> > > We thank the reviewer for their follow-up. We believe there may be a misunderstanding regarding the extent of the evidence provided in our rebuttal. Rather than merely acknowledging limitations, we have provided new technical results and clarifications that directly address the core concerns:
> > >
> > > **Q1/Q5 (exact vs. asymptotic scope; conditional independence):** We clarified the three regimes precisely: Random and Magnitude satisfy the Factorised Saliency Model exactly at finite width, while SNIP/GraSP are treated as asymptotic one-hidden-layer realizations / rank-equivalent approximations, with residual coupling terms vanishing in the infinite-width limit as shown in Appendices B.2 and C.2. This helps resolve the scope ambiguity by separating what is exact from what is asymptotic.
> > >
> > > **Q3 (empirical evidence):** We added new quantitative downstream results on ResNet-20 on CIFAR-10 across four sparsity levels. These results were not in the original submission and provide direct evidence for the qualitative prediction from our graphon/path-density picture: Random and SNIP behave similarly at moderate sparsity but diverge sharply at extreme sparsity.
> > >
> > > **Q4 (literature and novelty):** We gave a point-by-point explanation of how our contribution differs from prior finite-width combinatorial/path-based analyses. Our claim is not to introduce the general idea that topology matters, but to provide a rigorous continuum limit for a factorised saliency family, and to connect path/topological quantities to a deterministic graphon object and its associated Graphon-NTK analysis.
> > >
> > > **Q6 (deep-network extension):** We clarified that Appendix D is a conditional extension blueprint rather than a full theorem for practical deep PaI, and identified the precise remaining technical step needed for a complete result.
> > >
> > > More broadly, we wish to emphasise that the graphon convergence theorem is the foundational contribution of the paper. It replaces a discrete sparse mask, which is difficult to analyze directly, with a deterministic continuum object. This is exactly what enables the downstream analyses in Sections 5 and 6: the expressivity results become statements about graphon topology/active substructures, and the generalization analysis can be grounded in a deterministic Graphon-NTK object and its path-density decomposition.

---

### Decision · Program_Chairs · 2026-04-30

**Decision:**

Accept (regular)

**Comment:**

This paper makes steps toward theoretically analyzing expressivity and generalization in sparse neural networks obtained through pruning at initialization. It connects discrete pruning heuristics to graph limit theory via graphons. The rebuttal was engaging, and the reviewers concerns have been sufficiently addressed by the authors. Consequently, some reviewers increased their recommendations. In the end, all reviewers are positive about acceptance, while two of them are still expressing concerns about the amount of work required to polish the paper for the camera-ready version. The authors are advised to seriously consider all reviewers comments and suggestions, along with the rebuttal discussion, in the final version of the paper.